# Characterization of dark current signal measurements of the ACCDs used on-board the Aeolus satellite

Fabian Weiler[1], Thomas Kanitz[2], Denny Wernham[2], Michael Rennie[3], Dorit Huber[4], Marc Schillinger[5], Olivier Saint-Pe[5], Ray Bell[6], Tommaso Parrinello[7], Oliver Reitebuch[1]

[1]Deutsches Zentrum für Luft- und Raumfahrt, Institut für Physik der Atmosphäre, Oberpfaffenhofen, Germany
[2]European Space Agency-ESTEC, Keplerlaan 1, Noordwijk NL-2201 AZ, The Netherlands
[3] European Centre for Medium-Range Weather Forecasts, Shinfield Park, Reading RG2 9AX, United Kingdom
[4] DoRIT, 82239 Alling, Germany
[5] Airbus Defence and Space (Toulouse), Rue de Cosmonautes, 31400 Toulouse, France
[6] Teledyne e2v, 106 Waterhouse Lane, Chelmsford Essex CM1 2QU, United Kingdom
[7] European Space Agency-ESRIN, Largo Galileo Galilei 1, 00044 Frascati RM, Italy

*Correspondence to*: Fabian Weiler (Fabian.Weiler@dlr.de)

**Abstract.** Already shortly after the successful launch of the European Space Agency satellite Aeolus in August 2018, it turned out that dark current signal anomalies of single pixels (so-called “hot pixels”) on the Accumulation-Charge-Coupled Devices (ACCDs) of the Aeolus detectors detrimentally impact the quality of the aerosol and wind products potentially leading to wind errors of up to 4 m/s. This paper provides a detailed characterization of the hot pixels which occurred during the first one and a half years in orbit. The hot pixels are classified according to their characteristics to discuss their impact on wind measurements. Furthermore, mitigation approaches for the wind retrieval are presented and potential root causes for the hot pixel occurrence are discussed. The analysis of the dark current signal anomalies reveals a large variety of anomalies ranging from pixels with Random Telegraph Signal (RTS)-like characteristics to pixels with sporadic shifts in the median dark current signal. Moreover, the results indicate that the number of hot pixels has almost linearly increased during the observing period between 2018-09-02 until 2020-05-20 with 6 % of the ACCD pixels affected in total at the end of the period leading to 9.5 % at the end of mission lifetime. This work introduces dedicated instrument calibration modes and ground processors which allowed for a correction shortly after a hot pixel occurrence. The achieved performance with this approach avoids risky adjustments to the inflight hardware operation. It is demonstrated that the success of the correction scheme varies depending on the characteristics of each hot pixel itself. With the herein presented categorization, it is shown that multi-level RTS pixels with high fluctuation are the biggest challenge for the hot pixel correction scheme. Despite a detailed analysis in this framework, no conclusion could be drawn about the root cause of the hot pixel issue.

## 1 Introduction

The European Space Agency (ESA) satellite Aeolus was successfully launched into space on 22 August 2018 (Reitebuch et al., 2020). Aeolus was selected as one of the Earth Explorer Missions of the "Living Planet Programme" in 1999, (ESA, 2008). The satellite is equipped with the Doppler Wind Lidar (DWL) instrument ALADIN (Atmospheric LAser Doppler INstrument) to acquire wind profiles of the horizontal wind vector in the line-of-sight (LOS) direction of the instrument on a global scale from the ground up to 30 km altitude (Stoffelen et al., 2005). In doing so, Aeolus fills a major gap in the Global Observing System (Andersson, 2018). This has been already successfully confirmed as Aeolus data is already being assimilated into numerical weather prediction (NWP) models since May, 2020 (Rennie and Isaksen, 2020). Aeolus circles the earth in a sun-synchronous dusk/dawn orbit at an altitude of 320 km and with a repeat cycle of one week. In addition to wind products, Aeolus provides continuous measurements of aerosol and cloud properties such as backscatter and extinction coefficients (Ansmann et al., 2007; Flamant et al., 2008).

ALADIN operates at a wavelength of 354.8 nm and is designed to measure wind speed using the motion of aerosols and molecules (Reitebuch, 2012a). It consists of a laser emitter, a telescope in monostatic configuration, and a receiver unit to analyze the Doppler shift of the collected backscatter light. The receiver unit incorporates novel techniques that have never been applied in space before such as the innovative sequential arrangement of two optical spectrometers to measure the Doppler shift of the molecular scattering (Rayleigh channel) and scattering from aerosols as well as cloud droplets (Mie channel). Another novelty is the use of Accumulation Charge Coupled Devices (ACCDs) in the detection chain which allows to collect the output of the spectrometers with a high detection efficiency (ESA, 2008; Reitebuch et al., 2018). Thereby, ALADIN with its novel detection concept and as the first instrument ever to operate ACCDs in space environment has broken completely new ground.

Due to the measurement principle and the instrument design, the accuracy of Aeolus wind measurements is very sensitive to changes in the dark current of the ACCDs. Quite unexpectedly, already shortly after launch single pixels of the ACCDs showed suspicious behavior with increased dark current signals which lead to systematic errors in the wind results (Reitebuch et al., 2020). In order to monitor the evolution of dark current anomalies, a new dedicated dark current calibration technique has been introduced and performed throughout the mission on a regular basis providing a unique dataset for investigation. This paper presents a detailed characterization of the performance of the ACCDs during the first one and a half years in orbit (2018-09-02 until 2020-05-20). In particular, the various dark current anomalies are classified into categories. The categorization provides the basis to discuss the impact of these anomalies on the wind measurements and its correction in the wind retrieval. Furthermore, possible root cause scenarios for the increased dark currents are discussed.

This manuscript is structured as follows: The first section briefly describes the setup of the ALADIN instrument with a focus on the design and operating principle of the detection unit. An overview of typical anomalous dark current signal behavior for Charge Coupled Devices (CCDs) is provided, followed by the introduction to the Aeolus dark signal characterization method. The next section explains methods to analyze the dark signal time series and to detect anomalies as well as the monitoring of

the impact of uncorrected anomalies on the wind measurements. The processing and the correction of dark signals will be discussed by two case studies in detail in Sect. 3.2. Afterwards, the information gained on all pixels will be presented. The paper finishes with a discussion on potential root causes of the anomalies, mitigation approaches, and a summary.

## 2 ALADIN and dark current measurements

This section provides an overview of the design and measurement principle of ALADIN followed by a description of the in-orbit dark current measurements. Furthermore, typical dark current anomalies of CCDs are briefly discussed. Note that herein only a brief description of ALADIN is provided. Further information about the satellite, instrument and its products can be found in Straume et al. (2020), Kanitz et al. (2020) and Reitebuch et al. (2020). Detailed information about the laser employed in ALADIN can be found in Lux et al. (2020).

### 2.1 Measurement principle

Aeolus orbits the Earth in relatively low orbit at an altitude of ~320 km and carries one single instrument, the direct-detection wind lidar ALADIN. The instrument emits short 20-ns laser pulses at a repetition rate (PRF) of 50.5 Hz and a wavelength of 354.8 nm (Lux et al., 2020) into the atmosphere where the light is scattered by molecules and particles. Part of the outgoing light is also diverted to the detectors serving as internal reference to measure the frequency of the transmitted laser pulse. The backscatter light from the atmosphere is collected by a telescope with a diameter of 1.5 m and directed to the receiver unit. The receiver consists of two complementary channels to analyze the signal return from molecules (Rayleigh channel) and aerosols as well as cloud returns (Mie channel). The Mie channel which analyzes narrow bandwidth aerosol returns incorporates a Fizeau interferometer (FIZ) and is based on the fringe imaging technique (McKay, 1998). To measure the broad bandwidth return from molecules, the double-edge technique is used in the Rayleigh channel which is made up of two sequential Fabry Perot interferometers (FPI) (Chanin et al., 1989; Flesia and Korb, 1999).

The signal detection in both channels is based on CCDs. In fact, the use of CCDs as detectors is quite unusual and novel for DWL systems with only a few examples (Irgang et al., 2002; Reitebuch et al., 2009). Other detectors typically used for spaceborne lidars are based on photomultipliers (Markus et al., 2017; Winker et al., 2009) and avalanche photodiodes (Sun et al., 2006), such as on-board of the CALIPSO and ICESat satellites. In Aeolus, however, Doppler shifts in the Mie channel are measured by determining physical displacements of fringes imaged onto a detector. Thus, a detector with several sensitive areas such as CCDs is needed to resolve the spectrometer output (Reitebuch, 2012b). For the Rayleigh channel no image detection, i.e. CCD, would be necessary. Nevertheless, the Rayleigh channel was equipped with the same kind of CCD as the Mie channel to record the two circular spots of the FPI output. CCDs provide a very high quantum efficiency - 85 % at 355 nm reached by the Aeolus CCDs - in combination with a very low noise factor which cannot be simultaneously offered by

available avalanche photodiodes or photomultipliers. In the case of Aeolus, special CCDs are used, so-called accumulation CCDs. This allows the accumulation of backscatter atmospheric signals for consecutive laser pulses already on the chip in a dedicated memory zone to reduce the impact of read-out noise (ESA, 2008).

Two ACCDs manufactured by Teledyne e2V are used to detect the signals in both channels. Table 1 lists the main specifications of the Aeolus ACCDs. Each ACCD is a thinned back-illuminated silicon CCD and is mounted in a thermo-controlled housing with a 45 mm x 25 mm sized window (see Fig. 1, right). The ACCDs provide a high quantum efficiency of about 85 % optimized at a wavelength of 355 nm and a Charge Transfer Efficiency (CTE) of 99.99 %, meaning that only $1 \cdot 10^{-4}$ of all charges are lost per transfer of one row. The ACCDs consist of the illuminated imaging zone and the non-illuminated memory zone. The imaging zone has an area of 0.43 mm x 0.43 mm and is made up of 16 x 16 squared pixels with a pixel size of 27 µm x 27 µm. The memory zone has a size of 0.43 mm x 0.75 mm and has 25 x 32 pixels with a pixel size of 13.5 µm x 30 µm. Sixteen columns of the memory zone are the equivalent to the imaging zone and form the transfer section of the memory zone. A further sixteen columns interleaved between them form the memory storage section in which signal accumulation is performed. Figure 1 (left) indicates the imaging and the storage section of the memory zone for the two channels.

It illustrates how the two circular Rayleigh spots from the FPI and the Mie fringe from the FIZ are imaged on the ACCDs imaging zone. In the imaging zone the atmospheric return signal is integrated over time based on the settings for the vertical range gate timings. In Aeolus operations the range gate timings can be varied from 2.1 µs to 16.8 µs which correspond to a vertical sampling of 250 m to 2000 m, respectively, considering the 35° off-nadir viewing angle of the instrument. Subsequently, the signals of the imaging zone are pushed downwards and accumulated in the transfer row. The image zone is completely shifted within 1.0 µs. Afterwards, the signals are moved in the transfer columns where each of the 25 rows corresponds to one vertical range gate of the atmospheric profile. Once the signals of all range gates are acquired in the transfer section of the memory zone, the signals are horizontally shifted from the transfer to the storage columns of the memory zone. This concept allows on-chip signal accumulation over multiple successive atmospheric returns to the so-called “measurement” level. The number of accumulated pulses can be varied between 1 and 50. For the herein analyzed dark current measurements the number of pulses was 19 until Jan 2019 and then 18 to avoid a potential conflict in the onboard data management. The resulting residence time of the signals in the memory zone is on the order of 0.4 s, considering the PRF. After each accumulation sequence, the charges of the memory zone are read-out via the read-out register at a very low frequency of 48 kHz to minimize read-out noise and are further transferred to the Detection Electronics Unit (DEU) (Reitebuch et al., 2018). Here, the accumulated charges are digitized with 16-bit accuracy and converted into units of Least Significant Bits (LSB). The conversion rate of this process, also called radiometric gain, is about 0.68 LSB/e- and 0.44 LSB/e- for the Mie and Rayleigh channel, respectively.

While the signal acquisition process is the same for each vertical range gate, it has to be mentioned that for the atmospheric measurements only the first 24 range gates are used with the timings mentioned above. The 25$^{th}$ row is used to measure the solar background signal. For this purpose, the integration time for the in-orbit observations is set to 3564 µs. Measurements of

the solar background are used in the wind retrieval and in calibrations (see Sect. 2.3) to correct atmospheric signals from solar background contamination.

In addition to the solar background measurement, the Aeolus ACCDs also make use of so-called “overscan” or “virtual” pixels to determine the Detection Chain Offset (DCO). Virtual pixels provide zero-charge read-outs and are added before the readout process as electronic voltage offset to avoid negative values in the digitization and are measured for each range gate.

While the onboard accumulation process results in the so-called “measurement” level, ground processing optimizes the measurements further to so-called “observations”. For the analyzed dark current characterization measurements, the number of measurements per observation was 30 which leads to a duration for one observation of about 12 s.

For the following analysis, the row and column index is used to describe the position of the pixel on the ACCDs. For instance, Mie [15, 13] refers to row #15, and column #13 (counting starts at one) of the memory zone of the Mie ACCD.

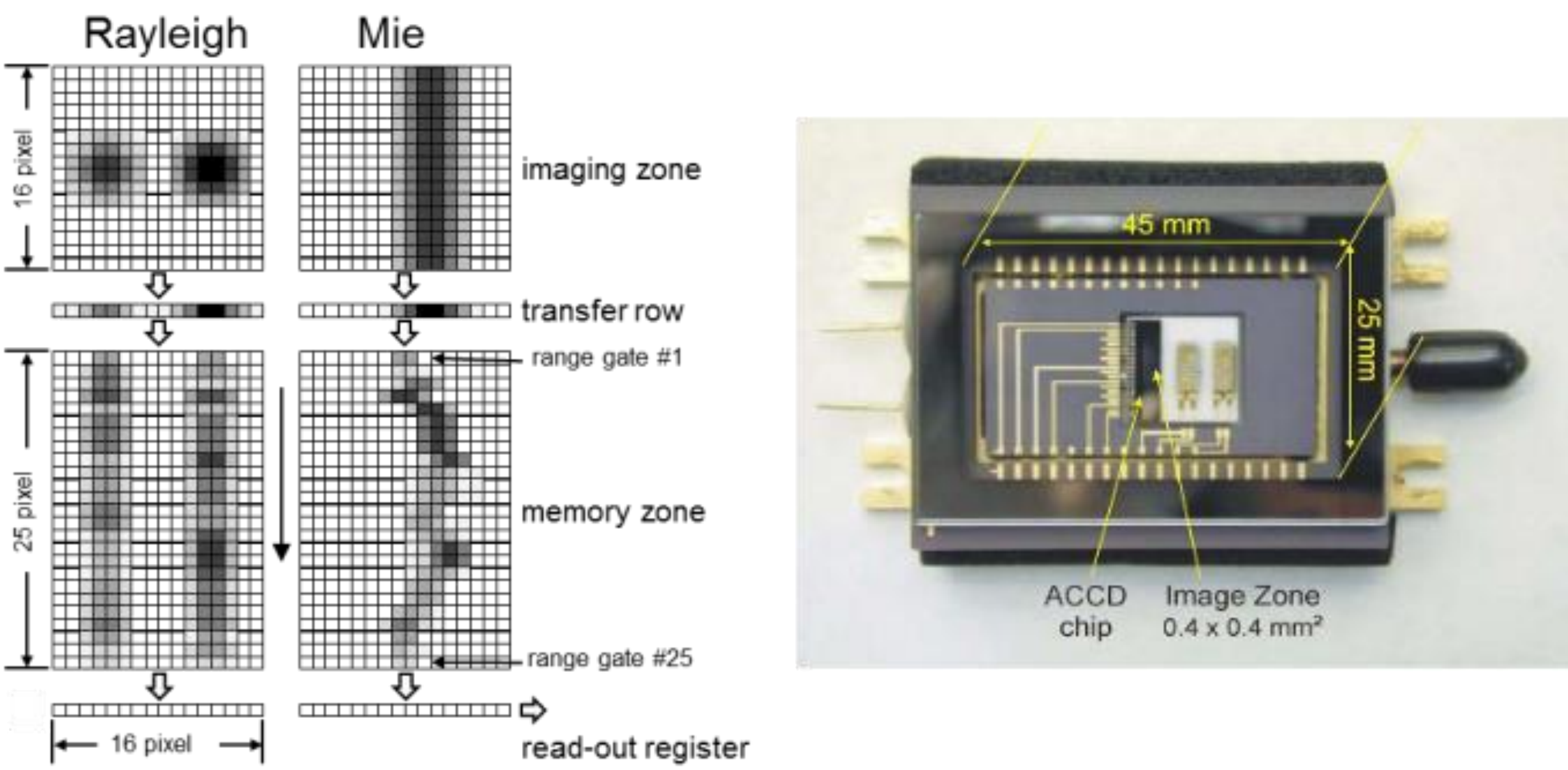

**Figure 1: (left): Illustration of the Aeolus ACCDs with imaging zone, transfer row and memory zone for the Rayleigh and Mie channel (adapted from Marksteiner, 2013). (right): The detector with the ACCD chip housed in a thermo-controlled hermetically sealed package (ESA, 2008).**

**Table 1: Specifications of the Aeolus ACCDs**

| Parameters | Value |
|---|---|
| Type | Thinned backside illuminated accumulation Si-CCD |
| Area | Imaging zone: 0.43 mm x 0.43 mm – 16 x 16 pixels<br>Memory zone: 0.43 mm x and 0.75 mm - 25 x 32 pixels |
| Pixel size | Imaging zone: 27 µm x 27 µm<br>Memory zone: 13.5 µm x 30 µm |
| Operating temperature | -30 C° |
| Temporal resolution | 2.1 µs – 16.8 µs / 250 m – 2000 m for atmospheric layers (#1 - #24)<br>3564 µs for solar background (layer #25) |
| Quantum efficiency | 0.85 |
| Charge transfer efficiency | 0.9999 |
| Radiometric gain | Mie: 0.68 LSB/e-, Rayleigh: 0.44 LSB/e- |

## 2.2 Dark current signals and anomalies

Even in the absence of light, a relatively small amount of thermally generated electrons is collected in the CCD. This is known as dark current and causes a non-negligible background signal on CCDs. In general, dark current signals play an important role for the random as well as systematic error budget of CCD based measurements.

On the one hand, the dark signal affects the random error budget by dark current signal noise. For a typical optical CCD instrument, the noise contributions are related to the signal itself, the noise of the dark current signal, and the read-out noise. The noise of Aeolus signals is dominated by the Poisson distributed shot noise as the levels for dark current and read-out noise are very low. Thus, the technique used for Aeolus is referred to as "quasi-photon" counting.

On the other hand, shifts in the mean dark current signal can potentially lead to systematic errors, which are far worse for the measurement principle of Aeolus than an increased dark signal noise. The mean dark current signal depends on the residence time of the signals in the CCDs and increases with increasing temperature (Janesick, 2001). Thus, the Aeolus ACCDs are operated at a temperature of -30 °C to minimize the dark signal level as far as possible. The variation of the dark current signals between different CCD pixels is called dark signal non-uniformity (DSNU). Dark current anomalies, as discussed in the following, can lead to sudden shifts of the mean dark current of single pixels and thus, significantly increase the DSNU. For Aeolus this can bias the quasi-photon counting lidar wind measurements.

CCDs have been widely used in the field of astronomical observations (de Bruijne, 2012; Massey et al., 2014) but have also found increasing application in Earth and planetary remote sensing from space (Burrows et al., 1999; Courrèges-Lacoste et al., 2017). However, CCDs have not been used for lidar applications from space. Since CCDs operated in space are exposed to harsh radiation conditions, radiation-induced effects are an important issue. In particular, the effects of high-energy particles such as cosmic electrons, ions, neutrons, and protons passing through CCDs have to be considered (Hopkinson et al., 1996). These particles are mainly of solar and interstellar origin and are often trapped and accumulate in the Van Allen radiation belt

(Feynman and Gabriel, 2000). The geographic region where the inner Van Allen belt comes closest to the Earth's Surface is called South Atlantic Anomaly (SAA). The SAA is a region of reduced magnetic intensity where satellites in Low Earth-Orbits (LEO) (< 1000 km altitude) are exposed to strong radiation (Anderson et al., 2018) and thus this region is of particular interest. Typically, the SAA is situated at an altitude of 200 km to 800 km over the Earth's surface (Nasuddin et al., 2018). A significant increase of dark signal levels in the region of the SAA has been observed on the CCDs of the Hubble Space Telescope which is operated at an altitude of 540 km (Kimble et al., 2000). But also effects on other detectors than CCDs were reported, such as for the photomultiplier tube on-board the CALIPSO satellite (Noel et al., 2014).

In general, radiation-induced effects can be categorized into three groups: ionization damage, displacement damage and transient effects. Ionization damage can lead to an increase of trapped charges in the oxide layer of the CCD and thus, may lead to an increased dark current and to a shift in the optimum operating voltages of the CCD. Displacement damage is caused by energetic particles (mainly protons) passing through the CCDs which may displace atoms from their lattice and create vacancy-interstitial pairs. Most of the pairs recombine but some of them may form stable displacement damages in the lattice. Displacement damage can lead to a degradation of the CTE and an increase of the dark current. So-called "hot pixels", pixels with enhanced dark current signals over a longer period of time, may evolve. In addition, displacement damage may also introduce burst noise, e.g. Random Telegraph Signals (RTS)-noise. RTS noise causes the dark current to change its state between two or more discrete levels at random and unpredictable times (Hopkins and Hopkinson, 1993; Smith et al., 2004). Hot pixels in combination with RTS phenomena were also observed for the CCD detectors of the Global Ozone Monitoring by Occultation of Stars (GOMOS) instrument on-board ENVISAT (Keckhut et al., 2010). In the framework of the Aeolus ACCDs development, proton tests have been performed to evaluate the probability of occurrence of such hot pixels and RTS pixels at an operating temperature of -30 °C. Transient radiation effects occur due to ionization-induced generation of charges within the CCDs and do not cause lasting damage. However, these effects might be visible as spurious signal spikes on one or more pixels and thus, also must be rejected in the quality control of the lidar signals analysis. Typically, optical sensors in Earth observation payloads are quite efficiently shielded from ionization damage. On the contrary, shielding from particles generating displacement damage which are typically high energetic protons is nearly impossible. This is why the CCD performance in space is quite often limited by displacement damage induced effects. A more detailed description of radiation effects is given in, e.g., Hopkinson et al. (1996) and Waltham (2010).

Besides radiation-induced effects increasing the conventional thermal dark current in the CCD, so-called clock induced charges (CIC) can cause an increase of the dark signal. CIC is a spurious signal generated by transferring measurement signals through the CCD and contributes to the dark signal. When clocking the charges through a register there is a small probability that additional charges are created which eventually manifest as additional dark signal (Bush et al., 2015). The level of CIC thereby mainly depends on the operation mode of the CCDs (inverted vs. non-inverted mode) and on the clock voltages and timing settings (e2V Technologies, 2015). In case the CCD is operated inverted, a negative baseline clock voltage is applied to the CCD which causes carriers to populate the free states at the silicon-oxide interface of the CCD. The level of CIC is higher for devices operated in inverted mode and is independent of the operating temperature of the CCD. Problems with CIC are

typically reported for Electron Multiplying Charged Coupled Devices (EMCCDs) due to the use of multiplication gain registers which amplify CIC (Wilkins et al., 2014). However, dark signal that arises from CIC may also not be neglected for the Aeolus ACCDs as the memory section is operated in inverted mode and special clocking is applied in the on-chip accumulation process. This clocking also accumulates CIC from a point defect, which would normally be distributed over a column in conventional CCD operation, in a single pixel and potentially increases the probability of CIC giving rise to the hot pixels which are observed. The distributed CIC observed in normal CCD operation has been shown to be increased by radiation (Bush et al., 2015) but there is little evidence for radiation-induced CIC generation in single pixels.

In the literature, hot pixels are often described as pixels with increased dark current over a longer period of time (Waltham, 2010). Thus, in the context of this paper a pixel is defined as "hot" if the pixel's dark signal time series shows a permanent increase of the dark current signal. In Sect. 3.1, an algorithm is introduced which is capable of detecting shifts in the median signal of time series. In case the algorithm detects a shift in the median, the pixel is classified as hot pixel. Pixels showing only transient events (see Sect. 3.2) are not classified as hot pixels in the following.

### 2.3 Dark current characterization measurements

Considering that the Aeolus ACCDs consists of 24 range gates to provide vertically resolved wind measurements from the lower stratosphere (e.g. 25 km) to the ground, a single hot pixel that contaminates the result of a range gate can have a detrimental impact on the relative quality of the final product. The low number of 16 CCD pixels per range gate and signal acquisition (see Fig. 1, left) limits the possibility to omit affected pixels which has been used, e.g., for the Ozone Monitoring Instrument (OMI) on-board EOS (Schenkeveld et al., 2017) and the Global Ozone Monitoring by Occultation of Stars (GOMOS) on-board ENVISAT (Bertaux et al., 2010). This is why, the dark current signals of Aeolus have to be acquired on a regular basis and corrected accordingly in the wind retrieval.

The in-orbit measurement procedure of the Aeolus detection chain considered the dark current signals characterization in imaging and memory zone with two specific measurement procedures. Initially, it was planned to use these procedures once at the beginning of the in-orbit phase for a one-time characterization of the DSNU of the ACCDs. These procedures were defined to be executed before the laser is switched to UV emission and would have required a transition of the laser to a lower mode to perform a rerun.

After the first identification of hot pixels in the nominal Aeolus wind lidar measurements, a new procedure to allow dark signal characterization of the memory zone during continuous laser operation was introduced, so-called DUDE (Down Under Dark Experiment)-measurements. During DUDE measurements the range gate timing settings are adjusted such that the theoretical return signal is acquired from below the Earth's surface. Figure 2 illustrates the difference in the data acquisition between wind (a) and DUDE (b) mode. In that way, dark current signals of all pixels of the memory zone can be measured without lidar signal contributions apart from the solar background signal.

In this paper, DUDE measurements obtained from the quasi raw Aeolus L1A data products were analyzed (Reitebuch et al., 2018). The specific L1A data product is generated after each DUDE characterization and contains geo-located but unprocessed dark current signals of both channels for 25 range gates and 16 pixels at the measurement level, i.e., in the same format as nominal wind lidar measurements which allows for a DSNU characterization. In a first step, the DCO was subtracted from each pixel value at measurement level. Next, the measurements were averaged to observations by calculating the mean over the measurements per observations.

Afterwards, quality checks were applied at the observation level. First, the dark current observations were filtered according to the height of their top range gate with respect to the Earth's surface. At the beginning of the newly introduced DUDE measurements, the detection range was not always below the Earth's surface and atmospheric signal could contaminate the characterization of the first range gates. Single observations of range gates were rejected when the top was above the Earth surface given by the Digital Elevation Model (DEM) (Reitebuch et al., 2018). Second, the signals were filtered for solar background contamination and accepted when the signal sum in range gate #25 was smaller than 5.0 LSB. Although optimizing the location of the DUDE measurements within the orbit, some dark current observations were still exposed to high solar background levels.

For the following statistical analysis, the DUDE observations were concatenated to a time series. It has to be noted that the observation frequency has not been constant. This needs to be considered when interpreting the results. Figure 3 shows the relationship between the DUDE observation number, i.e. the index of the concatenated data stream, and the observation time. Apart from that, the figure shows the development of the observation frequency for DUDE measurements (colored areas). At the beginning of the mission until the 26$^{th}$ of November 2018, DUDE measurements were only carried out intermittently (see Period A in Fig. 3). As more and more dark current anomalies became obvious, it was decided to perform DUDE measurements on a daily basis from then on. Period C has a high number of characterization measurements as the laser was in lower mode due an instrument anomaly (Lux et al., 2020). Afterwards, the frequency of DUDE measurements was increased to four per day (period D). The green dots in the plot indicate valid dark current observations which passed the two filtering steps described above. Note that during certain periods in March and October enhanced solar background values are measured along the whole orbit. This is why several observations were sorted out during these periods. During the switch-over period from the primary to the secondary ALADIN laser in June 2019 long-term DUDE measurements over several orbits and thus, also in the solar background maxima, were performed. As a result, these observations were sorted out as the solar background criterion was not met. In total, 39043 out of 72850 observations obtained from 2065 DUDE L1A files between 2018-09-02 until 2020-05-20 passed the quality checks and were analyzed in the framework of this paper.

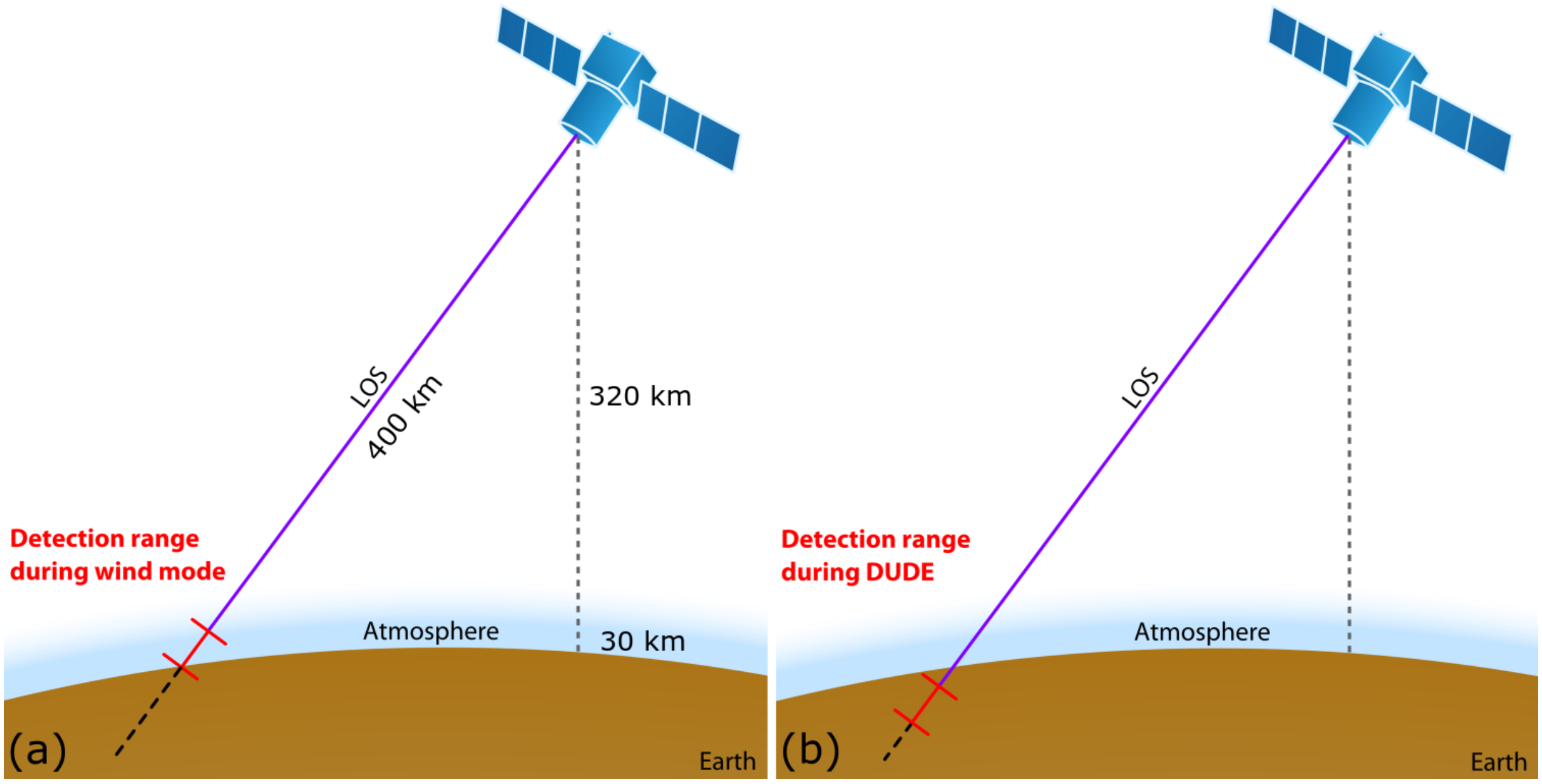

**Figure 2: Aeolus detection range during nominal wind measurement (a) and DUDE mode (b).**

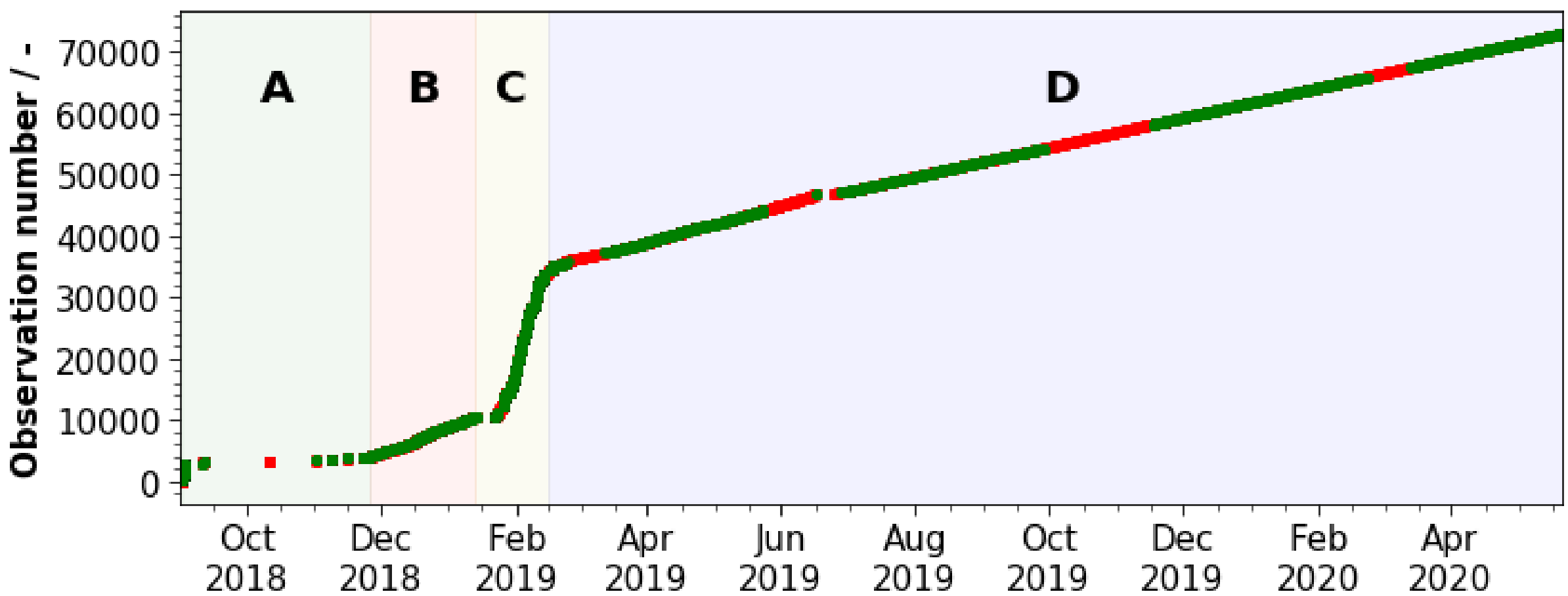

**Figure 3: Relationship between dark signal observation number (y-axis) and observation time (x-axis). Green points indicate valid observations which passed the filtering steps. The differently coloured areas mark periods of different sampling frequencies of DUDE observations: A: 2018-09-02–2018-11-26, B: 2018-11-26–2019-01-13, C: 2019-01-13–2019-02-15, D: 2019-02-15–2020-05-20.**

## 3 Dark signal analysis

This section describes methods used to detect and characterize dark current anomalies. The quality of Aeolus wind measurements and also the performance of the DUDE correction strongly depends on the characteristics of the dark current anomaly. Apart from that, also the frequency of DUDE characterizations is determined by the hot pixel characteristics. This is
why, an exact knowledge of the dark current characteristics is necessary. In the following, it is differentiated between transient and permanent dark current anomalies, also referred to as hot pixels. Here, the focus is on permanent dark current anomalies as they potentially have large impact on the data quality and the performance of the DUDE correction. Pixels that show permanent dark current anomalies are further divided into pixels exhibiting RTS-like features, which are particularly detrimental for the DUDE correction, and pixels that show only sporadic shifts of the dark current signal.
Firstly, an algorithm is introduced which is capable of detecting permanent dark current anomalies by screening the dark signal time series for shifts in the median dark current signal. Secondly, the algorithm used to detect transient events is described. Finally, the impact of hot pixel induced systematic errors of Aeolus wind observations is discussed in detail.

### 3.1 Detection of permanent dark current anomalies

The motivation for a detailed characterization of permanent dark current anomalies is twofold: a) it supports investigations for the underlying root causes of the hot pixel issue and b) the number and magnitude of dark signal shifts define the impact on the wind observations and the DUDE correction.
The blue dots in the top plots of Fig. 4 and Fig. 5 show two dark signal time series at observation level for Mie pixels [13, 6] and [13, 9]. The former time series depicts a pixel with nominal dark signal behavior whereas the latter sets an example of a hot pixel which exhibits RTS-like characteristics with multiple shifts in the mean dark signal. This can also be seen from the histograms of the dark signal intensities in the bottom left plots of Fig.4 and Fig. 5. For Mie pixel [13, 6] the dark signals are Gaussian distributed with a mean value of 0.27 LSB and a scaled median absolute deviation (MAD) of 0.69 LSB. Like the standard deviation, the MAD is a measure of the spread of a distribution which is more robust to outliers. In case of a normal distribution, the MAD multiplied by a value of 1.4826 (scaled MAD) is identical to the standard deviation. In contrast, the histogram of Mie pixel [13, 9] clearly indicates RTS characteristics with two dominant levels at ~8.0 LSB and ~15.0 LSB besides the base level in Fig. 5.
In order to scan for anomalies in the dark signal time series, the Python module *ruptures* is used (Truong et al., 2020). The problem of finding sudden shifts in the median dark current signal can be described as choosing the best possible segmentation of signal $y$ into $K$ segments according to a definable cost-function $c$ that must be minimized. The cost-function measures the goodness-of-fit of a sub-signal $y_{t_k \dots t_{k+1}}$ to a specific model:

$$\min_{\tau} \sum_{k=0}^{K} c(y_{t_k \dots t_{k+1}}) + pen(\tau) \tag{1}$$

where $\tau = \{t_1 \cdots t_K\}$ denotes the best possible segmentation and the penalty term $pen(\tau)$ defines the complexity of the segmentation. The last term is necessary as the number of change points is not known beforehand. To tackle the optimization problem of Eq. (1), the module provides many different models, cost functions and optimization methods.

The selection of the cost function determines the type of shifts that shall be detected. In this study, a robust detection of sudden shifts in the median of the distribution is needed. Thus, a cost function which detects sudden changes in the median of the signal is selected:

$$c(y_{a..b}) = \sum_{t=a+1}^{b} \left| y_t - y_{med,(a,b)} \right| \tag{2}$$

where $y_{med,(a,b)}$ denotes the median of the sub-signal $y_{a..b}$. As optimization method, a bottom-up approach is used which starts from the finest possible approximation of the time series and iteratively deletes less significant segments until a stopping criterion is met (Keogh et al., 2001). As the number of shifts is unknown, the sensitivity of the algorithm needs to be controlled with the aid of a penalty term (see Eq. (1). Here, a linear penalty term of the following form is used:

$$pen(\tau) = \beta|\tau| \tag{3}$$

where $\beta$ is the smoothing parameter to control the number of shifts. Too low values of $\beta$ favor the detection of many shifts, even those that are a result of noise. In contrast, a too large penalty might even detect no shifts at all. Here, for all pixels a smoothing parameter of 23.0 LSB is used. This value was thoroughly tuned and selected based on visual inspection of the entire dataset.

The results of the algorithm applied to the dark signal time series of Mie pixels [13, 6] and [13, 9] is depicted in Fig. 4 and Fig. 5. The detected signal segments are indicated as black lines in the top plots of both figures. For Mie pixel [13, 6], no shifts were detected which suggests nominal dark signal behavior. In contrast to that, multiple shifts were detected for Mie pixel [13, 9]. After a nominal phase until observation number 1160, the dark signal level suddenly changes to a higher level and shows step-like transitions from then on. After being able to properly detect shifts in the time series, the next task is the description of the RTS characteristics. After being able to properly detect shifts in the time series, the next task is the further subdivision of permanent dark current signal anomalies into RTS-like and sporadic anomalies.

In general, RTS can be described by the following parameters: mean time spent on a discrete level, signal amplitude of each level, and the number of levels (Goiffon et al., 2009). In principle, the number and amplitudes of the RTS levels could be retrieved by analyzing the histograms of the dark signals at observation level (see Fig. 5, bottom left). As already outlined above, this plot suggests that besides the base level, two dominant RTS levels at ~8.0 LSB and ~15.0 LSB exist. Nevertheless, there might be additional levels hidden in the noise. Thus, clever filtering is needed. The information about the location of the detected shifts is used to apply segment-wise median filtering using a window size of 20 observations to the signals. This allows for a better detection of the RTS levels. The resulting histogram of the segment-wise median filtered signal is shown in Fig. 5 (bottom right). Finally, Gaussian Kernel Density Estimation (KDE), available in the *statsmodel* library of Python, is applied to the median filtered signal to detect the RTS levels. The smoothness of the KDE is determined by the bandwidth

parameter. Insufficient smoothing results in a density estimate which is too rough and thus, contains spurious data artifacts. On the contrary, important features may be smoothed away when applying excessive smoothing. Here, maximum-likelihood cross-validation is used to determine the bandwidth parameter which is an established method for the objective, data-based derivation of the bandwidth parameter (Jones et al., 1996).

The number of modes of the resulting KDE defines the number of RTS levels. Mode estimation is done using Python's *SciPy* peak finding algorithm (Virtanen et al., 2020) which is capable of finding local extrema by comparing neighboring values of signal series. A minimum horizontal distance criterion of 0.2 LSB was selected as condition for the peak finding algorithm. As a consequence, minimum RTS amplitudes just above the noise level which is typically ~ 0.15 LSB for median filtered signals (see text box of Fig. 4, bottom right) are detectable. In case of Mie pixel [13, 9] this analysis reveals multi-level RTS with modes at 6.54 LSB, 8.31 LSB, 9.52 LSB, 11.14 LSB, 13.59 LSB, 14.95 LSB, and 16.09 LSB besides the baseline level at 0.20 LSB (see Fig. 5, bottom right).

Due to the temporal incoherence of DUDE measurements (as discussed in Sect. 2.3), it is difficult to properly assess temporal RTS characteristics. However, the frequency of the switching between the different RTS-levels is assessed by simply counting the number of detected steps in adjacent intervals of 500 observations starting from the index of the first detected segment. This yields a count of steps for each 500-observation long interval. The average count over all the intervals is used to assess the fluctuation frequency. For instance, for Mie pixel [13, 9] 1.39 steps per 500 observation - interval on average are counted.

In contrast to RTS-like anomalies, sporadic dark current anomalies do not show shifts between different dark current signal levels at a very high rate. Herein, a pixel is defined as RTS pixel if there are at least four consecutive shifts between two or more levels. Other pixels that were classified as hot pixels fall into the category of sporadic median shift pixels. Figure 6 illustrates a case where the dark signal shows an increase from 0.17 LSB to 0.56 LSB followed by a return to the base level. This pixel will not be defined as RTS pixel in the following.

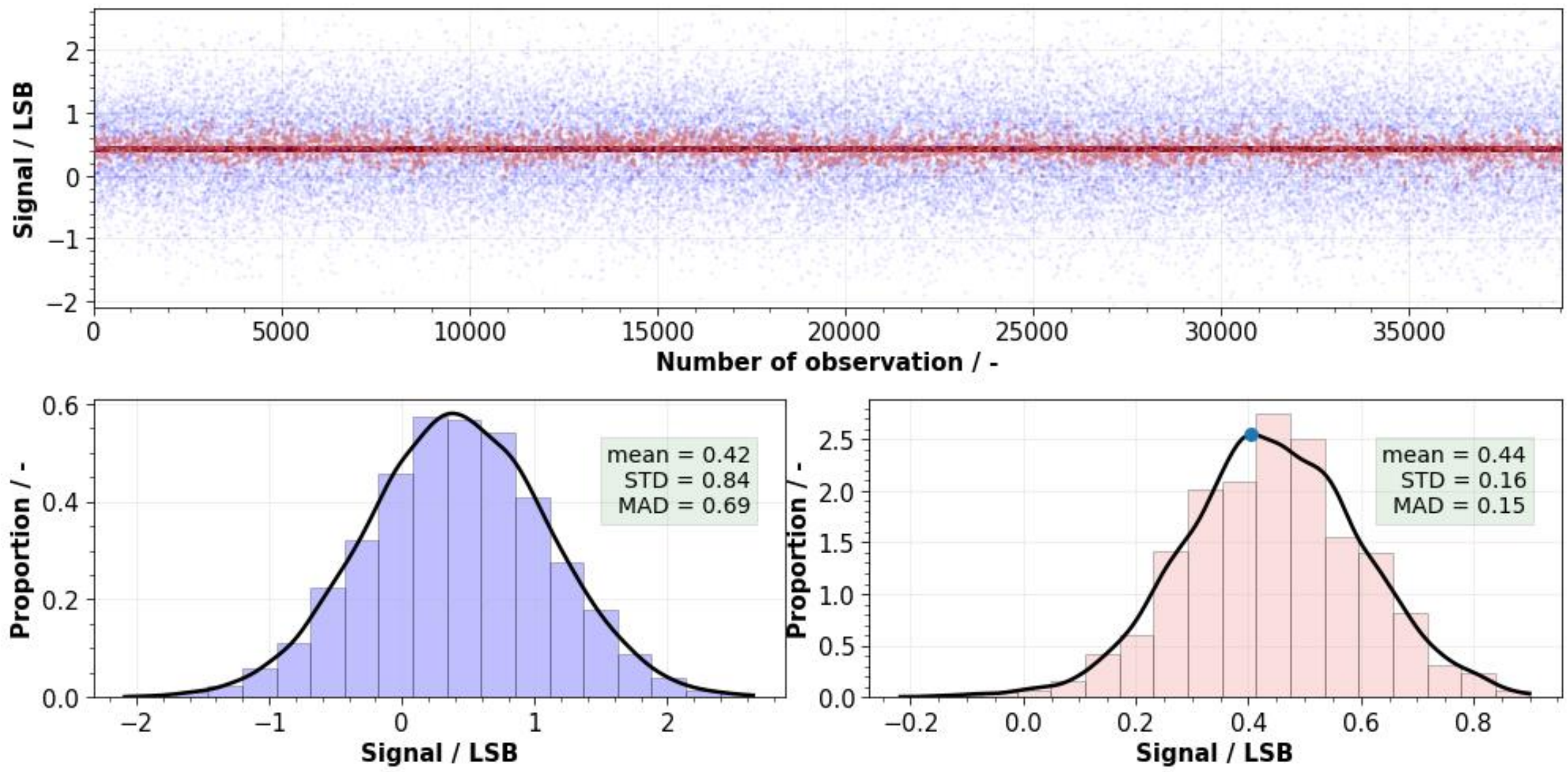

**Figure 4: Dark signals of Mie pixel [13, 6] – nominal dark signal behaviour. (Top): The blue dots indicate dark signal intensities at observation level. The solid black line(s) indicate the median value of each detected segment. The red dots show dark signal intensities after median filtering (window size of 20 observations) applied to each detected segment. (Bottom): Histograms of dark signal intensities at observation level (left) and after median filtering (right). The black curve indicates Gaussian density kernel estimation and the blue dot(s) show the mode(s) of the distribution. The text box contains values of sample mean, standard deviation (STD), and scaled median absolute deviation (MAD) in units of LSB.**

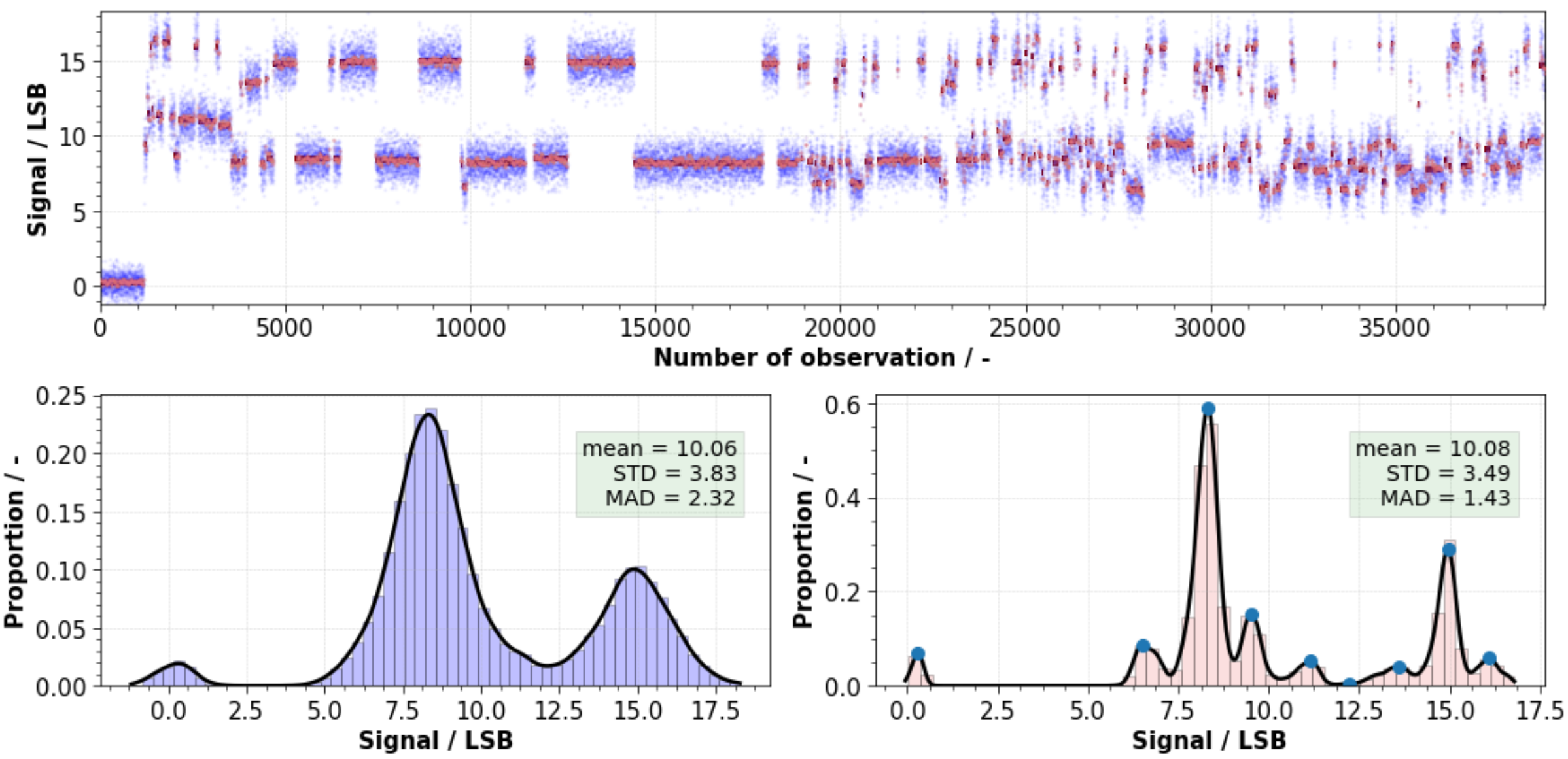

**Figure 5: Same as in Fig. 4 but for Mie pixel [13, 9] – RTS-like hot pixel behaviour.**

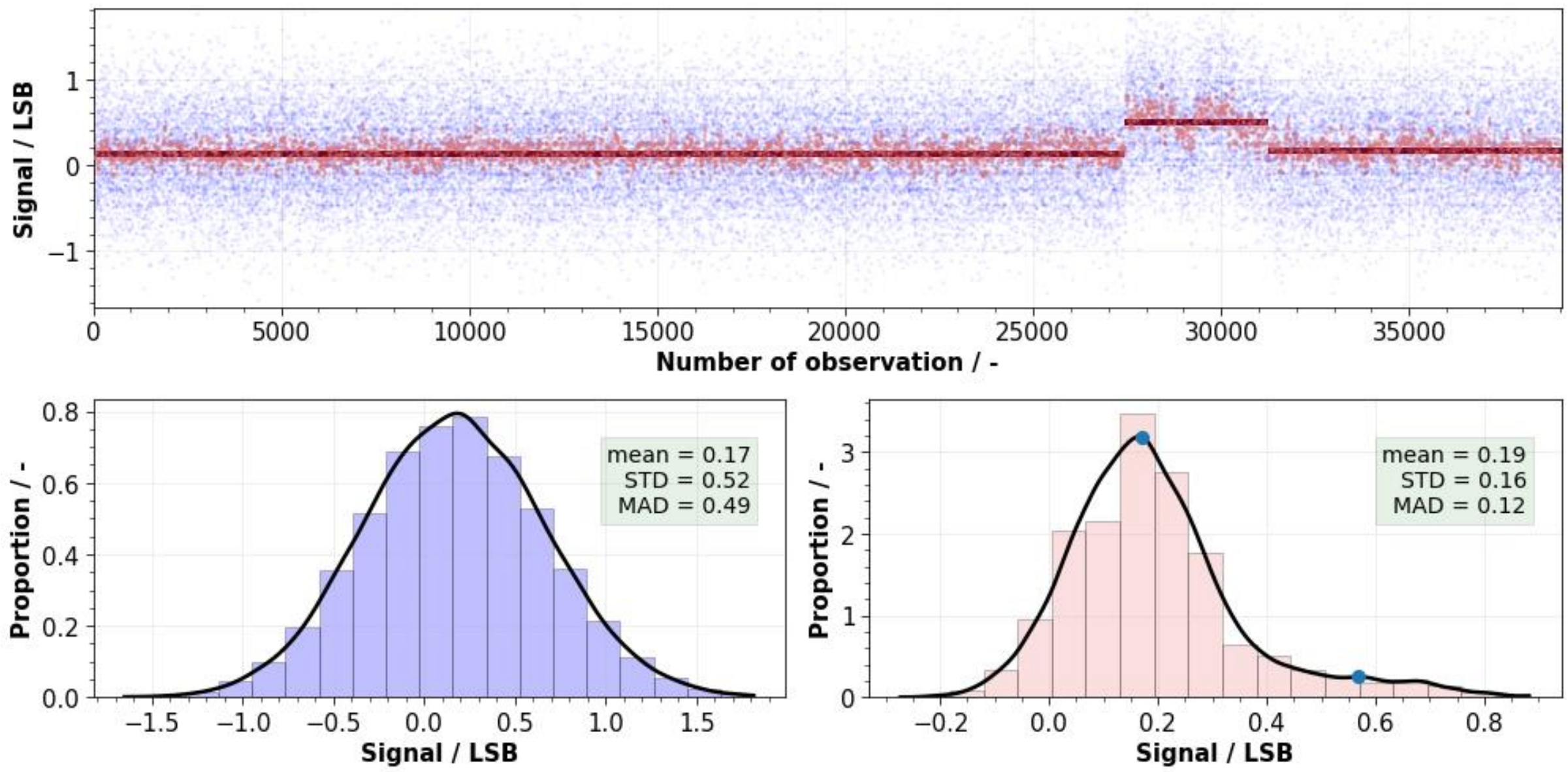

**Figure 6: Same as in Fig. 4 but for Rayleigh pixel [7, 8] – hot pixel with sporadic shifts.**

### 3.2 Detection of transient dark current anomalies

As outlined in Sect. 2.2 transient effects which cause spurious dark signal spikes may occur on the CCD. In contrast to the detection of permanent dark current anomalies (see Sect. 3.1), the detection of spurious spikes is performed at measurement level. Typically, transient events appear as isolated signal peaks only present in one measurement and show a wide range of amplitudes. Figure 7 shows dark signals (blue dots) of hot pixel [13, 9] of the Mie channel at measurement level for a selected section. This plot demonstrates that a simple threshold approach is not desirable as signal spikes should also be detected in the presence of increased and changing baseline dark current levels. To achieve this, Python's *SciPy* peak find algorithm provides suitable methods. In principle, the peak finding algorithm works by comparing neighboring signal values. Different selection criteria based on the peak properties can be specified to find the desired peaks. Here, the prominence of a peak is selected as the selection criterium. The prominence describes the amplitude between a detected peak and its lowest contour line and thus, measures how much the peak stands out from its surrounding baseline. As a consequence, this detection approach is independent of the underlying signal level of the baseline and also allows spike detection in the presence of dark current anomalies such as RTS phenomena. For this analysis, a minimum prominence value of 45.0 LSB is used for all pixels. This value was carefully tuned based on visual inspection of the results of all pixels. The red dots of Fig. 7 show transient events detected by the algorithm. For this pixel, a total number of 36 transient events out of 180310 analyzed measurements were detected. The amplitudes are in the range between 35.5 LSB and 7142.5 LSB.

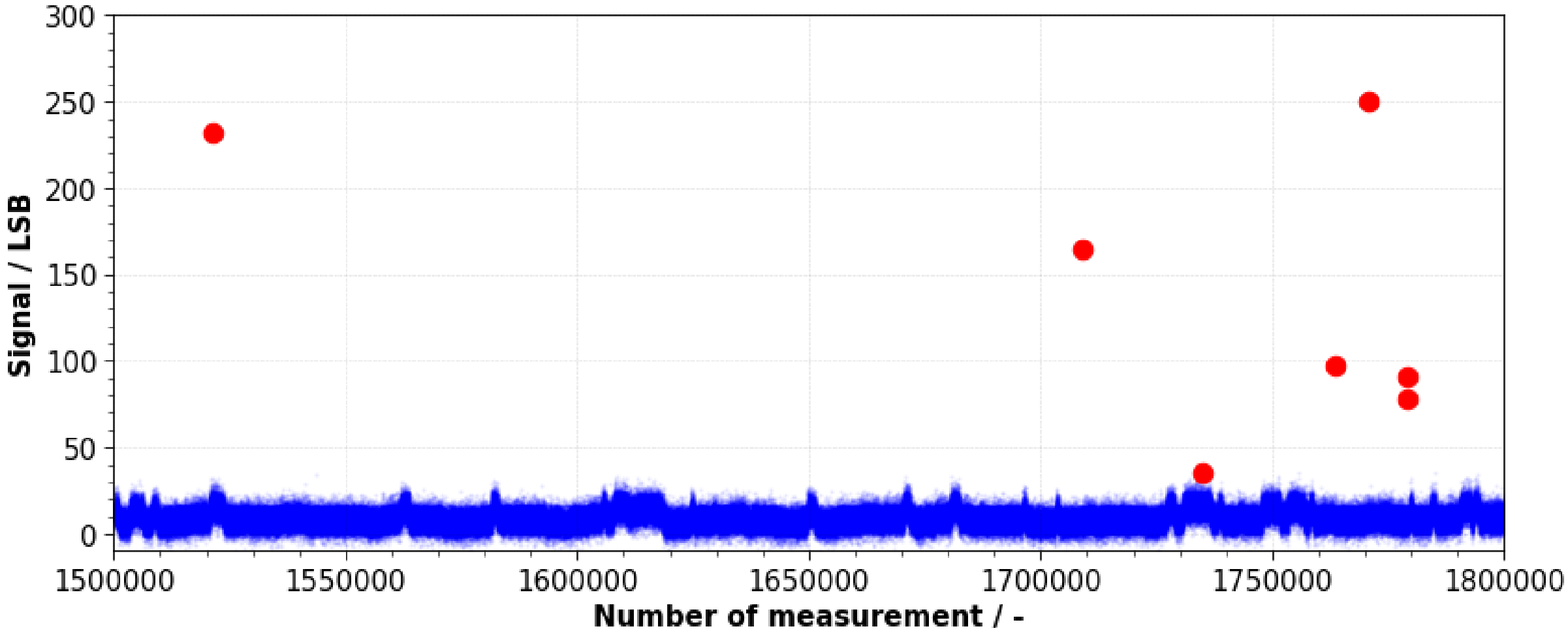

**Figure 7: Dark signal values (blue dots) of hot pixel Mie [13, 9] at measurement level for a selected section. The red dots indicate detected transient events.**

### 3.3 Impact of dark current anomalies on wind measurements and correction

Already shortly after launch it became obvious that already slightly increased dark current values of single pixels can lead to systematic errors in wind measurements in the affected range gate. Comparisons between the forecast model of the European Centre for Medium Range Weather Forecast (ECMWF) and Aeolus winds revealed suspicious horizontal features in the difference between observations and the model background. Figure 8 shows the deviation between Aeolus Level-2B Rayleigh-clear HLOS (horizontal line-of-sight) winds and the ECMWF model equivalent for October 2018. For the pressure level around 400 hPa (covered by range gate #11) an obvious offset to the other levels can be observed. This offset could be traced back to pixel [11, 2] of the Rayleigh ACCD. As shown later, the dark current of this pixel shows multiple step-like shifts between different dark current levels and thus, is classified as RTS pixel. These shifts are also imprinted on the wind results and cause the fluctuations of the dark current induced wind bias.

Due to the different retrieval algorithms used for the Mie and Rayleigh channel, the impact of dark current anomalies is slightly different for both channels. The determination of Mie winds is based on finding the centroid position of the fringe imaged onto the CCD (see Fig. 1, Reitebuch et al., 2018). A hot pixel causes a fake peak beside or on top of the imaged fringe. Depending on the signal strength of the atmospheric backscatter signal and the position of the hot pixel relative to the fringe, the hot pixel induced bias can be very different.

The wind retrieval of the Rayleigh channel is based on the measurement of the contrast $R = \frac{I_A - I_B}{I_A + I_B}$ between the signals $I_A$ and $I_B$ transmitted through two FPIs and detected within the left and right Rayleigh spots $A$ and $B$ (see Fig. 1, Reitebuch et al., 2018). A hot pixel leads to an enhancement of the signals of one of the Rayleigh spots depending on the location of the hot

pixels. This, in turn, leads to positively or negatively biased wind results. The following idealized example illustrates the magnitude of hot pixel induced effects for the Rayleigh channel. For example, assume same signal intensities through both Rayleigh filters $I_A = I_B = 1000\ LSB$ and a hot pixel induced signal elevation of 10 LSB present for Rayleigh spot $A$. The resulting contrast $R = \frac{10\ LSB}{2000\ LSB} = 5 \cdot 10^{-3}$ can be transferred to HLOS values in $m/s$ using a sensitivity of $6 \cdot 10^{-4}\ MHz^{-1}$ and a Doppler shift conversion factor of $5.63\ MHz\ m^{-1} s$ which yields a hot pixel induced error of already about 2.6 m/s HLOS.

Same as for the Mie channel the magnitude of the dark current signal-induced Rayleigh bias depends on the signal level of the backscatter signal. Generally speaking, the dark current induced bias is more constant in the Rayleigh than in the Mie channel due to the more constant Rayleigh signal compared to the strongly varying Mie signal from clouds and aerosols.

Already shortly after the hot pixel issue was recognized, an on-the-fly correction scheme was successfully implemented into the wind retrieval of the Aeolus operational processing chain on the 14th June 2019. For the correction, the dark signal characterization obtained from frequently performed DUDE measurements is used for a pixel-wise dark signal correction, i.e. a DSNU correction, of adjacent wind measurement signals. As this kind of correction was not foreseen before launch to be performed on a regular basis, dedicated instrument modes and algorithms had to be developed after launch. Figure 9 shows the effects of the hot pixel correction on the L2B Rayleigh-clear HLOS wind speeds (Rennie et al., 2020). Before the hot pixel correction (left to the vertical black line) the wind measurements show systematic biases which manifest as horizontal stripes at certain altitudes (at about 3, 11, and 20 km) which almost disappear after the activation of the hot pixel correction (right to the vertical black line). The measurement gap between 13:01 UTC and 13:05 UTC was related to a DUDE measurement which was used for the dark signal correction of the wind measurement signals of the adjacent orbit. It has to be mentioned that the DUDE correction can only correct for the DSNU. Potential hot pixel induced changes of the CTE (see Sect. 2.2) which may lead to different responses of the pixels to the incoming light, the so-called photo response non-uniformity (PRNU), cannot be corrected with the DUDE correction. This effect might explain slight remaining bias in hot pixel affected range gates also after the DUDE correction (see Figure 9).

Due to the frequent shifts of the dark signal level of RTS-like hot pixels, frequent dark signal characterization is necessary. This is why DUDE measurements with a duration of four minutes are performed four times per day. Thereby, the duration as well as the location of these measurements was optimized to influence wind measurements as little as possible but capturing hot pixel behavior sufficiently. Figure 10 shows the hot pixel correction approach in more detail on the basis of hot pixel [13, 9] of the Mie ACCD (see Fig. 5). The dashed red line shows the signal intensity not corrected for dark signals obtained from wind measurements of this pixel during 14th November 2019. To better distinguish between atmospheric and dark current signal the signal intensity was filtered using a median filter with a window length of 100 observations. The vertical green bars indicate the location of the four DUDE measurements which were used for the dark signal correction. The obtained dark signal correction value for this pixel is indicated by the blue line and gets updated every time after a new DUDE measurement was carried out. The dark current corrected signal intensity (dashed red line minus blue line) is shown by the solid red curve.

Comparing the two red lines already demonstrates that the correction scheme is capable of correcting for the overall elevated dark current signal level. However, there remains a problem for the near-real-time (NRT) processing when dark current transitions occur between two DUDE measurements. The uncorrected signal intensity (dashed red line) shows a dark current induced signal decrease of about 8.0 LSB at 14:15 UTC. Here, the dark current calibration based on the DUDE measurement from 13:15 UTC is still active. Thus, the dark current signal is overestimated and the dark signal corrected signal intensity (solid red line) shows the signal dip. This holds true until the new DUDE measurement is performed and gets used for the dark current calibration of the orbit which starts around 20:30 UTC. Afterwards, the dark signal corrected signal intensity is again at the same level as before.

This example demonstrates that even with DUDE measurements performed at high frequency, a perfect dark signal correction is not possible. It is also clear that the performance of the correction scheme depends on the behavior of the hot pixel. In case of RTS-like characteristics as shown for Mie pixel [13, 9] (see Fig. 5) the correction performs poor compared to hot pixels with sporadic shifts. Nevertheless, this approach works fine to remove the constant proportion of the dark current offset and in any case reduces periods of enhanced dark current induced bias sufficiently. In order to further mitigate hot pixel induced effects, a further check will be implemented for the Aeolus level 2B product in the future. This check is based on comparing the Aeolus with ECMWF model winds for each range gate. In case, the difference between both exceeds a certain threshold, the Aeolus winds of the affected range gate will be flagged invalid. For example, the period between 14:15 UTC and 20:30 UTC (see Fig. 10) would be flagged as invalid by the check.

However, the limitation of the discontinuous dark signal characterization in the NRT processing can be overcome in the reprocessing of Aeolus data. With the complete time series at hand the great advantage of reprocessing is the chance to use not only past DUDE measurements but also future measurements. For the reprocessing, there is the possibility to detect hot pixel induced steps in the wind measurement signals which remain after the nominal dark signal correction such as indicated in Fig. 10. This allows the introduction of additional dark signal characterizations in between the nominal ones to further mitigate hot pixel induced effects. The first reprocessed Aeolus dataset which covers the period from June to December 2019 was released in October, 2020. This dataset will include the improved dark signal correction as described above.

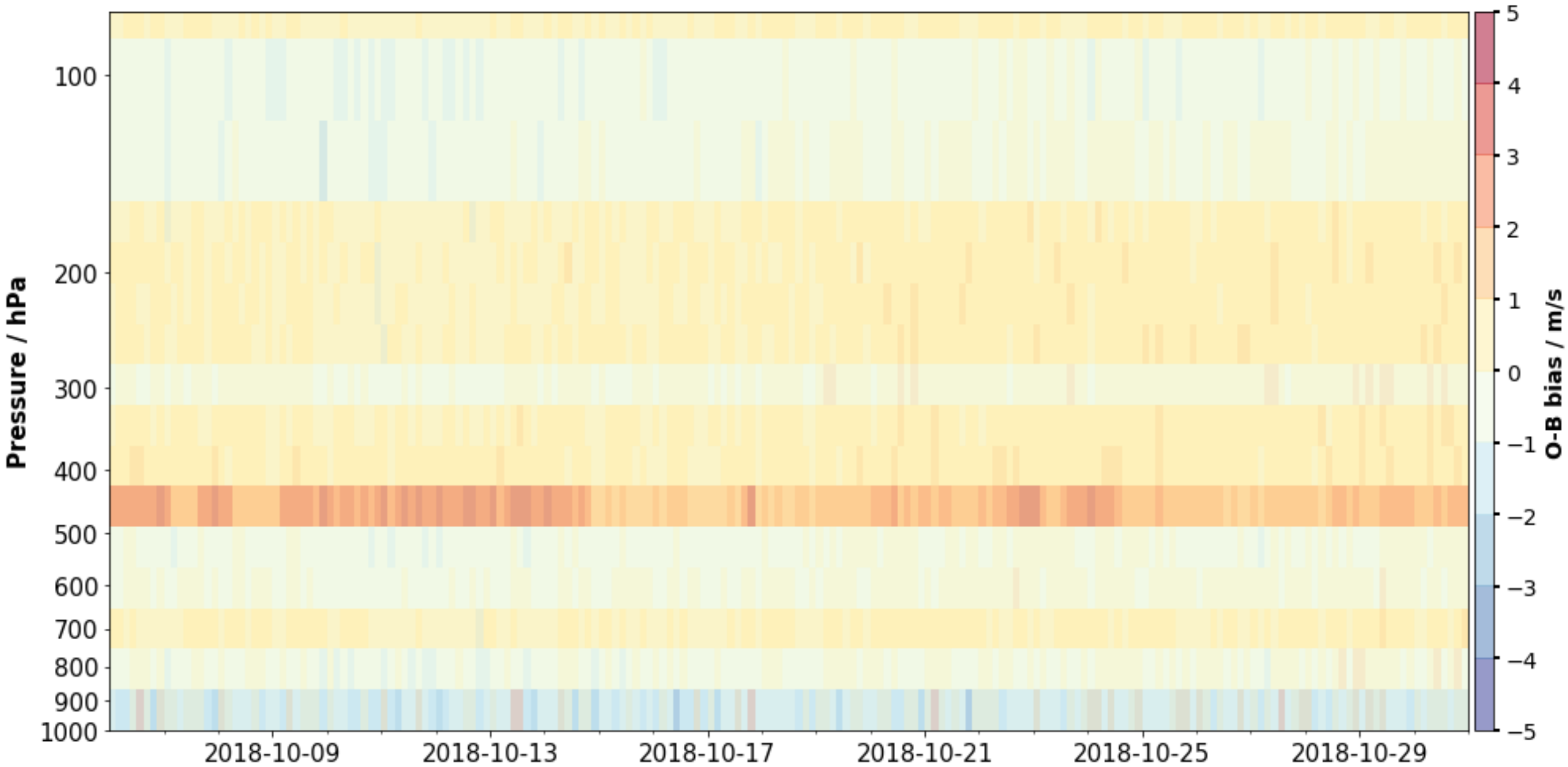

**Figure 8: Comparison between Aeolus L2B Rayleigh-clear HLOS winds and the ECMWF model equivalents between 2018-10-06 and 2019-10-31. The plot shows the mean difference between the observation (O) and the background (B) (short-range forecast) model field as a function of pressure and time.**

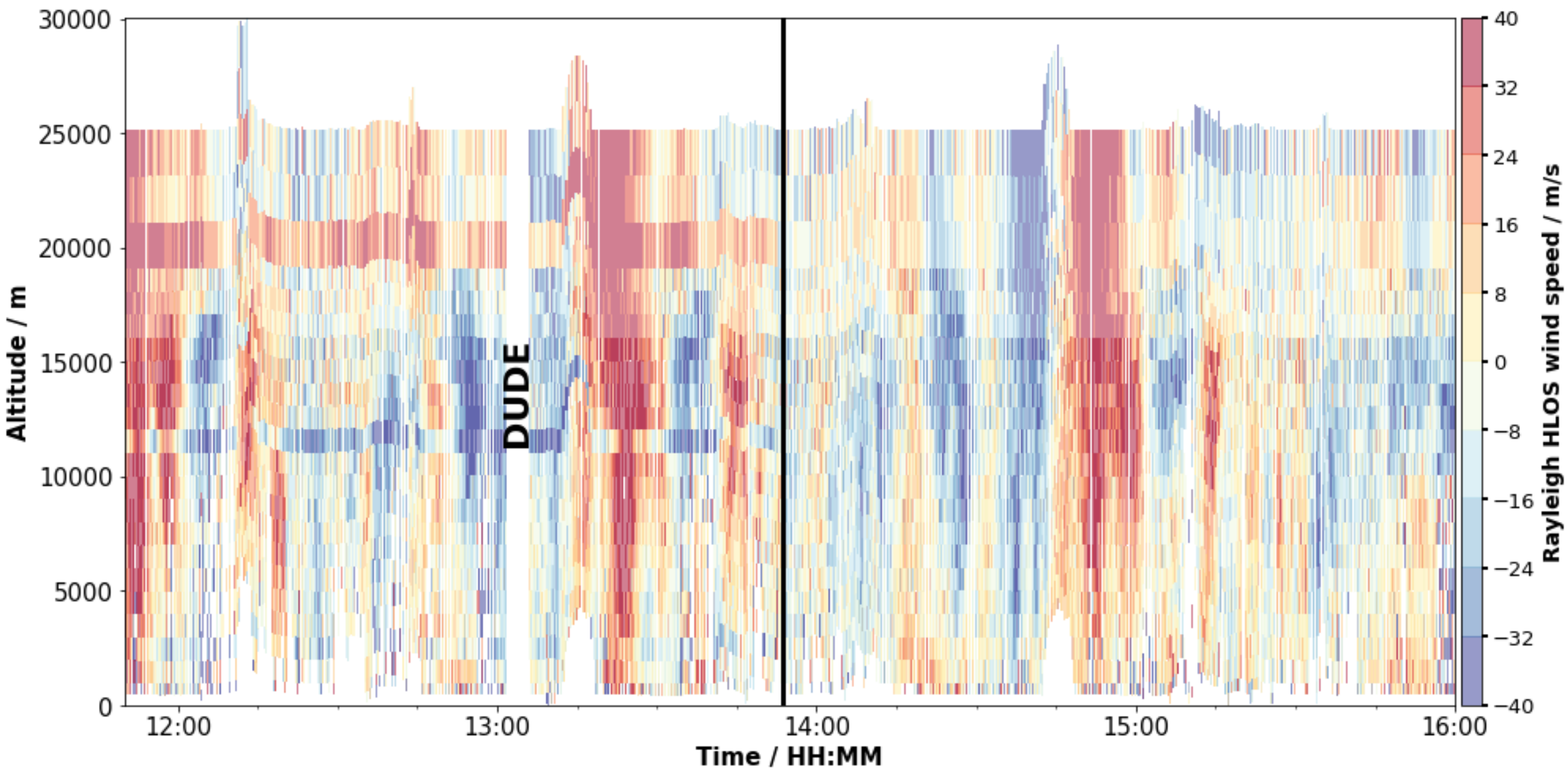

**Figure 9: Introduction of the hot pixel correction for the operational Aeolus processing on 2019-06-14. The plot shows L2B Aeolus Rayleigh-clear HLOS winds before (left to black line) and after the implementation of the correction scheme (right to black line). Between 13:01 UTC and 13:05 UTC a DUDE measurement was carried out which was used for the hot pixel correction of the subsequent orbit starting at 13:52 UTC.**

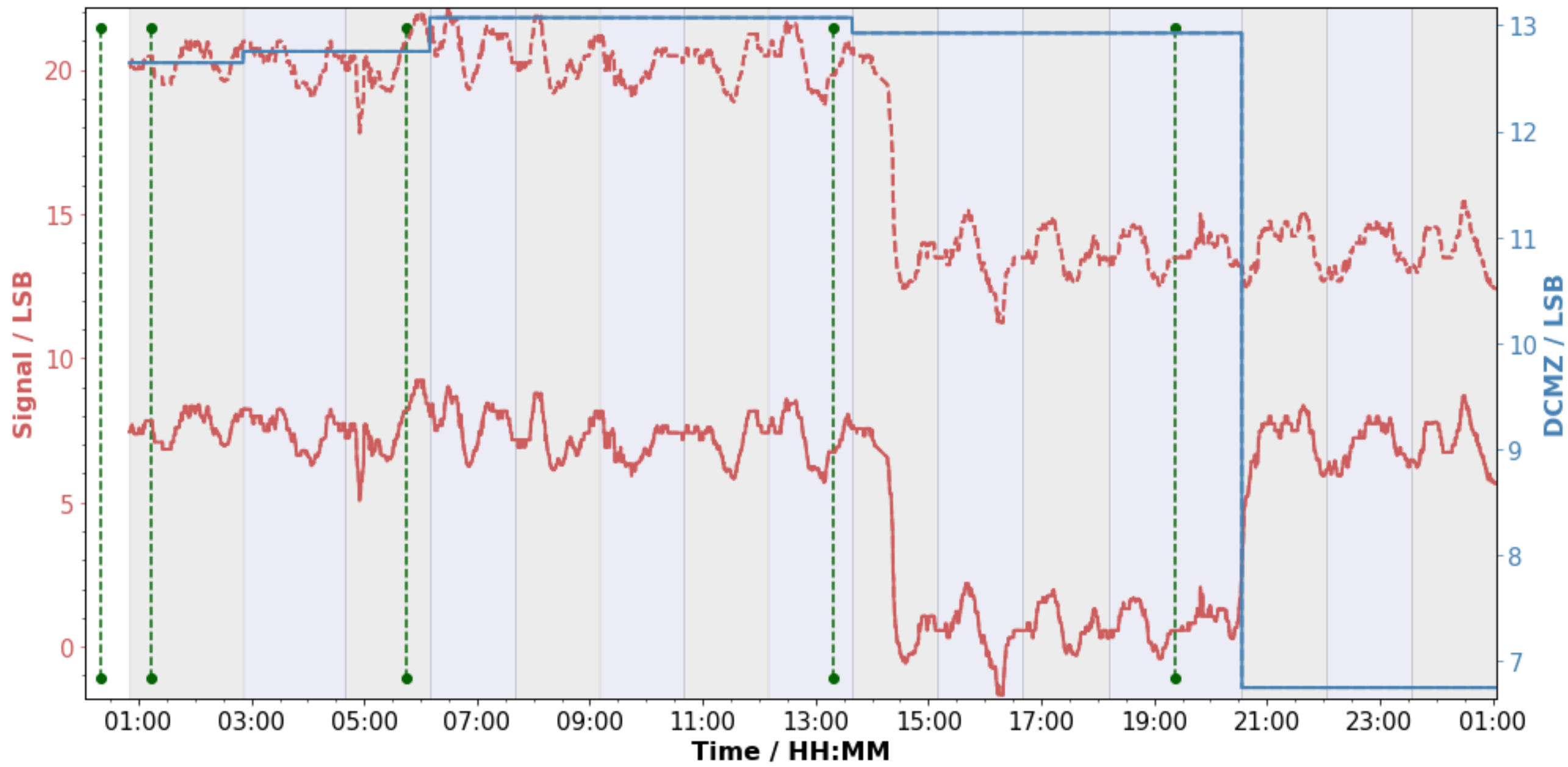

**Figure 10: Dark signal correction of Mie pixel [13, 9] on 2019-11-14. The solid and dashed red lines indicate median filtered (window length of 100 observations) signal intensity obtained during wind measurement mode corrected and uncorrected for dark current. The blue line with the second y-axis shows the corresponding dark signal correction value obtained from the four DUDE measurements – the location of the DUDEs is marked by the vertical dashed green lines. Additionally, the shaded areas indicate the duration of the measurement orbits (~ 90 minutes).**

## 4 Results

This section presents the statistical analysis of the outcome of the signal segmentation and transient event detection. The signal segmentation allowed the division of hot pixels into pixels which exhibit RTS like characteristics and into such pixels that show sporadic shifts without the presence of RTS features. Next, characteristics of the RTS features and properties of the dark signal shifts are characterized precisely. Finally, the occurrence of transient events is evaluated by statistical means.

### 4.1 Hot pixel location

As discussed in Sect. 2.2 such pixels where at least one shift in the median dark signal was detected by the algorithm introduced in Sect 3.1 were classified as hot pixels. This is illustrated in Fig. 11 which indicates hot pixels of both ACCDs as orange, red, and grey squares. Pixels marked orange indicate hot pixels that show sporadic shifts in the mean dark current (as depicted in Fig. 6) whereas red pixels mark hot pixels which exhibit RTS features (see Fig. 5). The grey hot pixels became apparent only towards the end of the observing period (2018-09-02 until 2020-05-20) which is why time was too short for a proper

subdivision of the dark current anomaly. In total, there are 23 hot pixels in the Mie channel and 22 in the Rayleigh channel. This means that after 20 months in orbit 6 % of the pixels of both ACCDs used to acquire atmospheric signals show dark current anomalies. As can be seen from Fig. 11 the distribution of hot pixels on the ACCDs can be considered rather random. Almost all layers and all columns are affected on the two devices.

Table 2 provides a more detailed description of the dark current anomaly of the hot pixels of both ACCDs. The same color-coding as in Fig. 11 is applied to categorize the dark current anomaly. Apart from that, the number of median shifts and number of RTS levels as well as the dates of the first appearance are indicated. The classification shows that almost half (45 %) of the hot pixels show RTS-like characteristics. It is notable that pre-launch characterizations already indicated the two hot pixels Mie [16, 15] and Mie [24, 3].

The temporal evolution of the first appearance of the hot pixel anomaly (as listed in Table 2) is displayed in Fig. 12. It can be seen that the increase of the hot pixel number with time is not perfectly linear. On the one side there seem to be periods where hot pixels occurred at a higher rate (e.g. 2019-01 to 2019-02) but on the other side there are also periods with very few anomalies (e.g. 2019-10 to 2020-01). However, no correlation between the hot pixel emergences and space weather activity was found. The mean time difference between two anomalies is 14.68 days with a rather large standard deviation of 12.25 days. Linear extrapolation, which assumes that the hot pixel generation rate does not change with time, as indicated by the blue dashed in line in Fig. 12 reveals that around 9.5 % of the pixels will be hot after three years in operation at the end of the nominal mission lifetime in November 2021. It is worth mentioning that the solar activity is currently at a minimum and will be increasing in the upcoming months. It will be interesting to see whether this will change the rate of the hot pixel generation.

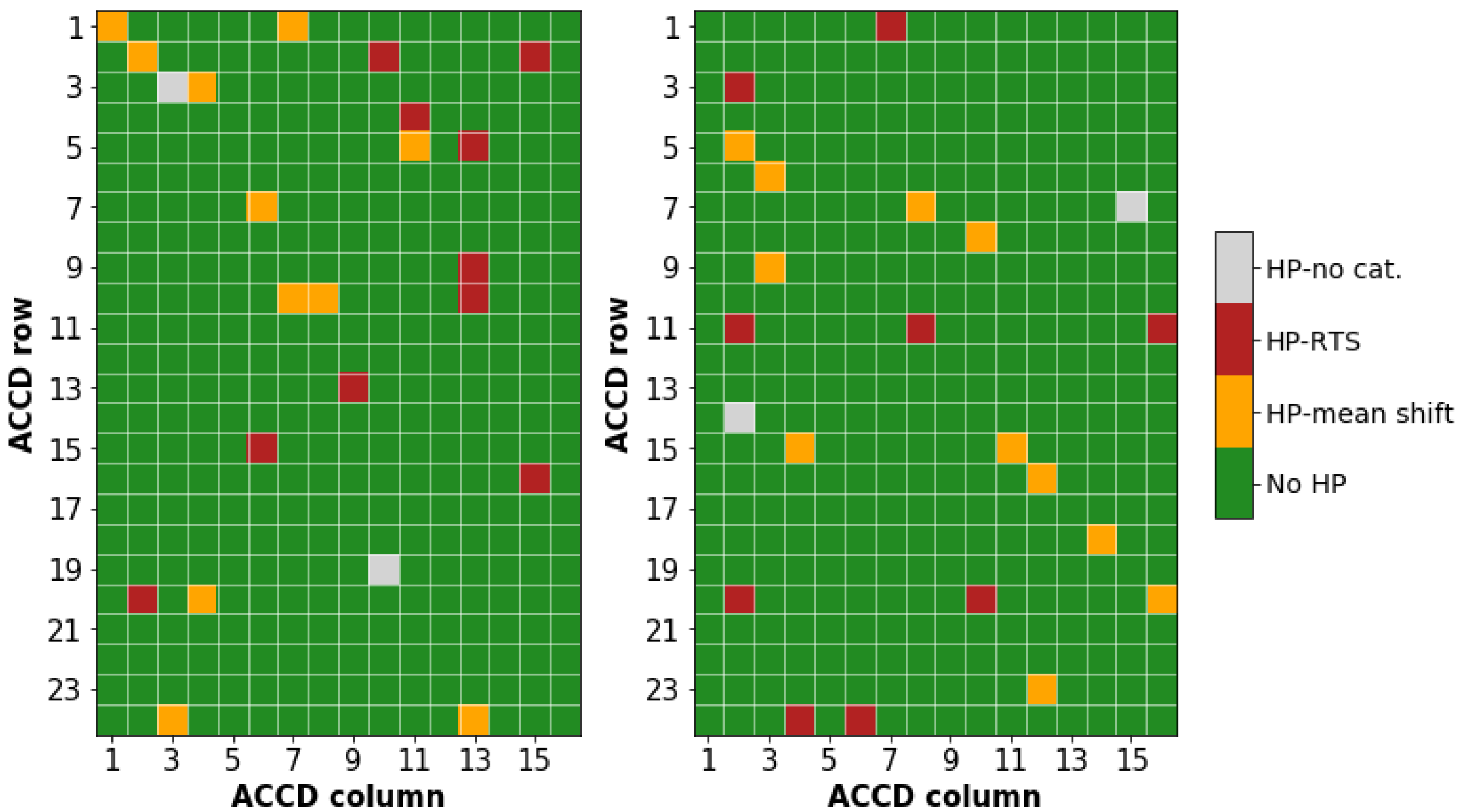

**Figure 11: Overview of hot pixels for the Mie (left) and Rayleigh (right) ACCDs (row #25 for the solar background not shown). Pixels with no detected permanent dark current anomaly are marked with green squares whereas orange, red and grey colours indicate pixels which exhibit permanent dark current anomalies. Orange pixels indicate pixels that sporadic show shifts in the median dark current signal and red pixels show RTS characteristics. For grey pixels the dark current anomaly occurred towards the end of the observation period such that further categorization was not possible. Status from 14-06-2020.**

| Position | Description | Date |
|---|---|---|
| [1, 1] | Single mean shift | 2018-09-03 |
| [1, 7] | Single mean shift | 2018-09-11 |
| [2,2] | Single mean shift | 2019-06-27 |
| [2, 10] | Multi-level RTS | 2020-02-28 |
| [2, 15] | Two-level RTS | 2018-10-24 |
| [3, 3] | - | 2020-05-10 |
| [3, 4] | Single mean shift | 2019-01-13 |
| [4, 11] | Two-level RTS | 2019-07-19 |
| [5, 11] | Mean shifts | 2019-10-03 |
| [5, 13] | Two-level RTS | 2019-01-09 |
| [7, 6] | Single mean shift | 2020-02-11 |
| [9, 13] | Multi-level RTS | 2019-08-08 |
| [10, 7] | Mean shifts | 2019-02-03 |
| [10, 8] | Mean shifts | 2019-02-03 |
| [10, 13] | Multi-level RTS | 2019-04-26 |
| [13, 9] | Multi-level RTS | 2018-10-21 |
| [15, 6] | Two-level RTS | 2020-03-14 |
| [16, 15] | Two-level RTS | before launch |
| [19, 10] | - | 2020-05-05 |
| [20, 2] | Multi-level RTS | 2019-03-31 |
| [20, 4] | Mean shifts | 2018-12-05 |
| [24, 3] | Mean shifts | before launch |
| [24, 13] | Single mean shift | 2019-05-21 |

| Coordinates | Description | Date |
|---|---|---|
| [1, 7] | Multi-level RTS | 2019-02-20 |
| [3, 2] | Multi-level RTS | 2019-05-08 |
| [5, 2] | Mean shifts | 2018-11-04 |
| [6, 3] | Mean shifts | 2018-12-17 |
| [7, 8] | Mean shifts | 2019-10-29 |
| [7, 15] | - | 2020-04-18 |
| [8, 10] | Single mean shift | 2020-01-28 |
| [9, 3] | Single mean shift | 2018-09-03 |
| [11, 2] | Multi-level RTS | 2018-09-07 |
| [11, 8] | Two-level RTS | 2019-06-15 |
| [11, 16] | Multi-level RTS | 2019-03-17 |
| [14, 2] | - | 2020-05-07 |
| [15, 4] | Mean shifts | 2018-11-24 |
| [15, 11] | Mean shifts | 2019-01-11 |
| [16, 12] | Mean shifts | 2020-02-05 |
| [18, 14] | Single mean shift | 2019-05-17 |
| [20, 2] | Multi-level RTS | 2019-08-01 |
| [20, 10] | Multi-level RTS | 2019-01-27 |
| [20, 16] | Mean shifts | 2019-08-17 |
| [23, 12] | Mean shifts | 2020-03-14 |
| [24, 4] | Multi-level RTS | 2019-08-29 |
| [24, 6] | Multi-level RTS | 2019-12-21 |

**Table 2: Mie (left) and Rayleigh hot pixels (right). Orange and red shaded rows indicate hot pixels that show sporadic shifts and RTS behaviour, respectively. Grey shaded rows mark hot pixels where further categorization was not possible. In the column "Date" the date of the first appearance of the anomaly is listed. Status from 14-06-2020**

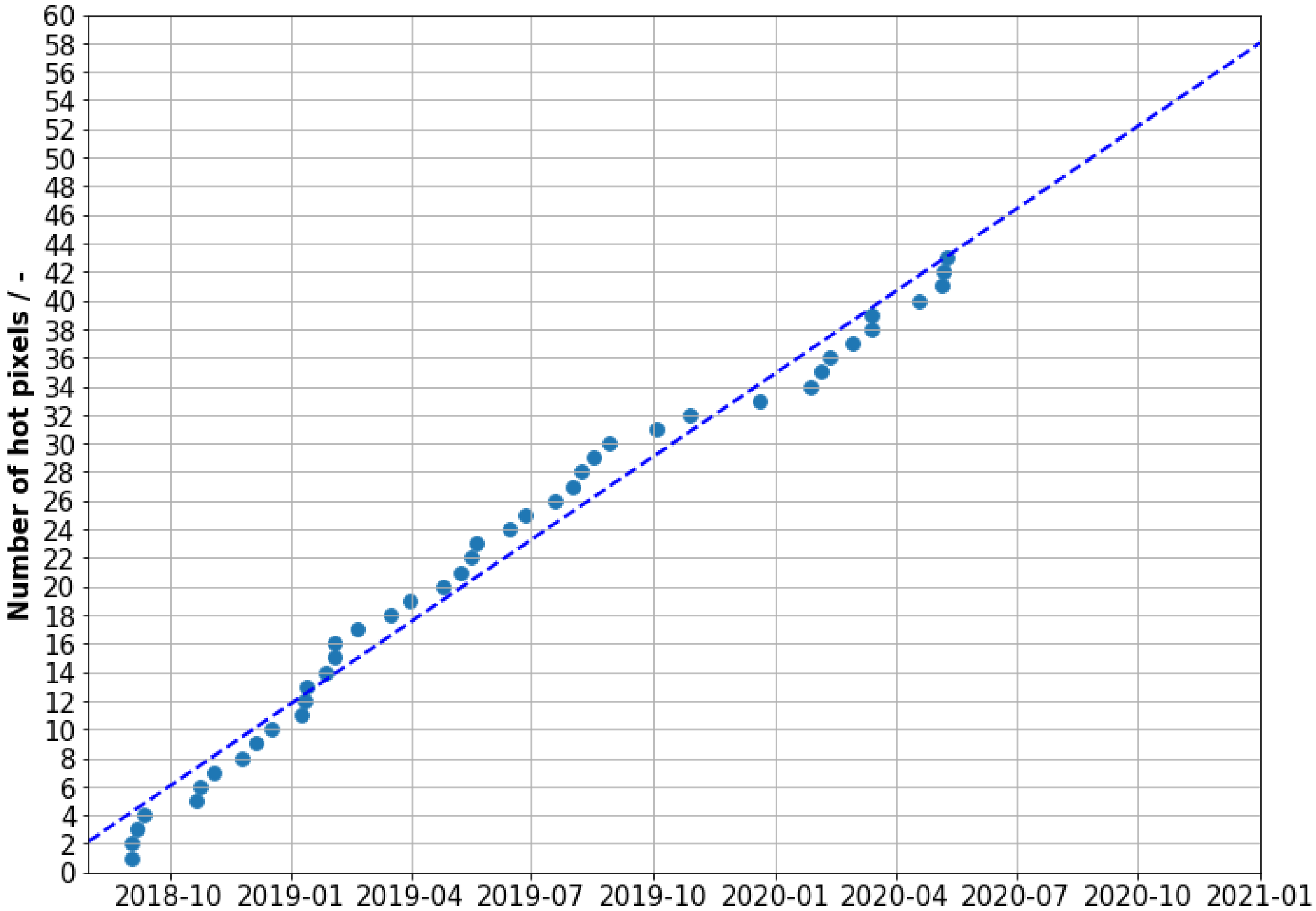

**Figure 12: Temporal evolution of hot pixel anomalies as listed in Table 2. The blue dots indicate the date of the first appearance of the dark current anomaly. The dashed blue line indicates a linear fit applied to the data points. Hot pixels Mie [16, 15] and [24, 3] which were already present before launch are not considered in the plot. Status from 14-06-2020**

## 4.2 Hot pixel signal levels

Figure 13 shows the median dark signal value of the Mie and Rayleigh hot pixels in ascending order of their dark current level. In order to show the spread of the dark signal values, the scaled MAD is indicated by the black error bars. Given that the dark signal values of pixels that show nominal behavior are Gaussian distributed (see Fig. 4), it might seem reasonable to use a hot pixel threshold based on the standard deviation and the mean. Thus, the dashed black lines in Fig. 13 indicate the median value + 3* scaled MAD of dark signal values obtained from all ACCD pixels after removing hot pixels which is 2.28 LSB and 1.54 LSB for the Mie and Rayleigh channel, respectively. Due to the fact that many Aeolus hot pixels only show very small shifts in the mean dark signal and even return to a normal dark signal after some time (see Sect. 4.2.2), many hot pixels would have been undetected using this simple threshold technique. This points out the necessity to use the sophisticated detection algorithm as introduced in Sect. 3.1.

Figure 13 also shows the wide range of the hot pixel median dark signal levels. For the Mie and Rayleigh channel the range is from 0.31 LSB to 19.83 LSB and 0.11 LSB to 15.93 LSB, respectively. As depicted in Sect. 3.3, the hot pixel induced bias can quite easily be estimated for the Rayleigh channel. Assuming atmospheric signal intensities of 1000 LSB through both

Rayleigh filters, a hot pixel induced signal elevation of 15.93 LSB leads to a hot pixel induced wind error of already about 4.11 m/s HLOS. Moreover, the figure gives a first glimpse of the different hot pixel characteristics in terms of dark signal level fluctuations as indicated by the error bars. The scaled MAD ranges from 0.69 LSB to 2.39 LSB and from 0.49 LSB to 2.57 LSB for the Mie and Rayleigh channel, respectively. Large values for the scaled MAD arise from a large spread of the RTS levels as explained in the following section.

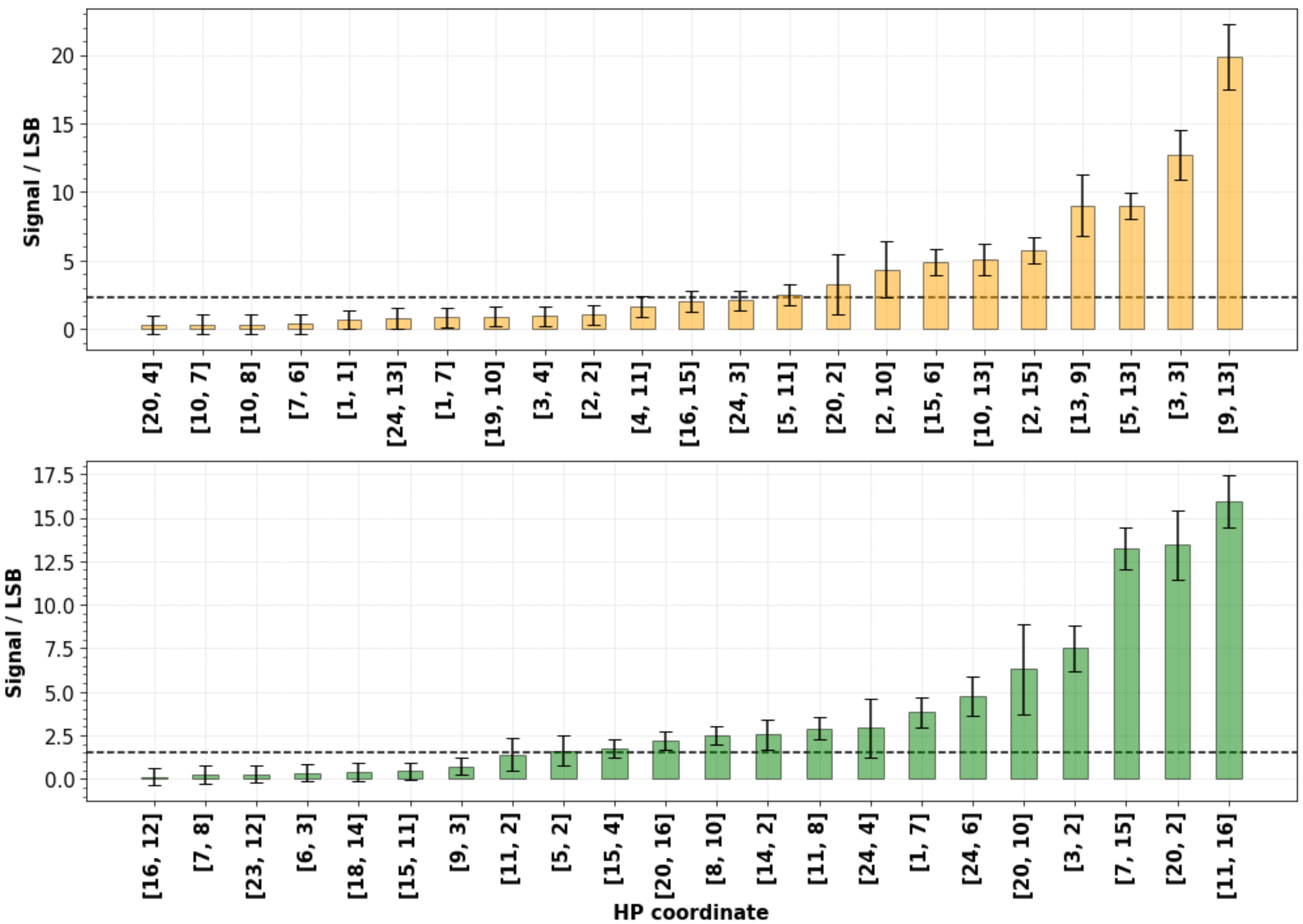

**Figure 13: Median dark signal values of the Mie (top) and Rayleigh (bottom) hot pixels obtained from observations after the first appearance of the hot pixel. The error bars represent the scaled MAD as a measure of the standard deviation. The horizontal dashed black line indicates the median + 3 * scaled MAD of the dark signal values obtained from all ACCD pixels after removing hot pixels.**

### 4.2.1 RTS characteristics

The majority of the hot pixels which were defined as RTS pixels show more than two levels. RTS characteristics with two distinct levels were only observed in 27 % of the cases. Apart from that, it is apparent that the RTS levels are quite different from each other. An overview of the different levels of the RTS steps and the temporal characteristics is reported in Fig. 14.

The dashed red line in the top plot which shows the average of the RTS levels per pixel indicates the large variation of the mean RTS levels. It ranges from 1.68 LSB as observed for Mie [4, 11] to 20.20 LSB observed for Mie [9, 13]. Also, the range (max-min) of the RTS level varies a lot. For multi-level RTS pixels, Mie [13, 9] shows the largest range with a value of 9.55 LSB. The minimum range of 1.74 LSB is observed for Ray [3, 2]. It appears that two-level RTS pixels exhibit a smaller range. For them, the range only varies between 0.30 LSB (Mie [15, 16]) and 0.77 LSB (Ray [11, 8]). Furthermore, there does not seem to be a correlation between the number and the range and average of RTS levels. The bottom plot of Fig. 14 shows the average number of steps within an interval of 500 observations. There appears to be the tendency that hot pixels with multiple RTS level such as Mie [9, 13] or Mie [13, 9] have higher switching frequencies than such RTS pixels with only two levels. There are also RTS pixels with a very moderate switching frequency with values close to zero, for example, Mie [4, 11] or Ray [11, 8].

For the two RTS pixels Mie [20, 2] and Rayleigh [20, 10] interesting on and off switching of RTS characteristics was observed. For instance, for Mie pixel [20, 2] (see Fig. 15, top) the RTS characteristics only appeared at observation number 27459. Initially, this pixel exhibited shifts of the mean dark current which started at observation 20205. Prior to the RTS period, the dark current was stable at an enhanced level of 1.30 LSB. A counterexample is Rayleigh pixel [20, 10] (see Fig. 15, bottom) which showed RTS-like features at the beginning and then stabilized towards the end.

Furthermore, it was found that some RTS pixels show a sharp initial signal rise after the onset followed by a fast decay and settling to an elevated signal level. The bottom plot of Fig. 15 shows dark signals of Rayleigh pixel [10, 16]. At the beginning, a signal increase from 0.15 LSB to 31.85 LSB occurred followed by a fast stepwise signal drop to a signal level of around 15.0 LSB. Similar effects were also observed for other RTS pixels such as Mie [5, 13], Rayleigh [1, 7], Rayleigh [7, 15], and Rayleigh [20, 2]. It is quite likely that even more RTS pixels show this effect but due to the rather coarse temporal resolution of DUDE measurements (4 times per day) it is possible that this effect was not captured at all times.

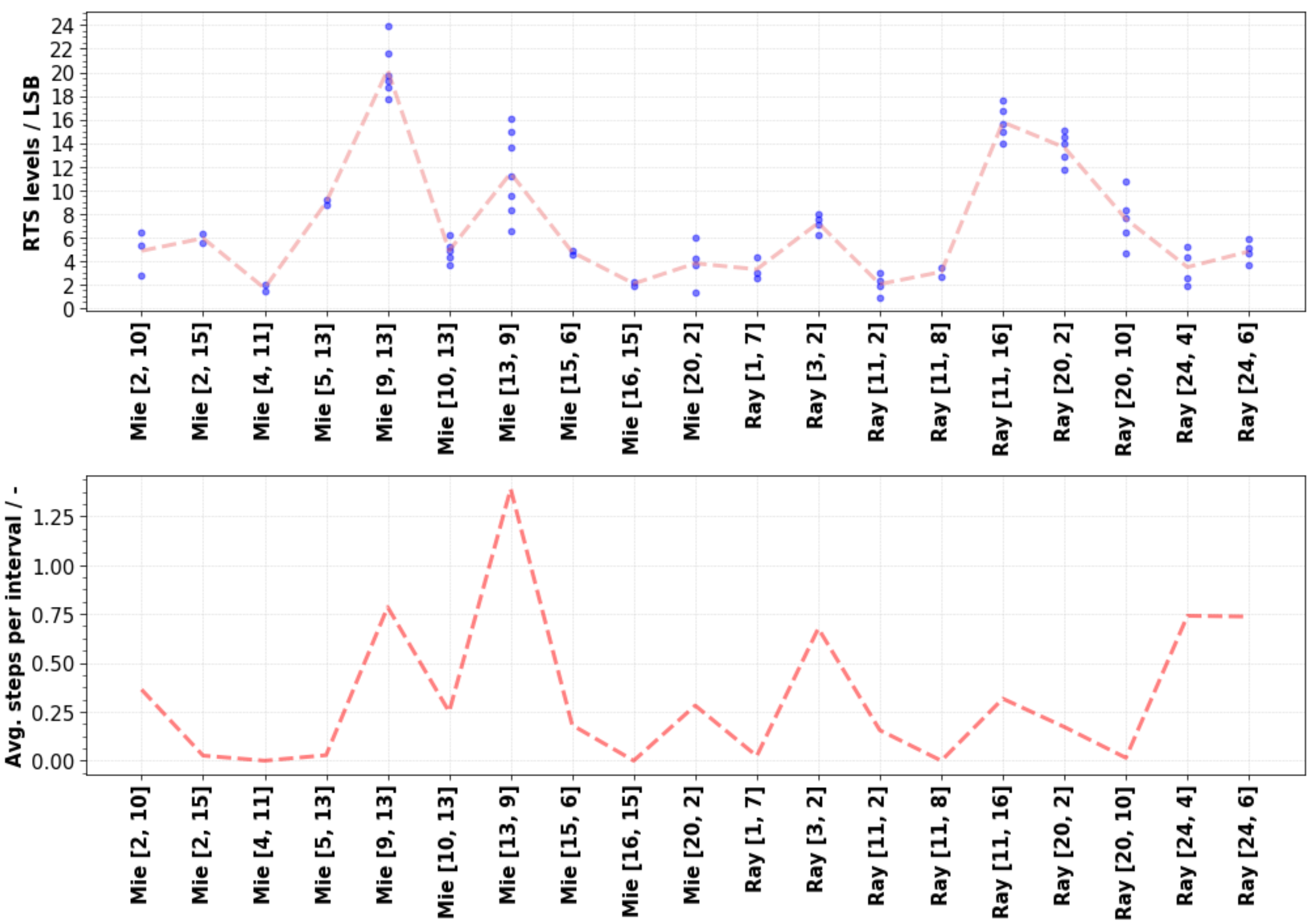

**Figure 14: (Top): RTS levels of the RTS hot pixels. The blue dots indicate the different levels of the RTS. The dashed red line indicates the average of the RTS levels of each pixel. (Bottom): Average number of steps per interval which spans 500 observations.**

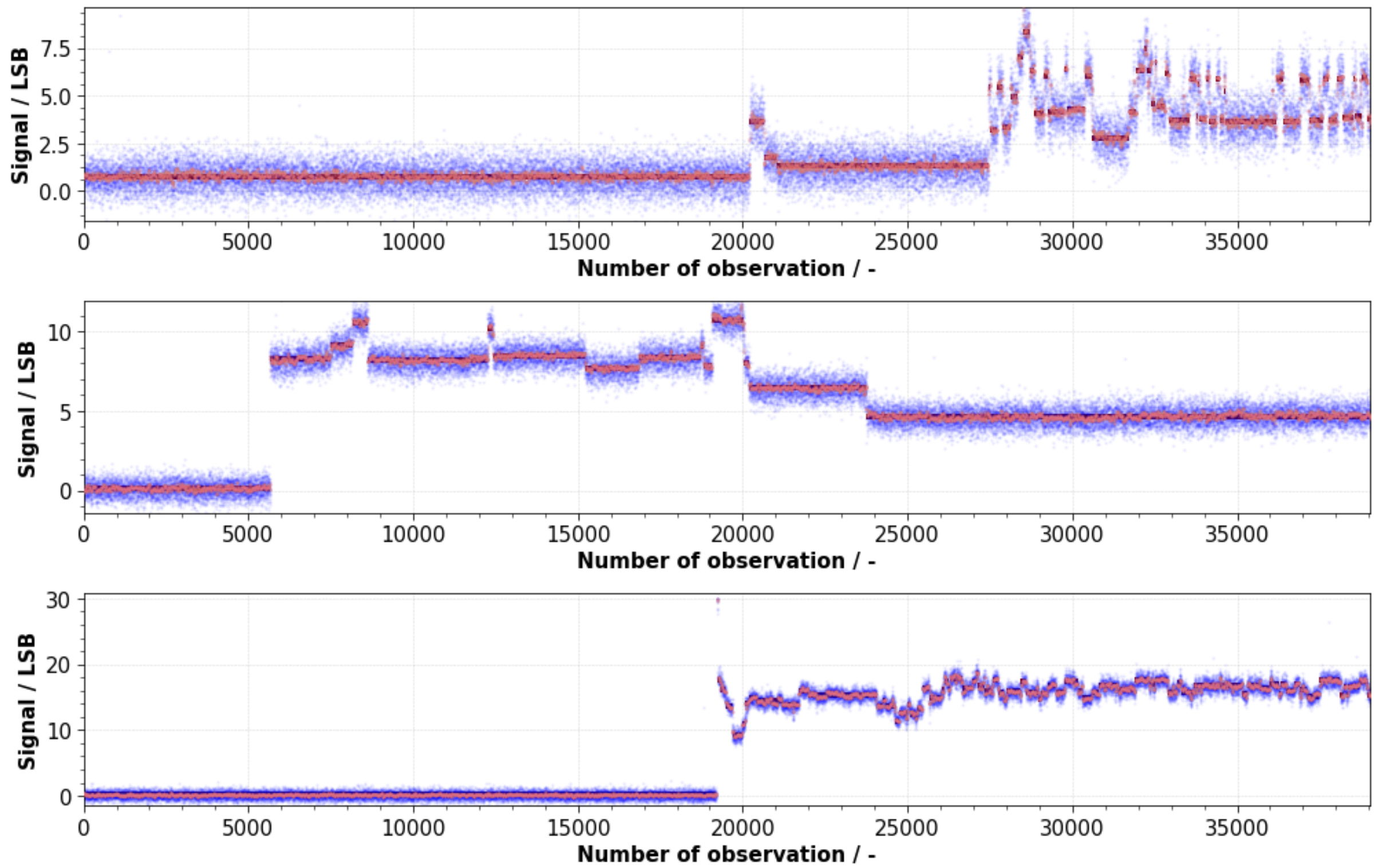

**Figure 15: Dark signals of Mie pixel [20, 2] (top), Rayleigh pixel [20, 10] (middle) and Rayleigh pixel [11, 16] (bottom). The blue dots indicate dark signal intensities at observation level. The solid black line(s) indicate the median value of each detected segment. The red dots show dark signal intensities after median filtering (window size of 20 observations) applied to each detected segment.**

### 4.2.2 Sporadic dark signal shifts

Figure 16 provides an overview of hot pixels that show sporadic median shifts of their dark signal level but were not classified as RTS pixels (orange in Fig. 11 and Table 2). The plot shows the median value of the detected signal segments (see Sect. 3.1) as a function of the observation number. The changes are shown relative to the baseline level, i.e. the signal level of the first value, such that negative and positive changes appear blueish and reddish, respectively. The figure indicates that 59 % of the shown pixels exhibit multiple shifts of the median signals whereas 41 % of the pixels show one single shift of the median dark signal.

The figure suggests that median signal levels are higher for pixels with multiple shifts than for pixels which only show one single shift. The majority of segments with median values larger than 2.0 LSB are observed for pixels that show multiple shifts. The only exception is Rayleigh [8, 10] which exhibits one large step from 0.18 LSB to 2.55 LSB. The highest level could be observed for Rayleigh pixel [20, 16] with a value of 3.45 LSB between observation number 26000 and 26075.

Interestingly, pixels with one single shift show changes towards higher as well as lower values. For instance, the dark signal of Mie pixel [1, 1] decreased at observation number 1065 from 0.95 LSB to 0.70 LSB. The same holds true for Mie pixel [7, 6] which dropped from 0.80 LSB to 0.40 LSB at observation number 33700. All other hot pixels that exhibit one single median shift show changes towards higher values, such as Mie [1, 7] or Rayleigh [8, 10].

Once a transition of the median occurred, the dark signal usually does not fall back to its original level. However, for a considerable number of hot pixels annealing effects with a drop back of the dark signal close to its original level could be observed. This effect was seen for Mie pixels [10, 7] and [10, 8]. Both showed a signal increase of 0.30 LSB to an elevated level of 0.65 LSB. Afterwards, the signal dropped back close to its original level of 0.30 LSB. Similar results were found for the following hot pixels: Mie [20, 4], Rayleigh [6, 3], Rayleigh [7, 8] and Rayleigh [16, 12].

Furthermore, Fig. 16 indicates that the changes for neighboring Mie pixels [10, 7] and [10, 8] happen almost simultaneously. In addition, the amplitude of the change is of similar value. For both pixels, the signal segmentation algorithm detected an intermittent signal increase to 0.65 LSB between observation numbers 11700 to 13950 and 11720 to 13930, respectively. The slight difference may arise from uncertainties of the algorithm in detecting the exact indices. Nevertheless, it is reasonable to assume that these anomalies arise from the same root cause.

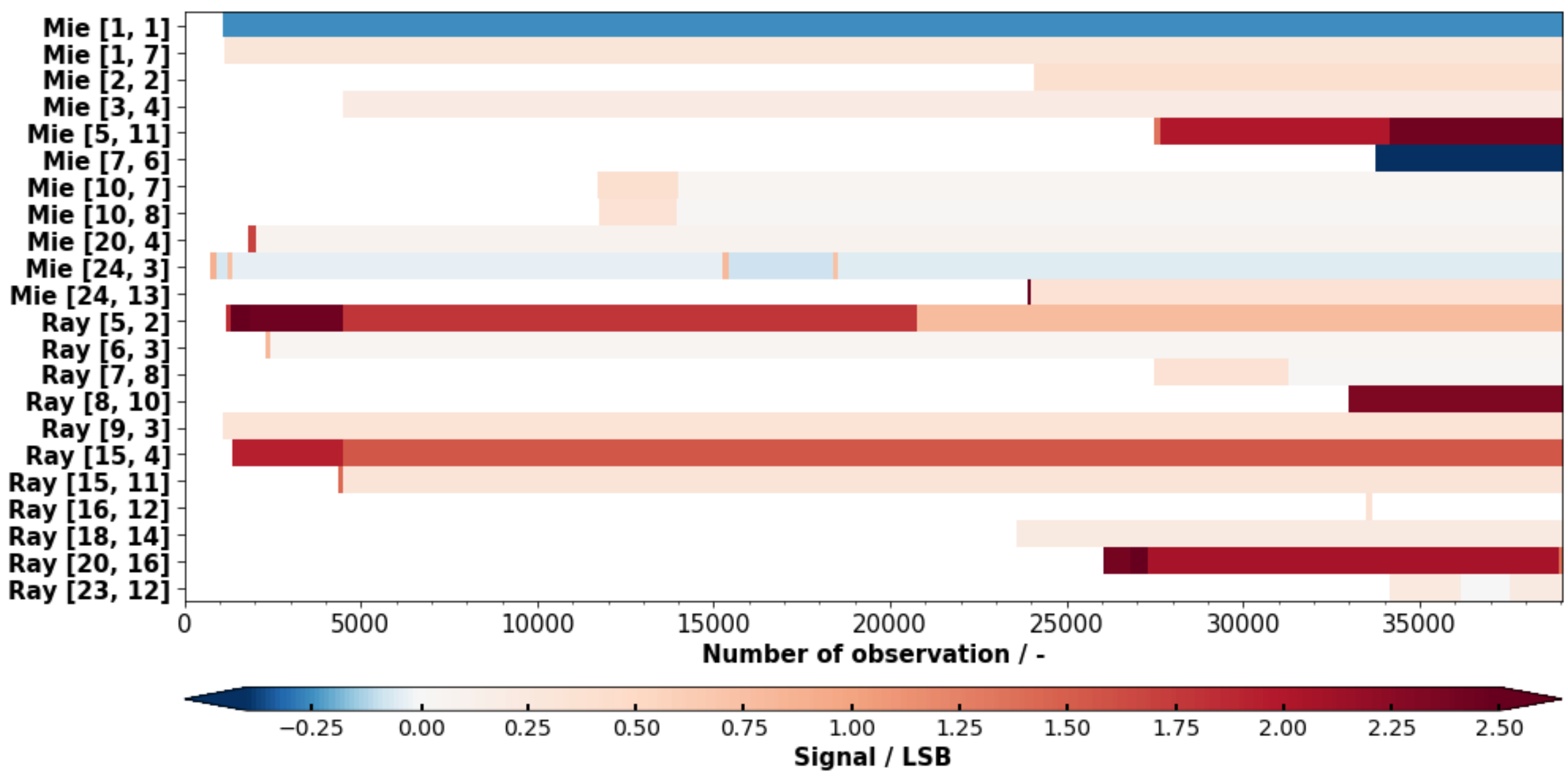

**Figure 16: Dark signal median shifts as a function of the observation number. Changes are shown relative to the start value. Negative changes appear blueish positive changes appear reddish.**

### 4.3 Transient events

The application of the transient event detection algorithm (described in Sect. 3.2) allows for a statistical analysis of the location as well as of the amplitudes of the signal spikes at measurement level. Considering all pixels of both channels, only 4192 out of 1808310 analyzed measurements are affected by transient events corresponding to 0.24 % of all measurements. Analysis of a continuous measurement section revealed a count rate of 48 and 52 transient events per 30 minutes for the Mie and Rayleigh channel, respectively. Figure 17 provides an overview of the outcome of the statistical analysis. The left plot shows that in about 50% of the cases more than one pixel is affected at the same time. The maximum number of simultaneously affected pixels was observed to be 29. This result is not surprising as an incoming particle may traverse multiple pixels on its path through the ACCDs (Waltham, 2010). The right plot of Fig. 17 shows the signal levels of the transient events at measurement level. The minimum and maximum observed signal levels were 16.5 LSB and 64362.0 LSB, respectively. This is very large given the fact the maximum detectable value is 65535 LSB. In about 50% of all cases, the signal levels are in the range between 100 LSB and 1000 LSB. Apart from that, the spatial distribution of transient events on both ACCDs was analyzed and found to be random (not shown here). Thus, an equal likelihood for the occurrence of transient events for all pixels with respect to the pixel number can be assumed.

Next, the geolocation of the transient events was analyzed. Therefore, the number of transient events for each pixel falling in latitude-longitude bins of 4°-width was counted. This number was divided by the total number of measurements falling in that bin yielding the relative frequency of occurrence as a function of the geolocation. This information is shown in Fig. 18. In general, the relative frequency of transient events is very low with a maximum value of only 0.059 % and occur all around the globe. However, there appears to be an accumulation of such events in the region of the SAA. In a region between 40°S to 60°S latitude and 60°W to 30°W longitude, the relative frequency of transient events is three times higher compared to other parts of the globe.

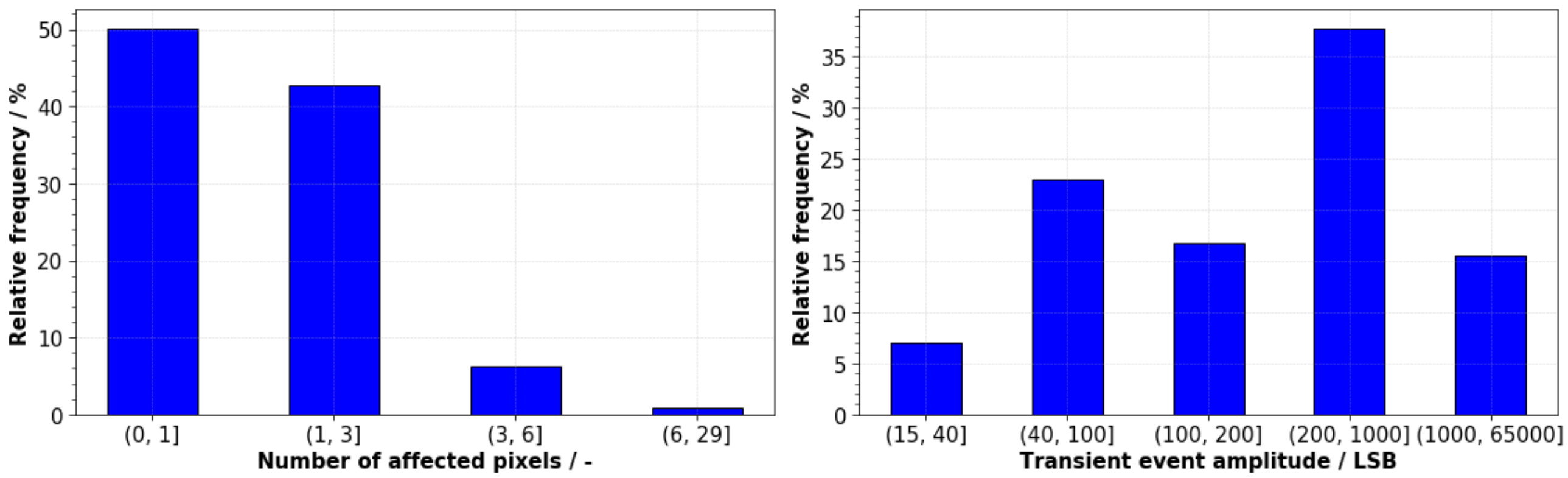

**Figure 17: Statistical analysis of transient events for both ACCDs. (Left): Histogram of the number of pixels affected by one transient event. (Right): Histogram of amplitudes of the transient events at measurement level.**

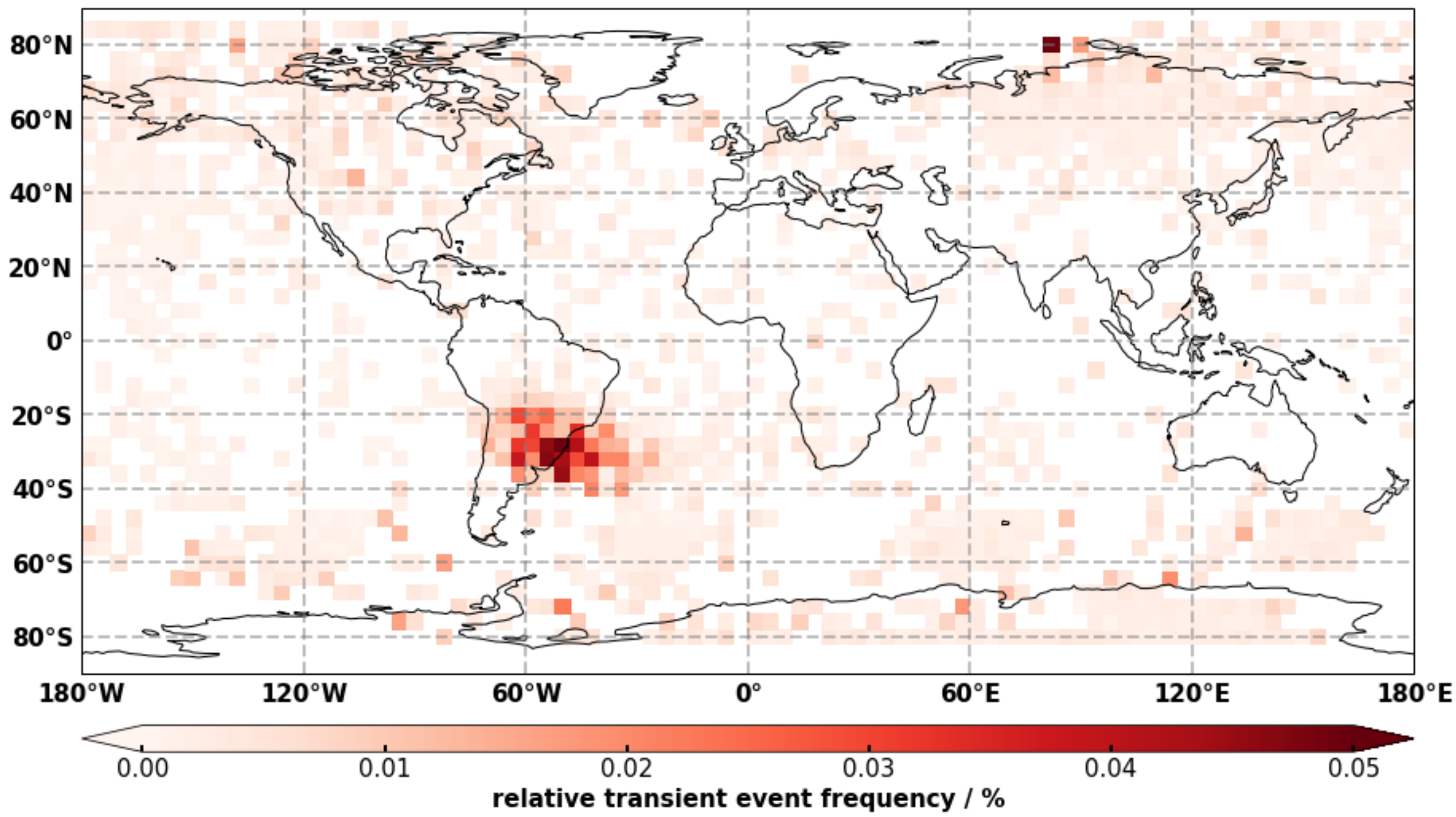

**Figure 18: Relative frequency of transient events as a function of geolocation obtained from DUDE measurements from September 2018 to June 2020.**

## 5 Discussion

This study revealed the existence of different kinds of dark signal anomalies for the Aeolus ACCDs. The use of a sophisticated segmentation algorithm showed that about half of the detected hot pixels exhibit RTS like features for which the majority has multiple levels. The other half of the hot pixels shows one or more sporadic shifts in the median dark signal. Despite this detailed analysis, the question of whether the occurrence of hot pixels is linked to the fact that Aeolus is operated in a space environment with harsh radiation conditions still remains. To answer this question, a clear discrimination of radiation-induced and CIC hot pixels would be necessary. Unfortunately, the herein presented categorization of hot pixels into RTS and sporadic shift pixels does not provide a clear answer. Although reported in literature that RTS effects are mainly observed in proton-irradiated CCDs and are mostly related to displacement damage (Hopkinson et al., 1996; Smith et al., 2004), it cannot be excluded that CIC pixels may exhibit RTS features, too. Dedicated measurements would be needed to distinguish between both effects. Typically, the response of radiation-induced hot pixels to changes of the operating temperature would be different compared to the response of CIC hot pixels. Furthermore, a sensitivity test with changing clocking parameters of the ACCDs would be helpful. Due to instrument safety concerns and technical limitations, such tests cannot be performed in-orbit during the operational phase of the mission. Therefore, on-ground sensitivity tests with structurally identical ACCDs would be needed for a clear discrimination between radiation-induced dark current and CIC hot pixels. However, setting up such a test campaign is not straightforward as no spare ACCD of the same batch as the ACCDs used in-orbit is available for such tests.

The fact that two hot pixels Mie [16, 15] and Mie [24, 3] – both of them in different hot pixel categories and with similar characteristics of hot pixels emerged in-orbit – were already present before launch supports the hypothesis of an origin which might be independent of space radiation. However, other radiation sources within the instrument or even within the ACCD package might play a role, too. It is worth mentioning that the optical window used for the housing of the ACCDs (see Fig. 1) is a BK7 glass. The material BK7 contains potassium and about 0.01 % of the potassium comprises the radioactive isotope potassium-40. It produces beta-radiation which may become visible as transient events on the ACCDs (Mackay, 1986). During on-ground tests before launch, the count rate of transient events due to the BK7 glass was measured to be 18 events per minute and $cm^2$. Considering the size of the memory zone with 0.00354 $cm^2$, this corresponds to 3.05 events per hour. The count rate for in-orbit transient events was determined to be 100 events per hour (see Sect. 4.3) which is 33 times larger than the count rate expected due to the BK7 window. This indicates that the BK7 window plays a minor role in the generation of transient events on the Aeolus ACCDs. However, the connection between transient events and the emergence of permanent dark current signal anomalies is unclear.

The results in Sect. 3.3 clearly demonstrate the effects of hot pixels on the wind retrieval. Although effects of RTS pixels were expected before launch, the effect on the quality of the wind products turned out to be much stronger than anticipated because of the lower than expected atmospheric path signals. The results show that already slightly increased dark signal values of single pixels have detrimental effects on the quality of the wind products of the affected range gate. Thus, robust mitigation approaches were needed to ensure the high quality needed for NWP models. It soon became clear that mitigation approaches on the hardware are not feasible. In principle, an attempt to anneal RTS pixels by heating-up the ACCDs device to temperatures above 80 °C for a longer time period would be possible (Nuns et al., 2007; Smith et al., 2004). However, due to safety concerns and technical issues this procedure cannot be performed in-orbit. The maximum temperature reached inflight was -10 °C during an instrument anomaly and the switch over to the second laser but this temperature was obviously too low for annealing. This is why a solution in the wind retrieval is considered to be the only practical way to address the hot pixel issue. Therefore, a dedicated hot pixel correction scheme to correct the DSNU of the ACCDs was successfully implemented into the operational processing already after launch using four DUDE measurements per day (see Sect. 3.3). In this paper, it was demonstrated that the correction approach significantly improved the quality of the wind products and paved the way for the operational use of Aeolus data in NWP models. However, it also was pointed out that the correction scheme is imperfect as dark signal shifts between two DUDE measurements cannot be corrected. Furthermore, potential changes in the CTE of single pixels and resulting PRNU can also not be corrected with the DUDE measurements.

The herein performed categorization allows for an estimation of the performance of the DUDE correction for the individual hot pixels. It is obvious that the performance is good for hot pixels with sporadic median shifts (orange in Fig. 11 and Table 2). The most critical hot pixels are considered to be multi-level RTS pixels with a high fluctuation frequency between the levels. According to the findings of Sect. 4.2.1, pixels [9, 13] and [13, 9] of the Mie and pixels [3, 2], [24, 4] and [24, 6] of the Rayleigh ACCD have the largest impact on the performance of the presented NRT hot pixel correction scheme. For these pixels, an increased frequency of DUDE measurements might be beneficial. This might generally be desirable considering the

fact that about 9.5 % of the atmospheric pixels will be hot at the end of the nominal mission lifetime in November 2021 assuming that their occurrence rate does not change. However, this also emphasizes the necessity of reprocessing Aeolus data products. Reprocessing allows correcting remaining hot pixel induced signal steps and thus, helps to further mitigate effects of multi-level RTS hot pixels.

## 6 Summary

The Aeolus satellite carrying the Doppler Wind Lidar ALADIN was successfully launched into space in August 2018. One of the major issues concerning the data quality has been related to increased dark current signals of single pixels on the Aeolus ACCDs leading to wind errors of up to 4 m/s. This paper presents a detailed characterization of the dark current anomalies, discusses the impacts of these anomalies on the wind data products, and introduces hot pixel correction schemes for the NRT and re-processing. The correction scheme for the NRT processing paved the way for the operational assimilation of Aeolus data products into NWP models already one and a half years after launch.

In this paper, special measurements which characterize the dark current signal on the Aeolus ACCDs were analyzed in the period between 2018-09-02 and 2020-05-20. For this, sophisticated algorithms were developed to scan the dark signals for anomalies and categorize them. To detect permanent dark current anomalies, i.e. so-called hot pixels, a signal segmentation algorithm is introduced which is capable of detecting sudden shifts in the median dark signal. As a next step, hot pixels were further categorized into pixels exhibiting burst noise, so-called Random Telegraph Signal noise, and pixels that only show sporadic shifts of the dark current signal. Important characteristics of RTS hot pixels, like the number of RTS levels and amplitudes, could be identified by combining the signal segmentation algorithm with appropriate signal filtering. To detect transient dark current anomalies a peak detection algorithm was used.

The results revealed that 6 % of the ACCD pixels are hot pixels and that around 13 % of the pixels will be affected at the end of extended mission lifetime in November 2022 assuming that the hot pixel generation rate does not change. Moreover, it was demonstrated that the hot pixels show a wide variety of characteristics ranging from pixel exhibiting RTS-like characteristics with different dark signal levels and time constants to pixels with sporadic shifts in the mean dark current signal. Apart from that, it could be shown that the relative frequency of the occurrence of transient events is three times higher in the region of the South Atlantic Anomaly compared to other parts of the globe.

Despite the detailed analysis, no conclusion about the root cause of the hot pixel issue could be drawn. It is still not clear whether the occurrence of hot pixels is due to clock induced charges CIC or linked to effects from the radiation environment in space. To answer this question, dedicated on-ground sensitivity tests with structurally identical ACCD devices would be needed. It was also discussed that mitigation approaches inflight on the hardware side are considered too high risk. Consequently, the preferred way to address the hot pixel issue is in the data retrieval of the ground processing.

A combination of dedicated instrument calibration modes and ground processors were developed to allow for a pixel-wise dark signal correction of the wind signals already shortly after launch. It was demonstrated that this correction is capable of

correcting for the dark signal non-uniformity arising from hot pixels on the ACCD. It is expected that this correction will work throughout the whole mission lifetime no matter how many hot pixels will be present. The execution of the calibrations was optimized to 4 min and 4 times per day in response to the fluctuations of the RTS levels of the hot pixels. It was shown that the performance of this correction scheme depends on the behavior of each hot pixel. With the detailed categorization of hot pixel anomalies, it was possible to assess the effect of hot pixels on the wind error as well as on the NRT hot pixel correction scheme and thus, on the quality of the data products. It turned out that mainly multi-level RTS pixels with high fluctuation between the dark signal levels have the largest impact on the performance of the hot pixel correction scheme.

For potential follow-on missions, it might be beneficial to operate the ACCDs at a lower temperature than -30 °C. This would not only decrease the dark signal level on the ACCDs but also lower the fluctuation rate of RTS pixels. In addition, it could be assessed whether it is possible to decrease the on-board accumulation of signals in the memory zone of the ACCDs to decrease the residence time of the signals in the memory zone.

*Data availability.* The analysis in this paper is based on Aeolus Level 1A products which is not publicly distributed via the ESA Aeolus Online Dissemination System. Access to the full collection of Aeolus products is only granted to Aeolus Cal/Val projects formally accepted by ESA.

*Author contributions.* FW performed the data analysis and prepared the manuscript. TK and OR largely contributed to the development of the presented methods and the in-orbit hot pixel correction. DW is the instrument manager of ALADIN at ESA-ESTEC and supported the investigation. DH developed the operational processor for the in-orbit hot pixel correction. MR provided the data for comparison between Aeolus L2B Rayleigh-clear HLOS winds and the ECMWF model equivalents and provided input to the discussion of the hot pixel induced wind bias. OSP, MS and RB helped to prepare the Sect. about the Aeolus ACCD and contributed to the interpretation of the results from an engineering point of view. TP is the Aeolus mission manager at ESA-ESRIN and supported the investigation.

*Competing interests.* The authors declare that they have no conflict of interests.

*Disclaimer.* The presented work includes preliminary data (not fully calibrated/validated and not yet publicly released) of the Aeolus mission that is part of the European Space Agency (ESA) Earth Explorer Programme. This includes wind products from before the public data release in May 2020 and/or aerosol and cloud products, which have not yet been publicly released. The preliminary Aeolus wind products will be reprocessed during 2020 and 2021, which will include in particular a significant L2B product wind bias reduction and improved L2A radiometric calibration. Aerosol and cloud products will become publicly available by spring 2021. The processor development, improvement and product reprocessing preparation are performed by the Aeolus DISC (Data, Innovation and Science Cluster), which involves DLR, DoRIT, ECMWF, KNMI,
CNRS, S&T, ABB and Serco, in close cooperation with the Aeolus PDGS (Payload Data Ground Segment). The analysis has been performed in the frame of the Aeolus Data Innovation and Science Cluster (Aeolus DISC).

*Special issue statement.* This article is part of the special issue "Aeolus data and their application (AMT/ACP/WCD inter-journal SI)".

*Acknowledgements.* The authors acknowledge the Aeolus Space and Ground Segment Operations teams, the Aeolus Data Innovation and Science Cluster as well as the technology centres at Airbus Defence and Space and ESA for their innovative spirit. We thank Florian Ewald for the internal review of this manuscript.

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
