# Peer review of "Characterization of dark current signal measurements of the ACCDs used on-board the Aeolus satellite"

_Atmospheric Measurement Techniques, 2020_

## Referee Comment (RC1) · Artem Feofilov (Referee) · 6 Jan 2021

General comments

The manuscript is dedicated to the improvement of experimental data coming from the ALADIN lidar onboard the Aeolus satellite, which provides continuous measurements of atmospheric winds, aerosols, and clouds. The manuscript seeks to solve an important problem of identifying and fixing the experimental issue associated with the so-called "hot pixels" of the ALADIN's ACCD detectors. The authors suggest a method for pinpointing these pixels, introduce dedicated calibration modes, correct the signals from the affected pixels, and show the results of wind retrievals from the fixed signals.

[Figure]

The problems of this kind are not new to the experimental physics, but in this case the study was hindered by the fact that the experimental setup was not available for direct testing in the lab. Still, the authors show that it provided sufficient amount of information for performing the analysis and for fixing the problem. In general, the real state of the atmosphere and the retrieved atmospheric data are linked through a number of conversions and convolutions, each of which can affect the quality of the retrieved parameters. In the case under consideration, the biggest challenge was associated with the missing or damaged pieces of information, needed for the retrieval, namely, with pixels providing the profiles yielded from fringe-imaging or double-edged techniques in some ACCD rows.

Since the backscattered photons carrying the information about the atmospheric properties in this setup are stored in ACCD matrix, one can split the solution of the problem to several steps: (i) identifying the pixels, which are not reliable; (ii) correcting or excluding these pixels from the retrieval; (iii) depending on the previous choice, one has either use the fixed values or modify the retrieval algorithm; (iv) since the initial retrieval algorithm did not take into account the possibility of hot pixels spoiling the inputs, one has a right to impose physical constraints on the retrieved data to fix the affected points. The authors did an excellent job for (i) and then they followed the correction scheme of (ii) and ended up with (iii). From this point of view, the work is impeccable. Still, I'd suggest to consider a bigger picture and to look at the problem at a different angle. Perhaps, the authors did it in the background and found that it didn't solve the problem, but I found no trace of it in the manuscript, so at least this is worth a discussion.

Let me explain. Looking at Fig. 1 of the manuscript and comparing it with the Fig. 11, one can see that the experimental setup has a certain redundancy in a sense that the peak in the Mie signal almost never is narrower than 3 bins and, in some cases, a naked eye distinguishes 4 bins filled with non-zero signal. At the same time, Fig. 11 tells us that the situations when two adjacent horizontal pixels are "hot" are rare. Knowing that this detector is characterized by a low noise, one can make use
of the remaining available information and still get a reliable result. To prove this, I performed a simple test illustrated in the attached Figure. The panel (a) shows the Mie detector mask, which is consistent with Fig. 11 of the manuscript, but converted to a binary (good/bad) form. Panel (b) shows the simulated signal, which qualitatively resembles that of Fig. 1 of the manuscript, but which passes through the hot pixels of the mask (a) for demonstration purposes. For each row, the exact position of the peak corresponding to exact value of the wind is stored for reference. Panel (c) shows the same signal with hot pixels masked out. The Poissonian noise was added to the pixel values to imitate the detector's behavior. Then the fitting procedure based on sliding profile correlation approach similar to those used in [Goldberg et al., 2012] and [Feofilov and Stubenrauch, 2019] was applied both to a full set of input data and to a masked one. The procedure uses the knowledge of the profile of the fringe-imaged signal along the columns and this profile is supposed to be known with high accuracy. The resulting retrieval uncertainties are shown in panel (d) of the Figure. As one can see, the position of the peak retrieved from incomplete data does not change that much compared to the retrievals from the unmodified datasets, and the uncertainty in pixels converted to wind speed uncertainty is of the order of 0.03 m/s. This is just a rapid exercise, which should be done in a different way for Rayleigh signals, but it leads to an important question – even though the fixed hot pixels provide a dataset compatible with the rest of the processing chain, wouldn't it be easier and safer to exclude them from the consideration and to update the procedure? I understand that this is not what the manuscript is about, but it's a major philosophic question whether one should use fixed values from a damaged detector or use a reduced dataset profiting from the redundancy of the data. The latter approach does not diminish the significance of the work, but if it proves to give more reliable data through a simpler procedure, it should be considered.

The second question is about aforementioned step (iv) – I believe, the retrieval procedure could profit from the physical constraints of the following kinds : (a) point-to-point wind speed change and (b) point-to-point aerosol/cloud properties change. Both are

easy to justify and both can serve as an additional quality control mechanism at early stage – if sudden unphysical jumps are found, the pixels are removed from the retrieval and the values are interpolated, masked, and so on.

I have chosen "minor revisions" in the decision, but I'd like to see these questions addressed in the final version of the manuscript as well as the specific comments below.

Specific comments

Lines 200-209: if CIC noise is important, how does this fact match the "low-noise detector" statements above? Some numbers are needed here, so that the reader could make his/her own conclusions.

Line 278: can one prove this statement about the DUDE correction with some formula or reference? At the moment, there are only qualitative statements here.

Lines 300-320: perhaps, it's a matter of preferences, but how does this approach compare to a simple 3-sigma test? Another approach, which could be also useful for detecting hot pixels as well as identifying the nature of the noise is building and analyzing Fourier spectra of the temporal sequences for each pixel. Most probably, the spectra of hot pixels will be different from those of "normal" ones and hot pixels of a different nature will reveal this in the spectra, too.

Line 338: again, spurious changes could have been filtered out by Fourier smoothing procedure

Lines 400-410: see the general comments – perhaps, the discussion should be updated.

Lines 430-435: how does this correction compare to vertical interpolation?

Line 439: a median correction is applied, which does not eliminate sporadic events. Even though it smooths them out, their erroneous nature is included in the results. On the other hand, gradient-based or Fourier filtering would have removed a non-physical

part of the signal.

Line 500: it would be interesting to recalculate these 6% into a weighted percentage of pixels used in retrievals. For example, pixel [9,13] is used often whereas [1,1] is not.

Line 550: Linear trend is interesting here. If the damage is due to high energy particles hitting the ACCD then the slope should change with time, but 6% is too small a number for this to be noticed.

Line 689: cosmic particles partially penetrate the atmosphere, so this is not a 100% proof.

Lines 701-702: first, we did not see this in the manuscript and second, it should be considered in the light of the exercise demonstrated in General comments.

Line 752: numbers are missing here: uncertainty/bias after the correction vs uncertainty/bias before the correction.

Technical corrections

Lines 301, 342, and elsewhere – in some PDF viewers, the font used for Python module names looks strange.

References

Feofilov, A. G. and Stubenrauch, C. J., "Diurnal variation of high-level clouds from the synergy of AIRS and IASI space-borne infrared sounders", Atmos. Chem. Phys., 19, 13957–13972, https://doi.org/10.5194/acp-19-13957-2019, (2019).

Goldberg, R.A., A.G. Feofilov, W.D. Pesnell, A.A. Kutepov, "Inter-hemispheric Coupling during Northern Polar Summer Periods of 2002-2010 using TIMED/SABER Measurements", J. Atm. Solar-Terr. Phys., doi:10.1016/j.jastp.2012.11.018 (2012).
* * *
[Figure]

**Fig. 1.**

ACCD row

ACCD column   ACCD column   ACCD column

a)   b)   c)

ACCD row

Retrieved peak position error in ACCD column coordinates

No hot pixels

Retrieval with
hot pixels excluded

d)

---

## Referee Comment (RC2) · Anonymous Referee #2 · 16 Mar 2021

The focus of this paper is on analyzing the on-orbit hot pixel characteristics and emergence trends in the novel ACCD launched on the space-based wind lidar ADM-Aeolus, and mitigation of hot pixel effects on wind retrieval accuracy. Though the paper does not draw any firm conclusions about the potential root cause(s) of hot pixel emergence, this paper nicely sets the stage for such a discussion. Most of my comments are geared towards this discussion. I should mention that, in my opinion, a discussion of the root cause(s)/damage mechanism(s) is optional, as the authors' description of the strategies for mitigating the impact of hot pixels on wind retrievals, and detailed characterization of these anomalies, make this a valuable work in its own right. In fact, the author

could consider de-scoping some of the discussion on the root cause from this paper, and deferring it to a future work, if the author so wishes. A more detailed discussion of the root cause might be beyond scope, but I offer the following comments/questions to address (optionally) that might aid a future publication/study on the issue, or satisfy a curious reader of this paper. General questions: -How much shielding exists around the ACCDs on Aeolus, and/or what is the shielded radiation environment/dose (yearly DDD, TID)? -Has there been a detectable, steady trend/increase in the dark current observed over the course of the mission for pixels that have not experienced an anomaly? -What design deltas between the ACCD and previously flown CCDs (e.g., Hubble) might explain the observed anomalies? Inversely, what design elements do the ACCDs share with the CCD detectors of GOMOS on ENVISAT? -What radiation testing was conducted on the ACCDs prior to launch (proton energies and fluence steps, TID dose steps, heavy ion, un/biased, un/cooled, etc.), and what were the results? Does the observed, on-orbit rate of hot pixel emergence, or anomalous behavior, align with expectations from ground testing? I assume not, but am curious as to why. -The memory zone pixels are ∼half the area of the imaging pixels. Are they "hit" half as often, or is it impossible to tell? -Will a version of these ACCDs fly on ATLID/EarthCARE? Have the observations/findings in this paper inform the design, testing, or con-ops of ATLID? Will similar mitigation strategies as herein need to be employed for ATLID? Referring to Section 4.1: -Which space weather variables were considered for correlation with the rate of damage/hot pixel emergence? (Line 512) -Can damage events be geolocated, like was done for the transient events in Section 4.3 (Fig. 18)? This might be helpful to show. -Did damage occur more frequently on the day/nightside of the orbit? If no correlation with the poles or SAA is observed, this might be suggestive of damage by untrapped particles, either energetic solar protons or galactic cosmic rays (GCRs). A day/night difference might be suggestive of a spacecraft charging connection. An anti-correlation of rate of hot pixel accumulation with solar activity, with a lag of a few months, might suggest a GCR connection. Data from the Alpha Magnetic Spectrometer on ISS might also be a good resource for GCR/high energy flux on-orbit.

Absence of correlation with these variables might be worth mentioning to the reader if already considered. Referring to Section 4.3: -Is there evidence for radiation-induced light emission (e.g., fluorescence, phosphorescence, Cherenkov, electroluminescence) originating from the ACCD cover glass, or other upstream optics/surfaces? This may be an explaining mechanism for the ∼50% of transients that were observed to affect more than one pixel simultaneously, assuming the pixels were clustered. -Were any transients clustered? -Can the timescale of the transients be resolved, or do they appear in exactly one range bin? If radiation-induced light emission has been ruled out by the author, some discussion of that fact may still benefit the reader. -Is there evidence for latent damage? That is, do any pixels begin to exhibit damage hours, days, or even weeks after they experience an initial transient?

---

## Author Comment (AC1) · 12 Apr 2021

**Response to Referee Comment #1 on**

*Characterization of dark current signal measurements of the ACCDs used on-board the Aeolus satellite*

The authors thank reviewer #1 for carefully reading the paper and providing very useful comments. In the following, referee comments are repeated in green and answers by the authors are provided directly below in black.

**General comments:**

The manuscript is dedicated to the improvement of experimental data coming from the ALADIN lidar onboard the Aeolus satellite, which provides continuous measurements of atmospheric winds, aerosols, and clouds. The manuscript seeks to solve an important problem of identifying and fixing the experimental issue associated with the so-called "hot pixels" of the ALADIN's ACCD detectors. The authors suggest a method for pinpointing these pixels, introduce dedicated calibration modes, correct the signals from the affected pixels, and show the results of wind retrievals from the fixed signals. The problems of this kind are not new to the experimental physics, but in this case the study was hindered by the fact that the experimental setup was not available for direct testing in the lab. Still, the authors show that it provided sufficient amount of information for performing the analysis and for fixing the problem. In general, the real state of the atmosphere and the retrieved atmospheric data are linked through a number of conversions and convolutions, each of which can affect the quality of the retrieved parameters. In the case under consideration, the biggest challenge was associated with the missing or damaged pieces of information, needed for the retrieval, namely, with pixels providing the profiles yielded from fringe-imaging or double-edged techniques in some ACCD rows.

Since the backscattered photons carrying the information about the atmospheric properties in this setup are stored in ACCD matrix, one can split the solution of the problem to several steps: (i) identifying the pixels, which are not reliable; (ii) correcting or excluding these pixels from the retrieval; (iii) depending on the previous choice, one has either use the fixed values or modify the retrieval algorithm; (iv) since the initial retrieval algorithm did not take into account the possibility of hot pixels spoiling the inputs, one has a right to impose physical constraints on the retrieved data to fix the affected points. The authors did an excellent job for (i) and then they followed the correction scheme of (ii) and ended up with (iii). From this point of view, the work is impeccable. Still, I'd suggest to consider a bigger picture and to look at the problem at a different angle. Perhaps, the authors did it in the background and found that it didn't solve the problem, but I found no trace of it in the manuscript, so at least this is worth a discussion.

Let me explain. Looking at Fig. 1 of the manuscript and comparing it with the Fig.11, one can see that the experimental setup has a certain redundancy in a sense that the peak in the Mie signal almost never is narrower than 3 bins and, in some cases, a naked eye distinguishes 4 bins filled with non-zero signal. At the same time, Fig. 11 tells us that the situations when two adjacent horizontal pixels are "hot" are rare. Knowing that this detector is characterized by a low noise, one can make use of the remaining available information and still get a reliable result. To prove this, I performed a simple test illustrated in the attached Figure. The panel (a) shows the Mie detector mask, which is consistent with Fig. 11 of the manuscript, but converted to a binary (good/bad) form. Panel (b) shows the simulated signal, which qualitatively resembles that of Fig. 1 of the manuscript, but which passes through the hot pixels of the mask (a) for demonstration purposes. For each row, the exact position of the peak

corresponding to exact value of the wind is stored for reference. Panel (c) shows the same signal with hot pixels masked out. The Poissonian noise was added to the pixel values to imitate the detector's behavior. Then the fitting procedure based on sliding profile correlation approach similar to those used in [Goldberg et al., 2012] and [Feofilov and Stubenrauch, 2019] was applied both to a full set of input data and to a masked one. The procedure uses the knowledge of the profile of the fringe-imaged signal along the columns and this profile is supposed to be known with high accuracy. The resulting retrieval uncertainties are shown in panel (d) of the Figure. As one can see, the position of the peak retrieved from incomplete data does not change that much compared to the retrievals from the unmodified datasets, and the uncertainty in pixels converted to wind speed uncertainty is of the order of 0.03 m/s. This is just a rapid exercise, which should be done in a different way for Rayleigh signals, but it leads to an important question – even though the fixed hot pixels provide a dataset compatible with the rest of the processing chain, wouldn't it be easier and safer to exclude them from the consideration and to update the procedure? I understand that this is not what the manuscript is about, but it's a major philosophic question whether one should use fixed values from a damaged detector or use a reduced dataset profiting from the redundancy of the data. The latter approach does not diminish the significance of the work, but if it proves to give more reliable data through a simpler procedure, it should be considered.

The second question is about aforementioned step (iv) – I believe, the retrieval procedure could profit from the physical constraints of the following kinds: (a) point-to-point wind speed change and (b) point-to-point aerosol/cloud properties change. Both are easy to justify and both can serve as an additional quality control mechanism at early stage – if sudden unphysical jumps are found, the pixels are removed from the retrieval and the values are interpolated, masked, and so on.

**Response to General Comments:**

Thank you very much for your valuable feedback. It is highly appreciated that you carried out a simulation study to test different approaches to tackle the hot pixel issue by omitting the hot pixels in the wind retrieval.

First of all, it is important to mention that Figure 1 of the manuscript is only a simplified sketch of the ACCD to illustrate its working principle. The figure does not consider the full Mie fringe shape which is spread over all 16 ACCD pixels – even at the edges of the ACCD the values do not approach zero. For simulation studies, this could be approximated with an Airy function. On top of that, also the spectrally broad bandwidth Rayleigh signal and the solar background are part of the Mie signal. I think these aspects have to be considered in your simulation study presented in Figure 1. Apart from that, it is not clear which values for the atmospheric signal levels and hot pixel amplitudes were used in your simulation. Typical values for the Mie channel are 15 LSB and 5 LSB (both measurement level) for the atmospheric signal level and the hot pixel offset, respectively.

There are several reasons not to exclude hot pixels from the wind retrieval. An important aspect is the limited number of available pixels on the ACCD. As illustrated in Figure 1 of the manuscript, the ACCD has only 16 columns which makes each single pixel indispensable for the retrieval. As already mentioned above, each pixel contains valuable information. Although the major part of the Mie fringe is only contained in three to four pixels in the center of the ACCD, the pixels at the edge still contain valuable information which is needed to correctly derive the fringe centroid position, the fringe width and the broad bandwidth Rayleigh offset, used for the computation of the SNR and scattering ratio.

Moreover, hot pixels are not damaged and thus, still provide valuable measurement signals which can be used in the wind retrieval. Their main characteristics are increased dark current signals that are changing over time. And as introduced in Sec. 2.3 of the manuscript, DUDE measurements allow for a pixel-wise determination of the dark current signals for the correction of the increased dark signal values.

As the omission of hot pixels is not feasible, the approach of correcting hot pixels using DUDE measurements is the most suitable solution in the Aeolus. On the one hand, this method was straightforward to implement for both channels without having to redesign the well-established wind retrieval algorithms. On the other side, this method is also capable of dealing with the steadily increasing number of hot pixels without having to adjust the algorithm after each hot pixel occurrence. This is also a very important aspect for an operational satellite mission like Aeolus. In addition to that, this method is also compatible with the Aeolus L2A aerosol retrieval algorithms.

"The second question is about aforementioned step (iv) – I believe, the retrieval procedure could profit from the physical constraints of the following kinds: (a) point-to-point wind speed change and (b) point-to-point aerosol/cloud properties change. Both are easy to justify and both can serve as an additional quality control mechanism at early stage – if sudden unphysical jumps are found, the pixels are removed from the retrieval and the values are interpolated, masked, and so on."

First of all, it's important to know that the Aeolus processing chain is strictly sequential due to near-real-time (NRT) requirement for Aeolus data products and does not contain feedback loops which means that at the L1B processing stage the L2B products are not yet available. For the NRT processing there is also the requirement to provide the L2B data products only 30 minutes after the downlink of the raw data. Since the dark signal correction is part of the L1B processor, we cannot use additional information about point-to-point wind changes at this processing stage. For the future, it is planned to implement an additional check into the L2B processor to detect hot pixel steps in the L2B winds-based comparisons with the ECMWF model background (see Sec. 3.3 of the manuscript). As demonstrated in Sec. 3.3 of the manuscript, some hot pixels induce large O-B deviations in the wind products. So, this check aims at detecting hot pixel induced bias steps which occur in-between two DUDE measurements by analyzing the wind speed difference w.r.t to the ECMWF model. Winds exceeding a certain threshold will be flagged as invalid at range bin level.

The prerequisite of masking hot pixels in the processing is the detection of hot pixel induced signal steps while the instrument is in measurement mode which is very challenging. In simple terms, the atmospheric return signal measured in the memory zone of the ACCD during wind mode is a composition of atmospheric and dark current signal. To properly detect hot pixel induced steps one must be able to distinguish between atmospheric and dark current signal induced steps. For this, the ratio between the atmospheric and the dark signal intensities is important. Thus, for the Rayleigh channel, the location of the pixel on the ACCD is the key factor. For instance, for pixels which are fully covered by the Rayleigh spots, i.e. Rayleigh column positions 1-5 and 9-13, the atmospheric signal is usually too dominant and variable compared to the dark current signal to distinguish between atmospheric and dark signal induced steps in the atmospheric return signal. Figure 1 below shall help to better understand this problem. It shows signal intensities (red line) for Rayleigh hot pixel [20, 2] which is covered by one of the Rayleigh spots at observation level. In order to minimize effects induced by changes in the range bin settings or the altitude of the range bin, the signal is median-filtered using a window size of 400 observations. The vertical dashed green lines in the plot indicate the locations of

the DUDE measurements and the blue line with the second y-axis shows the dark current signal correction value applied to the signal in LSB. The shaded areas which span certain time periods indicate the validity times of the different orbits during the day. The DUDE measurements at 05:45 UTC and 16:15 UTC indicate two jumps of about 2 LSB in the dark current signal on that day. Analyzing the atmospheric return signal (red line) shows that these jumps are not visible here.

[Figure]

*Figure 1: Median filtered (window size: 400 obs.) ALD_U_N_1A pixel intensities for Rayleigh hot pixel [20, 2] (red) on 2020-03-03 together with the DCMZ correction value in blue. The vertical green bars indicate the times of the DUDE measurements.*

A general discussion of potential hot pixel correction methods was added to Sec 3.3 of the manuscript. It is made clear that the omission of hot pixels is not feasible and advantages of the implemented correction scheme are clarified:

Same as for the Mie channel the magnitude of the dark current signal-induced Rayleigh bias depends on the signal level of the backscatter signal. Generally speaking, the dark current induced bias is more constant in the Rayleigh than in the Mie channel due to the more constant Rayleigh signal compared to the strongly varying Mie signal from clouds and aerosols. For the correction of hot pixels in the Aeolus NRT processing several options were considered. One way would be to omit hot pixels from the wind retrieval. However, this approach is not feasible due to several reasons. First of all, the very low number of 16 pixels per column makes each pixel indispensable in the wind retrieval. It is important to note that also the pixels at the edge of the ACCD contain valuable information necessary to retrieve the wind information. In addition, hot pixels are not damaged and still contain information that can be used in the wind retrieval Another correction method is the interpolation of hot pixels using the information from neighboring pixels. Considering the rather coarse vertical resolution of Aeolus measurement of 250 m up to 2000 m and the non-linear vertical wind shears in the troposphere, especially vertical interpolation could be highly erroneous depending on the vertical wind shear and the range bin settings.

As a result, it was decided to implement an algorithm which corrects the increased dark current signal offsets of hot pixels based on frequent dark current signal characterization measurement. This correction scheme was successfully implemented into the wind retrieval of the Aeolus operational processing chain on the 14th of June 2019.

In detail, the dark signal characterization obtained from frequently performed DUDE measurements is used for a pixel-wise dark signal correction, i.e. a DSNU correction, of adjacent wind measurement signals. As this kind of correction was not foreseen before launch to be performed on a regular basis, dedicated instrument modes and algorithms had to be developed after launch. The implemented correction approach has the advantage of being applicable to both channels without having to redesign the well-established wind retrieval algorithms. Moreover, this method is capable of dealing with the steadily increasing number of hot pixels without having to adjust the algorithm after each hot pixel occurrence and is also compatible with Aeolus L2A retrieval algorithms.

Moreover, the following paragraph of Sec 3.3 was modified to make clear that with the current processor configuration point-to-point wind speed changes cannot be used in the dark signal correction of the L1b processor:

This example demonstrates that even with DUDE measurements performed at high frequency, a perfect dark signal correction is not possible. It is also clear that the performance of the correction scheme depends on the behavior of the hot pixel. In case of RTS-like characteristics as shown for Mie pixel [13, 9] (see Fig. 5) the correction performs poor compared to hot pixels with sporadic shifts. Nevertheless, this approach works fine to remove the constant proportion of the dark current offset and in any case reduces periods of enhanced dark current induced bias sufficiently. In order to further mitigate hot pixel induced effects, a further check will be implemented for the Aeolus level 2B product in the future. This check is based on comparing the Aeolus with ECMWF model winds for each range gate. In case, the difference between both exceeds a certain threshold, the Aeolus winds of the affected range gate will be flagged invalid. For example, the period between 14:15 UTC and 20:30 UTC (see Fig. 10) would be flagged as invalid by the check. It is important to mention that the Aeolus processing chain is strictly sequential and does not contain feedback loops which means that at the L1B processing stage the model comparisons from the L2B products are not yet available. For the NRT processing there is also the requirement to provide the L2B data products only 30 minutes after the downlink of the raw data. Since the dark signal correction is part of the L1B processor, it is not possible to use this information to mask or flag hot pixel offsets in the L1B processing stage.

**Specific comments:**

*Lines 200-209: if CIC noise is important, how does this fact match the "low-noise detector" statements above? Some numbers are needed here, so that the reader could make his/her own conclusions.*

The paragraph you are referring to is related to the role of CIC in the generation of hot pixels. Thus, it is about CIC increasing the mean dark current signal rather than dark current signal noise. However, information about the dark current signal noise and read-out noise is added to Table 1 of the manuscript. Considering the pulse repetition frequency of 50.5 Hz and 18 pulses per measurements, the in-orbit dark current signal rates are 0.49 LSB/s and 0.24 LSB/s for the Mie and Rayleigh channel. Considering the Poisson distribution of the dark charges these values correspond to 0.78 e- – 0.89 e- rms dark current noise for a residence time of the signals in the ACCD of one measurement which is 0.4 s. Note that these numbers already include CIC. The numbers for the read-out noise of the detector are between 4 e- and 6 e- rms.

These values were added to Table 1 of the manuscript and the text of Sec. 2.1 was modified accordingly:

Table 1: Specifications and in-orbit performance of the Aeolus ACCDs

| Parameters | Value |
|---|---|
| Type | Thinned backside illuminated accumulation Si-CCD |
| Area | Imaging zone: 0.43 mm x 0.43 mm – 16 x 16 pixels
 Memory zone: 0.43 mm x and 0.75 mm - 32 x 25 pixels |
| Pixel size | Imaging zone: 27 µm x 27 µm
 Memory zone: 13.5 µm x 30 µm |
| Operating temperature | -30 C° |
| Temporal resolution | 2.1 µs – 16.8 µs / 250 m – 2000 m for atmospheric layers (#1 - #24)
 3750 µs for solar background (layer #25) |
| Quantum efficiency | 0.85 |
| Charge transfer efficiency | 0.9999 |
| Radiometric gain | Mie: 0.68 LSB/e-, Rayleigh: 0.44 LSB/e- |
| Dark current signal rate | Mie: 0.49 LSB/s, Rayleigh: 0.24 LSB/s (in-orbit values) |
| Dark current signal noise | 0.78 e- – 0.89 e- rms (root-mean-square) (in-orbit values) |
| Read out noise | 4 e- – 6 e- rms |

*Line 278: can one prove this statement about the DUDE correction with some formula or reference? At the moment, there are only qualitative statements here.*

The following example shall depict the dependency of the DUDE correction on the characteristics of the hot pixels. As shown in the manuscript, we carry out DUDE calibrations four times per day. In case, there are hot pixel induced shifts in-between the DUDE measurements, the dark current of the affected pixels is not properly corrected until a new DUDE measurement is performed. As illustrated in Sec. 4.2 of the manuscript, the characteristics, i.e. the fluctuation rate and level amplitudes, are very different for the hot pixels. So, for stable hot pixels which do not change their level often, the frequency of the DUDE measurements is not very critical. But for hot pixels with high fluctuation rates, it is much more likely to have dark signal induced steps in-between the DUDE measurements which leads to biased wind results until the next DUDE calibration is carried out. This effect is also illustrated in Figure 10 of the manuscript which shows the signal intensities of the very jumpy Mie pixel [13, 9] on 2019-11-14. It shows a dark current induced signal decrease of 8.0 LSB at 14:15 UTC. As a result, the dark current signal is overestimated as the dark current calibration based on the DUDE measurement from 13:15 UTC is still active. Consequently, the winds of range bin 13 are biased for the period between 14:15 UTC and the next DUDE update at 20:45 UTC (also explained in Sec. 3.3 of the manuscript).

*Lines 300-320: perhaps, it's a matter of preferences, but how does this approach compare to a simple 3-sigma test? Another approach, which could be also useful for detecting hot pixels as well as identifying the nature of the noise is building and analysing Fourier spectra of the temporal sequences for each pixel. Most probably, the spectra of hot pixels will be different from those of "normal" ones and hot pixels of a different nature will reveal this in the spectra, too.*

For our analysis an approach is needed which is not only capable of differentiating between normal and hot pixel behaviour but also can find the exact temporal index of the dark current signal shifts. This information is needed to derive information about the hot pixel signal amplitudes and the time spent at the different dark signal levels. That's the basis of the categorization of the hot pixels as shown in Sec. 4.1-4.2 of the manuscript and, especially, of the analysis of the RTS characteristics (Sec. 4.2.1).

A simple 3-sigma-test would have probably been suitable to detect hot pixels but it would not have been possible to derive further information about the hot pixel characteristics (amplitudes and time spent at a dark signal level) as mentioned above. Figure 2 down below shows the dark signals of Mie pixel [20, 2] (same as Figure 15 (top) of the manuscript) at observation level together with thresholds obtained from the mean value $\pm$ 3* standard deviation of the dark signal. It shows that the 3-sigma threshold is exceeded multiple times which would result in classifying this pixel as "hot". However, comparing Figure 2 with Figure 15 (top) of the manuscript clearly indicates the advantages of the sophisticated approach presented in the manuscript, i.e. the detection of the dark signal segments and the derivation of the dark signal levels.

I agree that an approach based on the analysis of Fourier spectra of the dark signals would also have helped to learn something about the different natures of the hot pixels. However, as mentioned above, this approach would not have provided the needed information about the indices of the switches between the dark current signal levels which is the prerequisite to derive temporal RTS characteristics (see lines 349-353 of the manuscript).

[Figure]

*Figure 2: Dark signals of Mie pixel [20, 2] at observation level.*

To clarify the need of the proposed time series segmentation algorithm, the introduction of Sec. 3.1 was changed as follows:

**3.1 Detection of permanent dark current anomalies**

295

The motivation for a detailed characterization of permanent dark current anomalies is twofold: a) it supports investigations for the underlying root causes of the hot pixel issue and b) the number and magnitude of dark signal shifts define the impact on the wind observations and the DUDE correction. In order to fulfill b) a more sophisticated algorithm such as a simple 3-sigma test, which can differentiate between normal and hot pixels, is necessary. The presented algorithm is also capable of detecting

300 temporal indices of the sudden shifts in the dark signal time series which is the basis of the categorization of the permanent dark current anomalies.

*Line 338: again, spurious changes could have been filtered out by Fourier smoothing procedure.*

In our case Gaussian Kernel Density Estimation (KDE) is used to estimate the probability distribution of the dark signal values. As depicted in the Sec. 3.1, the key parameter of the KDE is the bandwidth parameter which controls the size of the Gaussian kernel at each data point. A bandwidth value which is too high leads to an over-smoothed density estimation which probably hides important structure. On the contrary, a too narrow bandwidth will put too much emphasis on single points and thus, result in a density estimation curve with two many modes.  As a result, it's quite important to find suitable values for the bandwidth. There exist several algorithms for this task. The standard approaches are Scott's rule of thumb (Scott, 1992) or Silverman's rule of thumb (Silverman, 1986). However, these are simplified approaches which assume normally distributed data. In the presence of dark signal anomalies, the underlying distribution function is not known. This is why, non-parametric approaches are needed to estimate the optimum kernel bandwidth. One established non-parametric method is Maximum-Likelihood Cross Validation (Duin, 1976) which is a purely data-driven method to derive the bandwidth parameter. Here, a metric for different values of the bandwidth is computed by estimating the kernel function on a subset of data and computing and evaluating this function on the rest of the data. The advantage of this method is that it's purely data-driven, meaning that no assumptions on the underlying data is needed. In our case this method provided reliable results. The drawback of this method is its high computational costs. However, for the application to one-dimensional data that is not a big issue. If this would have been a problem, it could have been an option to implement a fast Fourier-based Kernel density bandwidth estimation as depicted in  (Gramacki and Gramacki, 2017).

To further motivate the selection of the KDE bandwidth selection method, the following was added to Sec. 3.1 of the manuscript:

On the contrary, important features may be smoothed away when applying excessive smoothing. There exist several algorithms

350 for this task. For the purpose of analyzing dark signal anomalies, a non-parametric method which does not require any assumptions of the underlying data distribution is needed to find the optimal bandwidth parameter. ThusHere, maximum-likelihood cross-validation is used to determine the bandwidth parameter which is an established method for the objective, data-based derivation of the bandwidth parameter (Jones et al., 1996). This method computes a metric for different values of the bandwidth by estimating the kernel function on a subset of data and computing and evaluating this function on the rest of

355 the data. The advantage of this method is that it's purely data-driven, meaning that no assumptions on the underlying data is needed.

*Lines 400-410: see the general comments – perhaps, the discussion should be updated.*

See our response to your general comments.

*Lines 430-435: how does this correction compare to vertical interpolation?*

So far, vertical interpolation has not yet been considered as valuable approach to correct for hot pixels. Considering the rather coarse vertical resolution of Aeolus measurement of 250 m up to 2000 m and the non-linear vertical wind shears in the troposphere, vertical interpolation could be highly erroneous depending on the vertical wind shear and the range bin settings. In case, Aeolus would be able to provide measurements with a better vertical resolution, vertical interpolation might indeed be an option.

As part of the modifications of Sec. 3.3, the mitigation approach of vertical interpolation is discussed (see screenshot under "General Comments").

*Line 439: a median correction is applied, which does not eliminate sporadic events. Even though it smooths them out, their erroneous nature is included in the results. On the other hand, gradient-based or Fourier filtering would have removed a non-physical part of the signal.*

You are correct signal spikes such as dark signal introduced transient events (introduced in Sec. 3.2 of the manuscript) can influence the result of the median filtering. In this case, the correct way to perform this kind of analysis would have been to identify transient events at measurement level and remove them before averaging the measurements to observations. However, the analysis has shown that on average only 0.24 % of the measurements are affected by transient events (see Sec. 4.3 of the manuscript). This is why, this effect is considered to be negligible.

*Line 500: it would be interesting to recalculate these 6% into a weighted percentage of pixels used in retrievals. For example, pixel [9,13] is used often whereas [1,1] is not.*

As explained in the response to your general comments, all pixels of both channels are used in the wind retrieval. As a result, it is not possible to derive a weighted average of the pixels used in the retrieval.

*Line 550: Linear trend is interesting here. If the damage is due to high energy particles hitting the ACCD then the slope should change with time, but 6% is too small a number for this to be noticed.*

Yes, this is correct. In line 515 of the manuscript it is mentioned that the solar activity is currently at a minimum and it will be interesting to see if this has an influence on the rate of hot pixel generation. Moreover, it is planned to redo this kind of analysis at the end of the mission when a larger dataset of hot pixels is available (see responses to reviewer #2).

*Line 689: cosmic particles partially penetrate the atmosphere, so this is not a 100% proof.*

This is correct. This argument will be reformulated as follows:

715   The fact that two hot pixels Mie [16, 15] and Mie [24, 3] – both of them in different hot pixel categories and with similar characteristics of hot pixels emerged in-orbit – were already present before launch supports the hypothesis of an origin which is not solely related to the fact that Aeolus is operated in space environment with very harsh radiation conditions . However, other radiation sources within the instrument or even within the ACCD package

*Lines 701-702: first, we did not see this in the manuscript and second, it should be considered in the light of the exercise demonstrated in General comments.*

The intention was to make this point clear in Sec. 3.3 of the manuscript. In this section, a simple calculation is provided to demonstrate the effects on the Rayleigh wind results (see lines 410-418 of the manuscript). It was demonstrated that a hot pixel induced signal elevation of only 10 LSB already results in an HLOS error of already about 2.6 m/s. In order to further clarify the correlation between the wind error and hot pixel offsets, the atmospheric return signal of Rayleigh hot pixel [11, 2] will be added to Figure 8 of the manuscript (see Figure Figure 3). The modified figure clearly demonstrates the correlation of the increased O-B bias values around 400 hPa and the dark signal induced steps in the atmospheric return signal.

[Figure]

*Figure 3: (Top): Comparison between Aeolus L2B Rayleigh-clear HLOS winds and the ECMWF model equivalents between 2018-10-06 and 2019-10-31. The plot shows the mean difference between the observation (O) and the background (B) (short-range forecast) model field as a function of pressure and time. (Bottom): Median filtered (window size of 400 observations) signal intensities of Rayleigh hot pixel [11, 2] during wind measurement mode.*

Figure 8 of the manuscript was changed as shown in Figure 3 above. Moreover, the explanation of this figure was updated in Sec. 3.3 of the manuscript.

*Line 752: numbers are missing here: uncertainty/bias after the correction vs uncertainty/bias before the correction.*

As shown in the manuscript, the hot pixel induced bias mainly depends on the hot pixel characteristics. In Sec. 3.3 of the manuscript, the effect is demonstrated on the basis of a simplified example for the Rayleigh channel. Here, a realistic hot pixel dark current signal offset of 10 LSB present for Rayleigh spot A is assumed. In case, no dark signal correction is applied this would result in a wind bias of 2.6 m/s HLOS. Also, Figure 3 (Figure 8 of the manuscript) depicts hot pixel induced wind bias values of several m/s. The random error of the wind measurements is not affected by the hot pixel correction.

As mentioned in Sec. 2.2 of the manuscript, the noise characteristics and thus, the random error, are mainly driven by the shot noise of the signal.

785   A combination of dedicated instrument calibration modes and ground processors were developed to allow for a pixel-wise dark signal correction of the wind signals already shortly after launch. It was demonstrated that this correction is capable of correcting for the dark signal non-uniformity arising from hot pixels on the ACCD and thus, successfully removes hot pixel induced wind bias of up to several m/s. It is expected that this correction will work throughout the whole mission lifetime no

*Lines 301, 342, and elsewhere – in some PDF viewers, the font used for Python module names looks strange.*

The Python module names in the manuscript will be changed to normal font.

---

## Author Comment (AC2) · 12 Apr 2021

**Response to Referee Comment #2 on**

*Characterization of dark current signal measurements of the ACCDs used on-board the Aeolus satellite*

The authors thank reviewer #2 for carefully reading the paper and providing valuable input. On the one side, your seed questions support the on-going root cause analysis of the Aeolus hot pixel issue and other side, they are also very useful to further improve the quality of the manuscript and provide the impetus for a potential follow-on paper focused on root-cause analysis. In the following, referee comments are repeated in green and answers by the authors are provided directly below in black.

**General comments:**

The focus of this paper is on analysing the on-orbit hot pixel characteristics and emergence trends in the novel ACCD launched on the space-based wind lidar ADM-Aeolus, and mitigation of hot pixel effects on wind retrieval accuracy. Though the paper does not draw any firm conclusions about the potential root cause(s) of hot pixel emergence, this paper nicely sets the stage for such a discussion. Most of my comments are geared towards this discussion. I should mention that, in my opinion, a discussion of the root cause(s)/damage mechanism(s) is optional, as the authors' description of the strategies for mitigating the impact of hot pixels on wind retrievals, and detailed characterization of these anomalies, make this a valuable work in its own right. In fact, the author could consider de-scoping some of the discussion on the root cause from this paper, and deferring it to a future work, if the author so wishes. A more detailed discussion of the root cause might be beyond scope, but I offer the following comments/questions to address (optionally) that might aid a future publication/study on the issue, or satisfy a curious reader of this paper. General questions: -

How much shielding exists around the ACCDs on Aeolus, and/or what is the shielded radiation environment/dose (yearly DDD, TID)?

In the framework of the Aeolus development, simulations have been performed to determine the shielding for the six instrument faces ($\pm X$, $\pm Y$, $\pm Z$) as seen by the detector. Equivalent shielding figures from 2 mm to 8 mm per face have been found for the most exposed ACCD, plus the 2.5 mm thickness BK7 window. The TID and TNID levels are respectively about 0.3 krad(Si)/year and 5E6 MeV/g(Si)/year for 400 km circular orbit, maximum solar activity being considered for the whole mission duration. Please notice that Aeolus altitude has been decreased to 320 km, reducing even more the radiation levels.

Has there been a detectable, steady trend/increase in the dark current observed over the course of the mission for pixels that have not experienced an anomaly?

No, there has not been an observable increase of the mean dark current signal for ACCD pixels that were not classified as hot pixels. Figure 1 below indicates the dark signals at observation level of an ACCD pixel (Rayleigh pixel [15,13]) which did not exhibit an anomaly. This plot does not show an increase of the dark current signal. Another example of a hot pixel time series with nominal behaviour is shown in Figure 4 (top) of the manuscript. In this case also no increase of the dark current signal could be observed.

To make this clear in the manuscript. The following sentence was added to Sec. 4.2 of the manuscript:

585 **4.2 Hot pixel signal levels**

Figure 13 shows the median dark signal value of the Mie and Rayleigh hot pixels in ascending order of their dark current level. In order to show the spread of the dark signal values, the scaled MAD is indicated by the black error bars. Given that the dark signal values of pixels that show nominal behavior are Gaussian distributed (see Fig. 4), it might seem reasonable to use a hot pixel threshold based on the standard deviation and the mean. Thus, the dashed black lines in Fig. 13 indicate the median value

590 + 3* scaled MAD of dark signal values obtained from all ACCD pixels after removing hot pixels which is 2.28 LSB and 1.54 LSB for the Mie and Rayleigh channel, respectively. It should be noted that no increase of the dark current of pixels which were not categorized as hot pixels over the mission lifetime was observed. Due to the fact that many Aeolus hot pixels only show very small shifts in the mean dark signal and even return to a normal dark signal after some time (see Sect. 4.2.2), many hot pixels would have been undetected using this simple threshold technique. This points out the necessity to use the

595 sophisticated detection algorithm as introduced in Sect. 3.1.

[Figure]

*Figure 1: Dark signals of Rayleigh pixel [15, 13] with nominal dark signal behaviour. The blue dots indicate dark signal intensities at observation level. The solid blue indicates the median filtered signal (window size: 1000 observations).*

What design deltas between the ACCD and previously flown CCDs (e.g., Hubble) might explain the observed anomalies? Inversely, what design elements do the ACCDs share with the CCD detectors of GOMOS on ENVISAT?

Please find below a summary of the most important characteristics of the Hubble CCD43 and the GOMOS CCD26:

Hubble (WFC3-UV): Inverted mode operation (IMO), back-illuminated, 2048 x 4096 pixels (15 μm x 15 μm pixel size) multi-pin-mode-operation (Windhorst et al., 2011)

GOMOS CCD26: IMO, back-illuminated, 143 x 1353 pixels (20 μm x 27 μm pixel size) (ESA, 2000)

The build of these devices in terms of silicon resistivity, dielectric thickness and doping levels is very similar to that used for the Aeolus detectors. Also, the channel doping is probably similar. But the Hubble CCD43 would have been IMO rather than Advanced-IMO (AIMO) with the barrier implant under the whole of the poly 3 electrode rather than just under one edge. As a consequence, the dose of the barrier implant is likely to have been lower so the potentials in the silicon and the numbers of holes at the surface under a low clock could be slightly different. This may have an impact on CIC generation.

However, the major difference between the Aeolus CCDs and anything previously designed or built by T-e2v is the memory section. This is almost unique in that the clock phases are cycled a large number of times with the surface going into pinning but without the charge being transferred. Any local

generation site for CIC generation will therefore be able to give a hot pixel rather than distributing the charge over a complete column.

What radiation testing was conducted on the ACCDs prior to launch (proton energies and fluence steps, TID dose steps, heavy ion, un/biased, un/cooled, etc.), and what were the results? Does the observed, on-orbit rate of hot pixel emergence, or anomalous behaviour, align with expectations from ground testing? I assume not, but am curious as to why.

In the framework of the Aeolus ACCDs development, proton tests have been performed to evaluate the probability of occurrence of such hot pixels and RTS pixels at an operating temperature of -30 °C (also mentioned in Sec. 2.2 of the manuscript). On-ground proton tests were performed at different temperatures between -30°C and 20°C in 2004 and fluence levels. Two samples were irradiated with 30 MeV protons (fluence: 2E9 protons/cm² and 1.35E9 protons/cm²) and two other samples were irradiated with 100 MeV protons (fluence: 4.2E9 protons/cm² and 2.7E9 protons/cm²). A significant increase in dark signal was observed at the maximum dose (~10x the beginning-of-life value for Aeolus. Despite the much higher radiation dose as in space only three anomalies were observed: one post-irradiation RTS pixel in one device + two suspicious pixels with increased dark current signals for another detector sample. It should however be noticed that the dark signal acquisition duration has not been optimized to track low frequency variations of the dark current signals (only 512 frames have been acquired continuously) and the operation mode with regard to the timing settings during the tests was not fully comparable with the settings used in-orbit. In addition, the post-processing algorithm sensitivity was not good enough to detect abnormal pixels amplitudes as low as observed in-orbit. Overall, the results show one post-irradiation RTS pixel in one device and two suspicious pixels with increased dark current signals observed for another detector sample.

A few details about the proton tests were added to Sec. 2.2 of the manuscript.

The memory zone pixels are ~half the area of the imaging pixels. Are they "hit" half as often, or is it impossible to tell?

It should be mentioned that each row of the memory zone of the ACCD consists of 16 transfer and 16 storage pixels. The 16 transfer pixels are the equivalent to the imaging zone and the form the transfer section of the memory zone. The storage pixels form the memory storage section in which the signal accumulation is performed. This is why the memory zone pixels are half of the area of the imaging pixels. However, for the dark signal generation the residence time of the signals in the imaging and the memory zone is more important than the size of the pixels (also explained in Sec. 2.1 of the manuscript). The residence time in the memory zone is with 0.4 s much longer compared to the residence in the imaging zone which is between 2.1 µs to 16.8 µs, depending on the range gate timing settings. Thus, the focus lies on hot pixels of the memory zone.

As mentioned in Sec. 2.3 of the manuscript, there are two specific measurement procedures to characterize the dark current in the imaging and memory zones of the ACCD. Both procedures were defined to be performed while the laser is not operating (e.g. before the switch-on). To overcome this problem and measure the dark current of the memory zone during continuous laser operation so-called "DUDE" measurements were introduced. This was possible by cleverly adjusting the range gate settings. However, mainly due to technical restrictions it is not possible to acquire dark signal measurements of the imaging zone while the laser is operating. As a result, the availability of imaging mode measurements is restricted to periods where the laser was in a lower measurement mode (e.g.

before the switch on). Thus, it is not possible to properly characterize the dark current signals of the imaging zone.

Section 2.3 of the manuscript was changed accordingly:

> After the first identification of hot pixels in the nominal Aeolus wind lidar measurements, a new procedure to allow dark signal characterization of the memory zone during continuous laser operation was introduced, so-called DUDE (Down Under Dark Experiment)-measurements. During DUDE measurements the range gate timing settings are adjusted such that the theoretical
> 245 return signal is acquired from below the Earth's surface. Figure 2 illustrates the difference in the data acquisition between wind (a) and DUDE (b) mode. In that way, dark current signals of all pixels of the memory zone can be measured without lidar signal contributions apart from the solar background signal. Due to technical limitations it is not possible to characterize the dark current of the imaging zone in the same way as for the memory zone. Thus, the availability of the imaging zone dark current measurements is restricted to periods where the laser is operated in a lower mode and not emitting laser pulses.

Will a version of these ACCDs fly on ATLID/EarthCARE? Have the observations/findings in this paper inform the design, testing, or con-ops of ATLID? Will similar mitigation strategies as herein need to be employed for ATLID?

The ATLID detectors were designed, tested by T-e2v and delivered to Airbus before the hot pixel issue on Aeolus was identified. There was therefore no possibility to influence the design of the CCD. During testing at T-e2v hot pixels associated with the flushing of the memory transfer register were identified and the proposed clock sequence for the flight instrument was modified to remove unnecessary flushing cycles. For the ATLD in-orbit operation, regular dark current calibration measurements will be carried out.

Referring to Section 4.1: Which space weather variables were considered for correlation with the rate of damage/hot pixel emergence? (Line 512)

As possible indicator for space weather, the information from www.spaceweatherlive.com has been checked. The "activation" of a hot pixel could not be correlated with the given scale of K-index, i.e., no threshold of activity could be identified.

Sec. 4.1 of the manuscript was changed as follows:

> The temporal evolution of the first appearance of the hot pixel anomaly (as listed in Table 2) is displayed in Fig. 12. It can be
> 550 seen that the increase of the hot pixel number with time is not perfectly linear. On the one side there seem to be periods where hot pixels occurred at a higher rate (e.g. 2019-01 to 2019-02) but on the other side there are also periods with very few anomalies (e.g. 2019-10 to 2020-01). However, no correlation between the hot pixel emergences and space weather activity (www.spaceweatherlive.com) was found. The "activation" of a hot pixel could not be correlated with the given scale of the K-index which is a measure of the disturbances of the horizontal component of the Earth's magnetic field, i.e., no threshold of
> 555 activity could be identified. The mean time difference between two anomalies is 14.68 days with a rather large standard

Can damage events be geolocated, like was done for the transient events in Section 4.3 (Fig. 18)? This might be helpful to show. Did damage occur more frequently on the day/nightside of the orbit? If no correlation with the poles or SAA is observed, this might be suggestive of damage by untrapped particles, either energetic solar protons or galactic cosmic rays (GCRs). A day/night difference might be suggestive of a spacecraft charging connection. An anti-correlation of rate of hot pixel accumulation with solar activity, with a lag of a few months, might suggest a GCR connection. Data from the Alpha Magnetic Spectrometer on ISS might also be a good resource for GCR/high energy flux on-orbit.

Absence of correlation with these variables might be worth mentioning to the reader if already considered.

For some hot pixels it is possible to identify the exact time stamp and geolocation of the hot pixel activation. This can be done by analysing Aeolus wind measurement signals (ALD_U_N_1A signals) for sudden hot pixel induced signal jumps. In the framework of root-cause analysis of the Aeolus hot pixel issue, we already performed this kind of analysis. First results gave a slight hint for an accumulation of activation events in the region of the SAA. However, due to the relatively low number of hot pixels and the resulting low statistical significance it was decided not to include this analysis into the manuscript. It might be better to redo this analysis again at the end of mission lifetime of Aeolus with more hot pixels. In this framework, also possible correlations with solar activity or data from Alpha Magnetic Spectrometer could be investigated in more detail.

Referring to Section 4.3: Is there evidence for radiation-induced light emission (e.g., fluorescence, phosphorescence, Cherenkov, electroluminescence) originating from the ACCD cover glass, or other upstream optics/surfaces? This may be an explaining mechanism for the ~50% of transients that were observed to affect more than one pixel simultaneously, assuming the pixels were clustered.

As stated in Sec. 4.3 of the manuscript, it is not surprising that transient events affect multiple pixels simultaneously as cosmic rays passing through the ACCDs are likely to hit more than one pixel. Figure 2 down below shows an example of one dark signal measurement obtained in the region of the SAA. The Rayleigh ACCD shows an interesting pattern with multiple transient hits across several range bins in the centre of the ACCD. However, except for the well-known beta/gamma emission from the $^{40}$K radioactive element part of the BK7 window, no other radiation effect coming from other instrument parts is known to the authors.

[Figure]

*Figure 2: A measurement of the dark signal of the Memory Zone obtained in the SAA with multiple transients observed at the same time.*

Were any transients clustered? Can the timescale of the transients be resolved, or do they appear in exactly one range bin? If radiation-induced light emission has been ruled out by the author, some discussion of that fact may still benefit the reader.

The timescale of transient events can be resolved as they occur in single Aeolus measurements (temporal granularity of 0.4 s). Note that the analysis of transient events in the manuscripts is also performed at measurement level. As mentioned above, it was observed that in many cases multiple pixels are affected at the same time. But not in all cases a clustering such as shown in Figure 2 could be observed. Temporal clustering of transients was only observed in the region of the SAA (see Figure 18 of the manuscript).

Is there evidence for latent damage? That is, do any pixels begin to exhibit damage hours, days, or even weeks after they experience an initial transient?

A detailed analysis to analyse the relationship between transient events and the occurrence events still needs to be performed. It might be worth to analyse accumulated number of transient events of a hot pixel before it became "hot" and compare this number to nominal pixels. In the discussion (Sec. 5) of the manuscript it is mentioned that the relationship between transients and the emergence of hot pixels is still unclear. This analysis could be performed for follow-on discussion paper about the root-causes of the Aeolus hot-pixel issue.

**References:**

ESA: ENVISAT GOMOS An Instrument for Global Atmospheric Ozone Monitoring, https://orfeo.kbr.be/bitstream/handle/internal/5299/Berteaux(2001a).pdf?sequence=1, 2000.

Windhorst, R. A., Cohen, S. H., Hathi, N. P., McCarthy, P. J., Ryan, J., Yan, H., Baldry, I. K., Driver, S. P., Frogel, J. A., Hill, D. T., Kelvin, L. S., Koekemoer, A. M., Mechtley, M., O'Connell, R. W., Robotham, A. S. G., Rutkowski, M. J., Seibert, M., Tuffs, R. J., Balick, B., Bond, H. E., Bushouse, H., Calzetti, D., Crockett, M., Disney, M. J., Dopita, M. A., Hall, D. N. B., Holtzman, J. A., Kaviraj, S., Kimble, R. A., MacKenty, J. W., Mutchler, M., Paresce, F., Saha, A., Silk, J. I., Trauger, J., Walker, A. R., Whitmore, B. C., and Young, E.: The Hubble Space Telescope Wide Field Camera 3 Early Release Science data: Panchromatic Faint Object Counts for 0.2-2 microns wavelength, https://doi.org/10.1088/0067-0049/193/2/27, 2011.

---

## Referee Report (RR1)

**General comments :**

This is a second review. In general, I'm satisfied with the answers and I think that the article can be published with minor corrections indicated below. However, I'd like to address the questions of the authors they posed in the reply regarding the test I made to show that the retrieval procedure could define the peak position on the ACCD with a sufficient accuracy even if the hot pixel information were not used at all.

The authors write "*The figure does not consider the full Mie fringe shape which is spread over all 16 ACCD pixels – even at the edges of the ACCD the values do not approach zero. For simulation studies, this could be approximated with an Airy function.*"
As for the fringe shape spread over all 16 ACCD pixels, the previous simulation used a Gaussian which was also spread over all pixels. The halfwidth of this Gaussian was selected in such a way that the shades of gray in Fig. 1b of (doi:10.5194/amt-2020-458-RC1) resembled those of Fig. 1 of the manuscript. For the second exercise shown below (Fig. 1), I took an Airy function and compared the results with Gaussian. One has to note that in general the sliding profile approach is not sensitive to the shape of the function as long as the function varies at the considered interval and its shape is known with high accuracy.

Then, the authors write that "*On top of that, also the spectrally broad bandwidth Rayleigh signal and the solar background are part of the Mie signal.*"
The broad features and constant offsets do not affect the peak position retrieval accuracy if the sliding profile approach is used.

The next comment is important, though: "*Apart from that, it is not clear which values for the atmospheric signal levels and hot pixel amplitudes were used in your simulation. Typical values for the Mie channel are 15 LSB and 5 LSB (both measurement level) for the atmospheric signal level and the hot pixel offset, respectively*"
Indeed, the discretization is important and in the previous exercise this was not taken into account properly, I apologize for overlooking it. If one assumes that the peak value of Mie detector is 15 LSB, then the absolute errors of peak position retrieval in units of ACCD detector row change from ~1.5e-3 to ~1e-1 that corresponds to ~2 m/s (Fig. 1). This is somewhat smaller than the random errors reported for Aeolus wind product, and the comparability of the profiles retrieved for a "healthy" detector with those retrieved from the simulations with hot pixels excluded tells us that the approach of skipping the hot pixels proposed in the first review is still valid. I do not require to include more discussion on this topic than what is already included in the present version of the manuscript (lines 441-444), it's just to draw an attention to this technique that works surprisingly well for different physical phenomena.

**Specific comments**

Lines 98-103: new text states that the read-out noise values were determined during pre-launch tests and then the discussion is based on this value measured on the ground. I understand that normally the noise of the amplifier should not change, but are there any estimates for an onboard read-out noise?

Lines 197: I would specify the actual numbers for the proton fluxes

Lines 199-200: what exactly was different and how this should affect the results?

Lines 301-303: "a more sophisticated" and "a simple 3-sigma" look strange in one sentence.

Lines 441-444: "each pixel indispensable in the wind retrieval". This is strange. Even though the authors use a different approach than the one used in my exercises, the information content of the whole row is still large, and the absence of a single pixel should not dramatically change the picture. One can carry out a simple "Gedankenexperiment", which usually is proven in real life with the real data. Imagine that a human eye observes a continuous function, or a continuous shape, or a graphic pattern. Next, let's imagine that a small patch covering 5% of the image is applied. No doubt that the brain will recover the original shape from this picture, especially if the shape is known. Of course, the bigger the patch, the poorer the accuracy, but for the objects like those shown in Fig. 1 of the manuscript a loss of up to 10% of information should not be critical. If so, the programming algorithm should exist which mimics the brain's algorithms and recovers the shape. The realization depends on the task, but the general idea should be clear. I believe that it can be realized in the framework of the authors' method, too. In any case, since the "hot" pixels are not completely erroneous, the authors' approach works, and this itself is a good achievement.

Lines 497-501: In general, the operational retrieval should not exclude the post-processing, during which the data quality might be improved. At this stage, one can impose different physical constraints and analyze/correct the retrieved data using ancillary information. Even in the current setup we observed an evolution of the products associated with the corrections introduced after validation.

Line 787: It is not clear whether the on-ground tests will reveal the root cause of the hot pixel issue. The problem was not revealed during the pre-flight tests, so it would be good to know what changes are planned in the new experimental setup compared to the previous one.

Technical corrections

Line 157, Table 3, last two lines: I would rewrite it as (a÷b) e- rms or (a±$\Delta$) e- rms

Lines 805, 809, 817, and 819: italicized fonts look strange in some of PDF viewers.

[Figure]

Fig. 1. Errors introduced by using an sliding profile approach to the peak position retrieval from 16×25 ACCD detector signals without hot pixels and with hot pixels excluded from consideration

---

## Author Response (AR2)

**Response to Referee Comment #1 on**

*Characterization of dark current signal measurements of the ACCDs used on-board the Aeolus satellite*

The authors thank reviewer #1 for carefully reading the first author's response. In the following, referee comments are repeated in green and answers by the authors are provided directly below in black.

**General comments:**

This is a second review. In general, I'm satisfied with the answers and I think that the article can be published with minor corrections indicated below. However, I'd like to address the questions of the authors they posed in the reply regarding the test I made to show that the retrieval procedure could define the peak position on the ACCD with a sufficient accuracy even if the hot pixel information were not used at all.

The authors write "*The figure does not consider the full Mie fringe shape which is spread over all 16 ACCD pixels – even at the edges of the ACCD the values do not approach zero. For simulation studies, this could be approximated with an Airy function.*" As for the fringe shape spread over all 16 ACCD pixels, the previous simulation used a Gaussian which was also spread over all pixels. The halfwidth of this Gaussian was selected in such a way that the shades of grey in Fig. 1b of (doi:10.5194/amt-2020-458-RC1) resembled those of Fig.1 of the manuscript. For the second exercise shown below (Fig. 1), I took an Airy function and compared the results with Gaussian. One has to note that in general the sliding profile approach is not sensitive to the shape of the function as long as the function varies at the considered interval and its shape is known with high accuracy.

Then, the authors write that "*On top of that, also the spectrally broad bandwidth Rayleigh signal and the solar background are part of the Mie signal.*" The broad features and constant offsets do not affect the peak position retrieval accuracy if the sliding profile approach is used.

The next comment is important, though: "*Apart from that, it is not clear which values for the atmospheric signal levels and hot pixel amplitudes were used in your simulation. Typical values for the Mie channel are 15 LSB and 5 LSB (both measurement level) for the atmospheric signal level and the hot pixel offset, respectively*". Indeed, the discretization is important and in the previous exercise this was not taken into account properly, I apologize for overlooking it. If one assumes that the peak value of Mie detector is 15 LSB, then the absolute errors of peak position retrieval in units of ACCD detector row change from ~1.5e-3 to ~1e-1 that corresponds to ~2 m/s (Fig. 1). This is somewhat smaller than the random errors reported for Aeolus wind product, and the comparability of the profiles retrieved for a "healthy" detector with those retrieved from the simulations with hot pixels excluded tells us that the approach of skipping the hot pixels proposed in the first review is still valid. I do not require to include more discussion on this topic than what is already included in the present version of the manuscript (lines 441-444), it's just to draw an attention to this technique that works surprisingly well for different physical phenomena.

**Response to General Comments:**

Thank you very much for your comments. It is highly appreciated that you addressed our points on your simulation study to mitigate hot pixel effects in the Mie channel and even extended it. Looking at your Fig.1, the comparability of the curves, especially for the Airy function with and w/o hot pixels, indeed looks very convincing. It seems to us that the sliding profile approach might be worth to be considered for future studies to improve the wind retrieval of the Mie channel.

**Specific comments:**

*Lines 98-103: new text states that the read-out noise values were determined during pre-launch tests and then the discussion is based on this value measured on the ground. I understand that normally the noise of the amplifier should not change, but are there any estimates for an onboard read-out noise?*

The determination of the total read-out noise requires special test modes which cannot be performed in space due to technical limitations. As a result, only numbers for the total read-out noise based on pre-launch tests are available. However, it is expected that these numbers have not changed significantly in-orbit. Specific measurements for the total noise (read-out noise + dark current noise) could be performed during periods with no laser operation, but are still under evaluation. We have no indications of enhanced noise sources for the detector.

The manuscript was changed as follows:

> available avalanche photodiodes or photomultipliers. In the case of Aeolus, special CCDs are used, so-called accumulation CCDs. This allows the accumulation of backscatter atmospheric signals for consecutive laser pulses already on the chip in a dedicated memory zone to reduce the impact of read-out noise. The total read-out noise was determined during pre-launch tests to be in the range between 3.9 e- and 4.7 e- root-mean-square (rms) error around a zero mean (Reitebuch et al., 2018).
>
> 100   Note that the determination of the total read-out noise requires special test modes which cannot be performed in space due to technical limitations. However, there are no indications that the total read-out noise has significantly changed in-orbit.

*Lines 197: I would specify the actual numbers for the proton fluxes*

Information about the proton energy and fluence levels were added to the manuscript.

190    vacancy-interstitial pairs. Most of the pairs recombine but some of them may form stable displacement damages in the lattice. Displacement damage can lead to a degradation of the CTE and an increase of the dark current. So-called "hot pixels", pixels with enhanced dark current signals over a longer period of time, may evolve. In addition, displacement damage may also introduce burst noise, e.g. Random Telegraph Signals (RTS)-noise. RTS noise causes the dark current to change its state between two or more discrete levels at random and unpredictable times (Hopkins and Hopkinson, 1993; Smith et al., 2004).

195    Hot pixels in combination with RTS phenomena were also observed for the CCD detectors of the Global Ozone Monitoring by Occultation of Stars (GOMOS) instrument on-board ENVISAT (Keckhut et al., 2010). In the framework of the Aeolus ACCDs development, proton tests (even at higher radiation doses as seen in-orbit) at different energy and fluence levels (for 30 MeV: $2 \cdot 10^9$ protons/cm², $1.35 \cdot 10^9$ protons/cm²; for 100 MeV: $4.2 \cdot 10^9$ protons/cm², $2.7 \cdot 10^9$ protons/cm²) have been performed to evaluate the probability of occurrence of such hot pixels and RTS pixels at an operating temperature of -30 °C

200    showing the presence of one-post irradiation RTS pixel. However, it has to be noted that the operation mode with regard to the timing settings during the tests was not fully comparable with the settings used in-orbit. Moreover, the dark signal acquisition and the applied post-processing sensitivity was not optimized to detect low frequency and low amplitude dark signal variations as observed in space. Transient radiation effects occur due to ionization-induced generation of charges within the CCDs and do not cause lasting damage. However, these effects might be visible as spurious signal spikes on one or more pixels and thus,

Lines 199-200: what exactly was different and how this should affect the results?

The dark signal acquisition during the test was not optimized to track the same low frequency dark signal variations as observed in-space (only 512 observations were acquired). In addition, a different timing diagram compared to in-orbit was applied to operate the ACCD. This had the effect that the residence time of the charges in the memory zone was different. Finally, the applied post-processing sensitivity algorithm was not good enough to detect dark signal anomalies with amplitudes as low as observed in-orbit. All these points limit the significance of the tests when discussing in-orbit dark signal anomalies. This information was added to the text (see screenshot above).

*Lines 301-303: "a more sophisticated" and "a simple 3-sigma" look strange in one sentence.*

You are right. The sentence was rephrased as follows:

The motivation for a detailed characterization of permanent dark current anomalies is twofold: a) it supports investigations for the underlying root causes of the hot pixel issue and b) the number and magnitude of dark signal shifts define the impact on the wind observations and the DUDE correction. In order to fulfill b) it is not only necessary to differentiate between normal and hot pixels but also to exactly characterize hot pixels

305 . The presented algorithm is  capable of detecting temporal

Lines 441-444: "each pixel indispensable in the wind retrieval". This is strange. Even though the authors use a different approach than the one used in my exercises, the information content of the whole row is still large, and the absence of a single pixel should not dramatically change the picture. One can carry out a simple "Gedankenexperiment", which usually is proven in real life with the real data. Imagine that a human eye observes a continuous function, or a continuous shape, or a graphic pattern. Next, let's imagine that a small patch covering 5% of the image is applied. No doubt that the brain will recover the original shape from this picture, especially if the shape is known. Of course, the bigger the patch, the poorer the accuracy, but for the objects like those shown in Fig. 1 of the manuscript a loss of up to 10% of information should not be critical. If so, the programming algorithm should exist which mimics the brain's algorithms and recovers the shape. The realization depends on the task, but the general idea should be clear. I believe that it can be realized in the framework of the authors' method, too. In

any case, since the "hot" pixels are not completely erroneous, the authors' approach works, and this itself is a good achievement.

It is important to mention the information content of each pixel strongly depends on the pixel position w.r.t to the ACCD column. For the Mie channel the key information is contained in the three to four central ACCD pixels covering the fringe. For the Rayleigh channel the pixels covered by the two Rayleigh spots pixels (three to four pixels for each spot) are important for the wind retrieval. To be able to omit hot pixels from the wind retrieval, one must know the model function at a very high accuracy. Here, also the high sensitivity of the measured wind speed towards change in the response of the instrument comes into play (Mie: 1 pixel → 17.7 m/s LOS). So, errors in the model function have large impact on the error of the wind retrieval. But, as mentioned in the manuscript, the hot pixels still contain information which can be used in the wind retrieval in combination with regular dark signal calibration measurements.

*Lines 497-501: In general, the operational retrieval should not exclude the post-processing, during which the data quality might be improved. At this stage, one can impose different physical constraints and analyse/correct the retrieved data using ancillary information. Even in the current setup we observed an evolution of the products associated with the corrections introduced after validation.*

As mentioned in the text, the NRT processing chain is setup in way that it is strictly sequential which implies that for instance no wind-speed information from the L2B products can be used in the dark signal correction that is part of the L1B processor. To make this possible a complete re-design of NRT processing chain would be required which is not possible due to timeliness requirements of the wind products and operational constraints. For the reprocessing, we approach the methodology proposed by the reviewer and use ancillary information. Here, the flexibility is higher and information about wind errors can be used to analyse and further mitigate dark signal anomalies. It is planned to summarize product improvements achieved in the reprocessing in a dedicated manuscript in the future.

*Line 787: It is not clear whether the on-ground tests will reveal the root cause of the hot pixel issue. The problem was not revealed during the pre-flight tests, so it would be good to know what changes are planned in the new experimental setup compared to the previous one.*

We assume that your comment is related to line 777 and not line 787. It is true that previous on-ground tests did not reveal the problem. However, the performed on-ground tests were not designed in a way to allow to distinguish between radiation-induced and CIC hot pixels. As mentioned in the discussion of the manuscript, on-ground sensitivity tests with identical ACCDs where the operating temperatures and clocking parameters of the ACCD are varied would be needed. However, for safety reasons such tests cannot be performed in-orbit. Moreover, setting up such a campaign is not straightforward as no spare ACCD of the same batch as the in-orbit one is available. But, it is expected that the knowledge gained from this detailed in-orbit dark signal investigations will help to improve on-ground testing CCD testing campaigns of future space missions such as Aeolus follow-on.

*Line 157, Table 3, last two lines: I would rewrite it as (a÷b) e- rms or (a±Δ) e- rms*

Thank you very much for the suggestions but e authors decided not to change the format of the noise specifications in Table 1 of the manuscript.

*Lines 805, 809, 817, and 819: italicized fonts look strange in some of PDF viewers*

The italicized font as headers for these paragraphs is part the AMT Word template and thus, should not be changed.

**Response to Referee Comment #3 on**

*Characterization of dark current signal measurements of the ACCDs used on-board the Aeolus satellite*

The authors thank reviewer #3 for carefully reading the manuscript and providing useful comments. Although it is the second review iteration, the reviewer came up with a lot of new points. Nevertheless, the authors tried their best to capture the comments as best as possible in the revised version of the manuscript. However, it also has to be considered that the focus of the manuscript is not solely on the technical aspects of the hot pixel generation but also on the effects on the quality of the measured winds. In the following, referee comments are repeated in green and answers by the authors are provided directly below in black.

**General comments:**

As well described in the paper, the working principle of ACCD implies the accumulation of collected charges into a given column of pixels into a transfer row. The charge transfer in the memory zone follows.

1/ Does the charge transfer occur from the transfer row to the range gate #1 before being transfer from #1 to #2 until #25 or does the first transfer row is stored in range gate #25? This question points out the CTI issue in the memory zone that could be impacted by the radiations. Depending on the operation mode, DCNU could reveal the worst degradation for memory gates resulting from several transfers. Moreover, it appears that the charge retention time varies from a range gate to another. Has the range gate DCNU been investigated by looking at the dark signal with no transfer from the CCD?

2/ The author also well described that the memory zone plays a role in the global dark current. I agree with this statement, which is explained in the paper by the generation centres into the storage structure. However, regarding the charge retention time (i.e. depending on the range position) which can reach 0.4s, it appears that the leakage current from these storage nodes can impact the global dark current signal. The authors are invited to give their analysis regarding this leakage which could lead to a loss of collected charges and therefore implying a decrease of the signal and even partially compensate for the dark current increase from the CCD. As a complex structure, the quantification of such DC sources is not required. However, the paper could include a detailed description of the dark current sources in the ACCD to provide a better overview of the underlying mechanisms.

**Response to General Comments:**

1)

The charge for each range gate is binned into one row of the memory transfer section then moved down the memory transfer section followed by the binned charge from the next range gate until all the range gate signals have been captured. When the memory transfer section is filled with signals from this first laser pulse the set of charges are transferred sideways into the corresponding pixels of

the memory storage section. This process is repeated until signals from typically 20 laser pulses have been captured. The accumulated signal is then read out. Please note that Fig.1 (left) of the manuscript is a simplified sketch which only shows the 16 storage rows of the memory zone. The charge retention time is therefore essentially the same for all range gates at approximately 20x the laser pulse period. As a result, no range gate dependent features of the DCNU were observed. Uniformity of dark signal in various operating modes was investigated in pre-flight testing but the instrument does not have the flexibility to do this in flight.

2)

As explained above the charge retention time is not a strong function of the range gate number but is essentially the same for all range gates. The total duration of the acquisition of echoes for each pulse is much less than the pulse repetition period and signals from typically 20 pulses are added.

A leakage mechanism leading to loss of charge stored in a CCD pixel is not often considered and the analysis of this effect is certainly beyond the scope of this paper. Here, we only want to state the several on-ground tests were performed to assess the "accumulation efficiency" of the Aeolus ACCDs. During these tests the signal level at the CCD output are compared when charge generated by several LED pulsed was accumulated by two different methods. The first was a simple integration of 50 pulses in the image section followed by a standard readout. The second was transfer of the signal from each of 50 identical LED pulses individually to the memory storage section for accumulation; with readout of this accumulated signal. The signal level from a large number of exposures was averaged in this test which was performed at three different accumulated signal levels of 1k 10k and 100k electrons per pixel. It is believed that the lack of a difference between the signal levels obtained in these two modes provides some evidence that we did not have any significant recombination when accumulating charge in the memory section but it is not fully conclusive.

**Specific comments:**

*Lines 101-105: This part could include the CVF to account for the number of collected charges. It is specified later. Retention time might be more appropriate rather than residence time. I agree with the statement that the noise is dominated by the read-out noise. The authors are invited to specified "before the mission" or "before radiation exposition".*

To be consistent with the noise estimation, the mean dark signal is now also expressed in e-/s. The authors do not see a problem using residence time to describe the amount of time of charges in the memory zone.

100 Note that the determination of the total read-out noise requires special test modes which cannot be performed in space due to technical limitations. However, there are no indications that the total read-out noise has significantly changed in-orbit. Considering the PRF of 50.5 Hz, and 19 pulses per measurements and conversion factors for the Mie and Rayleigh channel of 0.68 LSB/e- and 0.44 LSB/e-, the in-orbit dark current signal rates are 0.55 e-/s – 0.72 e-/s0.49 LSB/s and 0.24 LSB/s for the Mie and Rayleigh channel. Given the Poisson distribution of the dark charges these values correspond to 0.75 e- – 0.86 e- rms

105 dark current noise for a residence time of the signals in the ACCD of one measurement which is 0.376 s. Thus, it becomes

*Lines 105-110: Is this ACCD from Teledyne a COTS or a custom design? Can the reference of this ACCD be specified here? The number of lost charges (CTI) depends on the collected charges. The authors are invited to specify "at the typical integration time" or "based on typical operation".*

The CCD is designated CCD69 and is a custom design for Airbus D&S and ESA. There is no intent to make it commercially available. Charge transfer inefficiency results in charge being delayed from the correct pixel and read out in the corresponding following pixel of the row or column. Detailed measurements of any charge loss were performed at a range of signal levels by Teledyne –e2v before delivery of the devices to Airbus.

The manuscript was changed as follows:

[Figure]

*Lines 110-115: The authors are invited to compare the pixel pitch to the required fringes resolution. Does smaller pixel pitch could work and what could be the limit (Full Well Capacity and/or sensitivity). Charge accumulation can be preferred to signal accumulation.*

Analysis regarding the trade-off between pixel size and the full well capacity were performed during pre-launch tests. However, the description of the outcome of these tests is considered to be beyond the scope of this paper.

*Lines 115-120: The authors are invited to specify the operating mode of the CCD (rolling shutter) as well as how the memory zone is operated from a frame to another. Moreover, it looks like the limiting parameter in this application is the ADC frequency which implies storing several columns of accumulated frames in a memory zone. A brief description of this part would be appreciated.*

The operation mode of the CCD is described in the comments above and is also explained in the manuscript. The description of the operation mode in the manuscript was slightly extended to improve the understanding of the CCD (see screenshot below). However, it should be noted that the operation mode is not a conventional rolling shutter. Moreover, it is not the ADC performance which limits the pixel readout rate but the increase in the readout noise of the CCD output circuit with frequency and therefore bandwidth (Janesick, 2001).

It illustrates how the two circular Rayleigh spots from the FPI and the Mie fringe from the FIZ are imaged on the ACCDs imaging zone. In the imaging zone the atmospheric return signal is integrated over time based on the settings for the vertical

120 range gate timings. In Aeolus operations the range gate timings can be varied from 2.1 µs to 16.8 µs which correspond to a vertical sampling of 250 m to 2000 m, respectively, considering the 35° off-nadir viewing angle of the instrument. Subsequently, the  charges of the imaging zone are pushed downwards,  accumulated in the transfer row and then moved down into the transfer columns of the memory zone followed by the charges from the next range gate. The image zone is completely shifted within 1.0 µs. In the memory zone of

125 the ACCD each of the 25 rows corresponds to one vertical range gate of the atmospheric profile. Once the signals of all range gates are acquired in the transfer section of the memory zone, the  charges are horizontally shifted from the transfer columns into corresponding pixels of  the storage columns of the memory zone. This concept allows on-chip signal accumulation over multiple successive atmospheric returns to the so-called "measurement" level. The number of accumulated pulses can be varied between 1 and 50. For the herein analyzed dark current measurements the number of pulses was 19 until

130 Jan 2019 and then 18 to avoid a potential conflict in the onboard data management. The resulting residence time of the signals in the memory zone is on the order of 0.4 s, considering the PRF. After each accumulation sequence, the charges of the memory zone are read-out via the read-out register at a very low frequency of 48 kHz to minimize read-out noise and are further transferred to the Detection Electronics Unit (DEU) (Reitebuch et al., 2018). Here, the accumulated charges are digitized with 16-bit accuracy and converted into units of Least Significant Bits (LSB). The conversion rate of this process, also called

135 radiometric gain, is about 0.68 LSB/e- and 0.44 LSB/e- for the Mie and Rayleigh channel, respectively.

*Lines 135-140: Do the virtual pixels allow to monitor the applied offset?*

ALADIN uses two "overscan" pixels, which are generated at the end of each row by applying more clock pulses to the register than the total number of physical register pixels, to monitor the so-called Detection Chain Offset (DCO). These pixels provide a nominally zero charge reference level which can be used to adjust the working point of the ADC to ensure that even in the presence of random noise no negative values are output.

*Lines 145-150: What limits the size of the memory zone? Does a direct quantification of the transfer row possible? The authors are invited to specify the link with the readout circuit.*

The size of the memory zone was defined on the basis of the vertical spatial resolution required for the instrument. 25 rows give 25 range bins within the atmosphere. An increase in the number of range bins is being proposed for future instruments but this tends to be at the expense of SNR as the echo signal is spread over more pixels in case higher vertical resolutions are applied.

*Lines 160-165: "the random as well as systematic error budget of CCD based measurements." Noise or additional dark current shot noise could be preferred to random. Moreover, it does not appear clear to me what the budget means here. "the noise contributions are related to the signal itself" Total noise comprises DC noise and read-out noise. "The noise of Aeolus signals is dominated by the Poisson distributed shot noise as the levels for dark current and read-out noise" The dark current noise is also a shot noise. This sentence needs to be rephrased.*

The authors intended to say that the systematic and random errors of CCD based lidar measurements usually depend on the dark signals of the CCD. To be clear, the sentence was rephrased (see screenshot). Random error is considered to be the correct antonym for systematic error which is why this was not changed in the manuscript. Moreover, the noise arising from the signal itself was specified as photon shot noise in the manuscript.

> Even in the absence of light, a relatively small amount of thermally generated electrons is collected in the CCD. This is known as dark current and causes a non-negligible background signal on CCDs. In general, dark current signals play an important
> 165 role for  random as well as systematic error  of CCD based measurements.
> On the one hand, the dark signal affects the random error budget by dark current signal noise. For a typical optical CCD instrument, the noise contributions are related to the signal itself, the noise of the dark current signal, and the read-out noise. The noise of Aeolus signals is dominated by the Poisson distributed photon shot noise as the levels for dark current and read-out noise are very low (see Table 1). Thus, the technique used for Aeolus is referred to as "quasi-photon" counting.

*Lines 165-170: "Thus, the technique used for Aeolus is referred to as "quasi-photon" counting." It does not appear that this device can integrate and measure a single photon. The noise is dominated by a readout shot noise, but photon-counting required other characteristics that are not met here. Systematic errors could be corrected easily with an appropriate offset correction. The main issue may lie in the need to evaluate the DC along with the mission as well as the DC non-uniformity along with the pixel array. The authors are invited to introduce these concepts as well as the possible impact on the reduction of the dynamic range which might impact the minimum flux detection that could affect the mission.*

In addition to the Poisson photon shot noise from the signal itself, the main electronic noise contributor from the ACCD is read-out noise (4 e- − 6 e- rms). For number of P=19 laser pulses the number, the noise for one laser pulses can be calculated by dividing the read-out noise by $\sqrt{P}$ which results in a value of about 1 e-/pixel/laser pulse. As 1 electron-hole pair is produced for 1 photon at the wavelength of 355 nm (Janesick, 2001) , the equivalent noise is in the order of 1 generated photon. Hence, this technique is referred as "quasi-photon" counting due to the low read-out noise.

Systematic errors originating from dark signal anomalies in the memory zone are successfully corrected by the introduction of dedicated dark signal calibration measurements that performed four times per day. This concept is introduced in Sec. 3.3 of the manuscript. The dark signal calibrations measure the DSNU of the 25x16 memory zone pixel array which is used to correct subsequent measurement signals.

*Lines 170-175: "Dark current anomalies" Dark current increase and RTS are expected for this kind of mission. The term dark current increase should be preferred to Dark current anomalies where or not it has an RTS behaviour or not. Main comment: DC origin and impact on the measurement must be explained with the operation mode used by the instrument. A clear description of the DC accumulation over the SNR can highly facilitate understanding. More importantly, the accumulated dark charges are stored in the memory area for almost half a second (0.4s). such retention time makes the leakage current very important in this structure and can potentially lead to a reduction of the total stored charges implying a negative offset on the signal. Authors are invited to explain how important this leakage current is on the output signal and, if applicable, how it can to a certain extend compensate the DC.*

The dark current anomalies refer to an increase in dark signal and in particular hot pixels which are apparently not consistent with the expected increase due to radiation. However, in the manuscript the information is added that dark signal anomalies are defined as dark signal increase (see screenshot below). The impact of dark signal anomalies on the Aeolus wind measurements is depicted in Sec. 3.3 of the manuscript where the effects on both measurement channels are discussed.

*Lines 170-175: Why CMOS imagers have not to be used for such applications? Space qualification history and legacy from other missions need to be highlighted.*

The detector was designed and manufactured over 20 years ago when the maturity of CMOS technology was much lower. The mission was unfortunately delayed due to problems with the development of the laser. Even today there is no obvious architecture for a CMOS imager to achieve the binning and accumulation of laser echoes apart from the implementation of CCD structures in a CMOS technology. Please note that there is also the PhD thesis "Estimation and modelling of key design parameters of Pinned PhotoDiode CMOS image sensors for high temporal resolution application by Alice Pelamatti (Pelamatti, 2015).

*Lines 175-180: A clear distinction between solar events, trapped particles, and cosmic rays should be made here. Cosmic rays are also referred to as heavy ions due to their mass.*

The reviewer's suggestion was introduced as follows:

180   harsh radiation conditions, radiation-induced effects are an important issue. In particular, the effects of high-energy particles
      such as cosmic electrons, ions, neutrons, and protons passing through CCDs have to be considered (Hopkinson et al., 1996).
      These particles can be categorized into particles trapped in the Van Allen radiation belt (Feynman and Gabriel, 2000) and the
      transient environment. Particles trapped in the Van Allen belt are composed of energetic protons, electrons as well as heavy
      ions and the transient radiation consists of galactic cosmic rays and solar events are mainly of solar and interstellar origin and
185   are often trapped and accumulate in the Van Allen radiation belt (Feynman and Gabriel, 2000). The geographic region where
      the inner Van Allen belt comes closest to the Earth's Surface is called South Atlantic Anomaly (SAA). The SAA is a region
      of reduced magnetic intensity where satellites in Low Earth-Orbits (LEO) (< 1000 km altitude) are exposed to strong radiation
      (Anderson et al., 2018) and thus this region is of potential harm for satellite measurements particular interest. Typically, the
      SAA is situated at an altitude of 200 km to 800 km over the Earth's surface (Nasuddin et al., 2018). A significant increase of
190   dark signal levels in the region of the SAA has been observed on the CCDs of the Hubble Space Telescope which is operated

*Lines 180-185: SAA, a "region of particular interest" The authors are invited to briefly describe this intertest. Either it is for scientific purposes or a better understanding of the impact on electronics. The Hubble dark current increase rather lies in the repetitive flyby of the area implying an increase of the deposited dose. Transient effects are photogenerated charges coming from the ionizing radiation during the flyby.*

The author's intention was to state that the SAA is a region where low-orbit satellites are exposed to higher radiation than usual and thus might affect the measurements of the satellite. For Aeolus measurements these effects are discussed in Sec. 4.3 of the manuscript. Please notice that some previous European space LEO missions have been dramatically impacted by displacement damages generated by SAA larger protons fluence, such as GOMOS on board Envisat ESA mission (Keckhut et al., 2010) and MIR channel on board Spot 4 / 5 CNES missions. The section about the SAA was changed as follows:

| 185 | . The geographic region where
the inner Van Allen belt comes closest to the Earth's Surface is called South Atlantic Anomaly (SAA). The SAA is a region
of reduced magnetic intensity where satellites in Low Earth-Orbits (LEO) (< 1000 km altitude) are exposed to strong radiation
(Anderson et al., 2018) and thus this region is of potential harm for satellite measurements . Typically, the
SAA is situated at an altitude of 200 km to 800 km over the Earth's surface (Nasuddin et al., 2018). A significant increase of |
| 190 | dark signal levels in the region of the SAA has been observed on the CCDs of the Hubble Space Telescope which is operated |

*Lines 185-190: Dielectric materials could be preferred to oxide layers.*

Agreed, it is at the interface between the oxide and nitride layers in the gate dielectric where most charge is trapped.

In general, radiation-induced effects can be categorized into three groups: ionization damage, displacement damage and
transient effects. Ionization damage can lead to an increase of trapped charges in the dielectric materials  of the
195 CCD and thus, may lead to an increased dark current and to a shift in the optimum operating voltages of the CCD. Displacement
damage is caused by energetic particles (mainly protons) passing through the CCDs which may displace atoms from their

*Lines 190-195: Vacancy-interstitial pairs, also called Frenkel pairs recombine at room temperature. It must be specified here that it is still the case at -30. "Random Telegraph Signals (RTS)-noise" RTS is a signal – as specified in the name - not a noise. This terminology lies in the description of the signal from the mathematical point of view. Therefore, the preferred terminology is DC-RTS or RTS. Some papers used RTN for random telegraph noise when studying transistor RTS (trap-detrap mechanisms in the canal) and DC RTS (generation centres) at the same time. Level should be preferred to state which rather lies in a defect configuration.*

The information about the Frenkel pairs was added to the manuscript (see screenshot below). We agree that RTS are signals, however, the expression RTS noise is an industry standard terminology and is widely understood. However, to be correct the terminology in the manuscript was changed as follows:

In general, radiation-induced effects can be categorized into three groups: ionization damage, displacement damage and
transient effects. Ionization damage can lead to an increase of trapped charges in the dielectric materials  of the
195 CCD and thus, may lead to an increased dark current and to a shift in the optimum operating voltages of the CCD. Displacement
damage is caused by energetic particles (mainly protons) passing through the CCDs which may displace atoms from their
lattice and create vacancy-interstitial pairs, also referred to as Frenkel pairs (Janesick, 2001). Most of the pairs recombine (also
at the ACCD operating temperature of -30° C) but some of them may form stable displacement damages in the lattice.
Displacement damage can lead to a degradation of the CTE and an increase of the dark current. So-called "hot pixels", pixels

*Lines 195-200: "In the framework of the Aeolus ACCDs development, proton tests (even at higher radiation doses as seen in-orbit) have been performed to evaluate the probability of occurrence of such hot pixels and RTS pixels at an operating temperature of -30 °C showing the presence of one-post irradiation RTS pixel. However, it has to be noted that the operation mode regarding the timing settings during 200 the tests were not fully comparable with the settings used in-orbit." If neither results nor conclusions can be extracted from this test campaign, it should be withdrawn from the paper. It could eventually be mentioned in the discussion section.*

The authors decided to keep the information about the proton tests in the manuscript mainly to highlight the importance of proper on-ground testing campaigns for potential follow-on missions.

*Lines 200-205: "Transient radiation effects occur due to ionization-induced generation of charges within the CCDs and do not cause lasting damage. "I do not agree with this statement. SEE does result in remaining TID and DDD and can even lead to latchup. "quite efficiently shielded from ionization damage." The authors are invited to specified until which TID level. Globally, TID and DDD exposition are not reported in the paper.*

Teledyne-e2v CCDs, unlike CMOS detectors are not susceptible to latchup. There are no NPNP thyristor like structures between power supplies.

In the framework of the Aeolus development, simulations have been performed to determine the shielding for the six instrument faces ($\pm$X, $\pm$Y, $\pm$Z) as seen by the detector. Equivalent shielding figures from 2 mm to 8 mm per face have been found for the most exposed ACCD, plus the 2.5 mm thickness BK7 window. The TID and TNID levels are respectively about 0.3 krad(Si)/year and 5E6 MeV/g(Si)/year for 400 km circular orbit, maximum solar activity being considered for the whole mission duration. Please notice that Aeolus altitude has been decreased to 320 km, reducing even more the radiation levels. However, it is considered to be beyond the scope of this manuscript to confront the AMT readers with TID and DDD exposition levels.

*Lines 210-215: Integration time could be preferred to timing settings.*

It is preferred to keep the term "timing settings" as "integration time" is used in the manuscript to describe the vertical range gate resolution of the manuscript.

*Lines 210-215: RTS must be preferred to transient events.*

This paragraph describes the impact of transient events, and not RTS signals on the Aeolus measurements.

*Lines 255: "at measurement level." This term must be specified> Does it refers to data processing or analogue correction?*

The term "measurement level" is introduced in Sec. 2.1 of the manuscript and describes the temporal resolution (0.4 s) of the signals after the on-board accumulation of charges in the memory zone. A reference to Sec. 2.1 was added to the sentence:

> 260 current measurements is restricted to periods where the laser is operated in a lower mode and not emitting laser pulses
> In this paper, DUDE measurements obtained from the quasi raw Aeolus L1A data products were analyzed (Reitebuch et al., 2018). The specific L1A data product is generated after each DUDE characterization and contains geo-located but unprocessed dark current signals of both channels for 25 range gates and 16 pixels at the measurement level (as introduced in Sec. 2.1), i.e., in the same format as nominal wind lidar measurements which allows for a DSNU characterization. In a first step, the DCO
> 265 was subtracted from each pixel value at measurement level. Next, the measurements were averaged to observations by calculating the mean over the measurements per observations.

*Lines 391: Transition must be preferred to dark signal spike/peaks.*

Agreed. The manuscript was changed accordingly:

**3.2 Detection of transient dark current anomalies**

As outlined in Sect. 2.2 transient effects which cause spurious dark signal  transitions may occur on the CCD. In contrast to the detection of permanent dark current anomalies (see Sect. 3.1), the detection of spurious spikes is performed at measurement level. Typically, transient events appear as isolated signal peaks only present in one measurement and show a

*Lines 460-465: Could also lie in a small temperature shift during the dark measurement and the collected data.*

The ACCD temperatures were investigated and no significant temperature shift could be observed.

*Lines 465-470: 4 minutes is very short to account for RTS. Most of the high RTS amplitudes cannot be seen in such a small period. Additional references on RTS in the CMOS literature might give a global overview of expected RTS behaviours.*

Yes, it is correct that 4 minutes are short to analyse RTS. This is why, we analysed the concatenated data stream of DUDE measurements which allows to derive an estimate about the RTS amplitudes. Moreover, for some hot pixels it also possible to see RTS effects in the wind measurement signals (see Fig. 8 (bottom) of the manuscript). The paragraph about RTS in Sec. 2.2 of the manuscript was extended and additional references were added:

200   current signals over a longer period of time, may evolve. In addition, displacement damage may also introduce burst noise, e.g. Random Telegraph Signals (RTS) . RTS  causes the dark current to change its state between two or more discrete levels at random and unpredictable times (Hopkins and Hopkinson, 1993; Smith et al., 2004; Srour and Palko, 2013). RTS is observed in CCD as well as in Complementary metal-oxide-semiconductor (CMOS) devices (Goiffon et al., 2009; Woo et al., 2009). The time spent on the different levels can be in the range between seconds to days and also the RTS

205   amplitudes typically cover a wide range (Virmontois et al., 2013; Liu et al., 2020; Capua et al., 2021). Hot pixels in combination with RTS phenomena were also observed for the CCD detectors of the Global Ozone Monitoring by Occultation of Stars (GOMOS) instrument on-board ENVISAT (Keckhut et al., 2010) or for the CCDs used in BRITE Nano-satellite image sensors (Popowicz, 2018). In the framework of the Aeolus ACCDs development, proton tests (even at higher radiation doses as seen

*Lines 480-485: RTS can show period lasting from seconds to hours and even days and this is much more important as the temperature decrease. This is the main reason why the correction is not possible.*

Yes, it is correct that due to RTS pixels only a mitigation of hot pixel effects can be achieved.

490   dark current signal level. However, there remains a problem for the near-real-time (NRT) processing when dark current transitions occur between two DUDE measurements which may happen for RTS-type hot pixels. The uncorrected signal intensity (dashed red line) shows a dark current induced signal decrease of about 8.0 LSB at 14:15 UTC. Here, the dark current calibration based on the DUDE measurement from 13:15 UTC is still active. Thus, the dark current signal is overestimated and the dark signal corrected signal intensity (solid red line) shows the signal dip. This holds true until the new DUDE

495   measurement is performed and gets used for the dark current calibration of the orbit which starts around 20:30 UTC. Afterwards, the dark signal corrected signal intensity is again at the same level as before.

*Lines 540-545: "not perfectly linear." The data are linear, but the interesting parameter here could be the total mean dark current over the pixel array. Moreover, DDD and TID ranges must be specified and compared to other results in the literature if possible.*

To the authors the temporal evolution of hot pixels is not truly linear as there are periods with enhanced hot pixel rate. However, the overall tendency appears to be somewhat linear. It should be noted that this manuscript is more focused on the impact of dark current anomalies on the quality of

the wind measurements. Thus, the mentioned aspects are considered to be out of scope. However, it is planned to prepare a follow-on paper with a focus on the root-cause and the technical aspects after end of the mission lifetime. In the framework of this analysis, aspects like the evolution of the mean dark current, DDD and TID can be discussed.

*Lines 545-550: "the hot pixel generation rate does not change with time" Unless for RTS pixels…*

The assumption of the linear extrapolation is that the hot pixel generation rate stays at the same level (one new hot pixel every 14.68 days) throughout the mission lifetime.

*Lines 605-610: "RTS pixels show more than two levels." Usually the case for DDD-induced RTS.*

This is correct. The information with the corresponding reference was added to the text.

> **4.2.1 RTS characteristics**
>
> The majority of the hot pixels which were defined as RTS pixels show more than two levels which is usually the case for displacement damage induced RTS pixels (Virmontois et al., 2011). RTS characteristics with two distinct levels were only
>
> 26

> 635 observed in 27 % of the cases. Apart from that, it is apparent that the RTS levels are quite different from each other. An

*Lines 605-625: These on-flight observations are consistent with RTS observed in other imagers CCD/CMOS. References here could support these observations and confirm that these results are not surprising but highlight the need to perform more investigation on RTS in imager for space applications. RTS behaviors present an infinity of different shapes that cannot be described. This is the reason why other studies use high-resolution imagers to outline major trends on thousands of RTS. See recent RTS studies in CMOS imagers.*

It is correct that these observations are consistent with other missions. This section was changed as follows:

> **4.2.1 RTS characteristics**
>
> The majority of the hot pixels which were defined as RTS pixels show more than two levels which is usually the case for displacement damage induced RTS pixels (Virmontois et al., 2011). RTS characteristics with two distinct levels were only
>
> 26

> 635 observed in 27 % of the cases. Apart from that, it is apparent that the RTS levels are quite different from each other. Overall, the observed RTS features are consistent with RTS observed in other CCD-based satellite instruments such as the CoRoT or PARASOL mission (Gilard et al., 2010; Bardoux et al., 2017). For both missions a continuous increase of hot pixels (including RTS) throughout the mission lifetime could be observed.

*Lines 705-710: CIC does exhibit RTS but with a much smaller RTS amplitude. The different RTS signatures between DDD-induced RTS and CICI-RTS can be made base on the RTs amplitudes. DDD-induced RTS amplitudes are more important. Reported results show large RTS amplitudes that can be linked to DDD RTS. Another method could be to use temperature dependence to extract activation energies. Again, the authors are invited to see the last published RTS studies on CMOS imagers.*

The authors were not aware of the fact that it is possible to distinguish between CIC and DDD-induced effects based on the RTS amplitudes. However, from Fig. 14 of the manuscript which shows the RTS amplitudes it does not seem straightforward to identify a threshold for the amplitudes which would allow a categorization of RTS pixel into CIC- and DDD-induced RTS.

*Lines 750-755: Can a high-resolution imager with a smaller pixel pitch be foreseen to maintain sufficient sensitive pixels along the column while disabling RTS ones?*

Alternative architectures are being considered for a future LIDAR instrument which will provide a near-real-time hot pixel map to minimise the impact of RTS type hot pixels.

References:

Janesick, J. R.: Scientific Charge-coupled Devices, SPIE Press, 936 pp., 2001.

Keckhut, P., Hauchecorne, A., Blanot, L., Hocke, K., Godin-Beekmann, S., Bertaux, J.-L., Barrot, G., Kyrölä, E., van Gijsel, J. a. E., and Pazmino, A.: Mid-latitude ozone monitoring with the GOMOS-ENVISAT experiment version 5: the noise issue, 10, 11839–11849, https://doi.org/10.5194/acp-10-11839-2010, 2010.

Pelamatti, A.: Estimation and Modeling of Key Design Parameters of Pinned Photodiode CMOS Image Sensors for High Temporal Resolution Applications, Federal University of Toulouse Midi-Pyrénées, Toulouse, 214 pp., 2015.